# A Continual Learning Perspective to Entropy Regularized Deep Reinforcement Learning

## Abstract

Research on Continual Learning (CL) tackles learning with non-stationary data distributions. The non-stationary nature of data is also one of the challenges of deep Reinforcement Learning (RL), and as a consequence, both CL and deep RL rely on similar approaches to stabilize learning, from the use of replay buffers to the choice of regularization terms. However, while dynamic neural architectures that grow in size to learn new tasks without forgetting older ones are well researched in CL, it remains a largely understudied research direction in RL. In this paper, we argue that Policy Mirror Descent (PMD), a regularized policy iteration RL algorithm, would naturally benefit from dynamic neural architectures as the current policy is a function of the sum of all past Q-functions. To avoid indefinitely increasing the neural architecture, we study PMD-like algorithms that only keep in memory the last $M$ Q-functions, and show that a convergent algorithm can be derived if $M$ is large enough. This theoretical analysis provides insights on how to utilise a fixed budget of Q-functions to reduce catastrophic forgetting in the policy. We implement this algorithm using a new neural architecture that stacks the last $M$ Q-functions as 3-dimensional tensors to allow for fast GPU computations. StaQ, the resulting algorithm, is competitive with state-of-the-art deep RL baselines and typically exhibits lower variance in performance. Beyond its performance, we argue that the simplicity and strong theoretical guarantees of StaQ's policy update makes it an ideal research tool over which we can further build a fully stable deep RL algorithm.

## 1 Introduction

Continual Learning (CL) moves from the usual i.i.d assumption of supervised learning towards a more general assumption that data distributions change through time (Parisi et al., 2019; Lesort et al., 2020; De Lange et al., 2021; Wang et al., 2024). A CL setting usually goes in pair with memory constraints, limiting the storage of past data. As it cannot access old data, it is desirable for a CL learner to possess i) stability, which is the ability to retain prior knowledge ii) plasticity, which is the ability to learn new knowledge. When training a neural network (NN), these two properties are typically conflicting and we talk of the plasticity-stability dilemma (Grossberg, 1988), which if not handled properly can lead to catastrophic forgetting (McCloskey & Cohen, 1989), i.e. a sudden loss of prior knowledge when trying to acquire new knowledge. To fight catastrophic forgetting, CL applies methods that can be categorized in three groups (De Lange et al., 2021): a) rehearsal methods (Robins, 1995; Rolnick et al., 2019) that keep samples from older data distributions to periodically retrain a learner on them, b) regularization-based methods (Li & Hoiem, 2016; Kirkpatrick et al., 2017; Zenke et al., 2017), that prevent an NN from changing too fast and c) parameter isolation methods (Rusu et al., 2016; Xu & Zhu, 2018; Yoon et al., 2018; Li et al., 2019) that grow an NN, usually while freezing older weights, to increase plasticity without compromising on stability.

Somewhat independently, deep RL has known rapid development in the past decade, achieving super-human results on several decision making tasks (Mnih et al., 2015; Silver et al., 2016; Wurman et al., 2022). RL is not always a true CL setting in the sense that an agent may not need to continually adapt to remain optimal, as most environments considered in RL are of a stationary nature, unlike in CL (Abel et al., 2023). However, deep RL and CL both face non-stationary datasets. This non-stationarity coupled with the use of neural networks makes deep RL very sensitive to hyper-parameters (Henderson, 2018) and makes its empirical behavior often poorly align

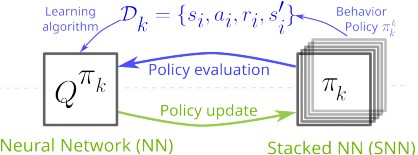

Figure 1: Overview of StaQ, showing a continual training of a Q-function (left), from which we periodically "stack" frozen weight snapshots to form the policy (right). See Sec. 5 for more details. At each iteration $k$, two steps are performed. i) Policy evaluation, where we generate a dataset $\mathcal{D}_k$ of transitions that are gathered by a behavior policy $\pi_k^b$, typically derived from $\pi_k$, and then learn $Q^{\pi_k}$ from $\mathcal{D}_k$. ii) Policy update, performed by "stacking" the NN of $Q^{\pi_k}$ into the current policy. The policy update is optimization-free and theoretically grounded (Sec. 4), thus only the choice of $\pi_k^b$ and the policy evaluation algorithm remain sources of instabilities in this deep RL setting.

with our theoretical understandings (Ilyas et al., 2020; Kumar et al., 2020; van Hasselt et al., 2018). As they face similar problems, deep RL and CL often end-up using similar remedies (see discussion in Sec. 2.4.3 of Lesort et al. (2020)). For instance, the rehearsal methods in CL have their counterpart in the use of replay buffers in DQN (Mnih et al., 2015) or SAC (Haarnoja et al., 2018). As for regularization-based CL methods, approaches such as EWC (Kirkpatrick et al., 2017) and its online version (Chaudhry et al., 2018) are extremely similar to TRPO (Schulman et al., 2015)—contrast the regularized cost of Eq. 5 in Chaudhry et al. (2018) with the Kullback-Leibler divergence (further referred to as $D_{\mathrm{KL}}$) in regularized policy update of TRPO, they are identical up to differences in loss functions between supervised learning and RL. However, while parameter isolation methods have been tested in *continual* RL settings (Rusu et al., 2016), their use in *single* task RL problems remains largely understudied.

In this paper, our main inspiration from the CL research community is the use of growing neural architectures to implement our policy update. Such neural architectures belong to the class of parameter isolation methods in the CL literature, which offer some of the best stability-performance trade-offs (see Sec. 6 in De Lange et al. (2021)). While the policy's growing architecture eliminates catastrophic forgetting in the policy by construction, it still seems to be present in the Q-function network (Sec. 6.3), causing instabilities of the learning process. We provide some ad hoc workarounds in this paper, but we believe several CL approaches—as those discussed in Sec. 7—could be helpful to further stabilize deep RL, and we hope that one side effect of this paper is to further encourage the transfer of ideas between the RL and CL communities.

Our policy update that uses a growing neural architecture is from a class of Policy Mirror Descent algorithms (PMD, Neu et al. (2017); Abbasi-Yadkori et al. (2019); Geist et al. (2019); Zhan et al. (2023)). PMD with $D_{\mathrm{KL}}$-regularization is a policy iteration algorithm where the policy at iteration $k$ is given by a weighted average of past Q-functions, such that

$$\pi_k \propto \exp\left(\alpha \sum_{i=0}^{k} \beta^i Q^{k-i}\right), \tag{1}$$

for temperature $\alpha > 0$ and decay factor $\beta \in [0, 1]$, where $\beta$ is strictly $< 1$ if an entropy bonus is used in addition to the $D_{\mathrm{KL}}$ regularizer (see Sec. 3 for more details). The averaging over previous Q-functions induced by the $D_{\mathrm{KL}}$ regularizer is known to average out approximation errors over the true Q-functions and has stabilizing properties in practice (Geist et al., 2019; Abbasi-Yadkori et al., 2019). The policy in Eq. 1 naturally calls for a parameter isolation approach that would store all past Q-functions, leading to exact entropy-regularized policy update. Abbasi-Yadkori et al. (2019) performed an experiment of the sort on an Atari RL task (Bellemare et al., 2013), keeping in memory the past 10 Q-functions, and noted increased stability and performance over vanilla DQN.

Overall, this paper extends prior work on PMD (Abbasi-Yadkori et al., 2019; Lazic et al., 2021; Zhan et al., 2023) in several ways. i) We propose an NN architecture called Stacked NNs (SNNs) that stacks NN weights through a third dimension (representing the iteration number, see Sec. 5 for more details), allowing us to efficiently compute on a GPU several hundreds of Q-functions, approaching the exact policy formulation in Eq. 1. ii) We study theoretically the convergence of PMD-like algorithms that store up to $M$ most recent Q-functions, replacing the oldest Q-function with the newest one when $k > M$. These theoretical results extend those of (Zhan et al., 2023) that

studied PMD with an inexact policy update. In Eq. 1, when $\beta < 1$, one can clearly see that the weight of old Q-functions decays to $0$ as $k$ goes to infinity. Thus, by choosing a large enough $M$, we can control the error stemming from the use of a finite number of Q-functions. A more subtle result of our theoretical analysis is that Q-functions become increasingly similar as learning progresses. As a result, deleting an old Q-function affects less and less the policy, and we can show that there is a finite $M$—that depends on $\beta$ and the discount factor $\gamma$ of the decision task—large enough to ensure convergence to the optimal policy without residual error, although at a reduced rate than exact PMD. iii) Experiments on a large set of tasks confirm the preliminary results of Abbasi-Yadkori et al. (2019): the averaging over past Q-functions stabilizes learning and improves performance in several cases. The proposed algorithm, which we call StaQ, uses a closed-form policy update that reduces the sources of errors and increase the overall stability. Still, some instabilities may remain coming from the choice of the behavior policy (see Fig. 1) or the policy evaluation algorithm. Yet, we believe that the simplicity of StaQ's policy update, its strong theoretical foundation and its empirical behavior brings us closer to a completely stable deep RL algorithm.

## 2 RELATED WORK

**Entropy-regularization in RL.** Entropy regularization has seen widespread usage in RL. It was used with (natural) policy gradient (Kakade, 2001; Schulman et al., 2015), policy search (Deisenroth et al., 2013), policy iteration (Abbasi-Yadkori et al., 2019; Zhan et al., 2023) and value iteration methods (Fox et al., 2016; Vieillard et al., 2020). Common choices of regularizers include minimizing the $D_{\mathrm{KL}}$ between the current and previous policy (Azar et al., 2012; Schulman et al., 2015) or encouraging high Shannon entropy (Fox et al., 2016; Haarnoja et al., 2018), but other forms of entropy regularizers exist (Lee et al., 2019; Alfano et al., 2023). We refer the reader to Neu et al. (2017); Geist et al. (2019) for a broader categorization of entropy regularizers and their relation to existing deep RL methods. In this paper, we use both a $D_{\mathrm{KL}}$ penalization w.r.t. the previous policy and a Shannon entropy bonus in a policy iteration context. In Vieillard et al. (2020), both types of regularizers were used but in a value iteration context. Abbasi-Yadkori et al. (2019); Lazic et al. (2021) are policy iteration methods but only use $D_{\mathrm{KL}}$ penalization. Finally, Zhan et al. (2023) is in the same setting as our work but is mostly theoretical. In this paper, we study a family of practical deep RL algorithms that extend the analysis of the approximate algorithms in Zhan et al. (2023), and provide both a theoretical analysis and an extensive empirical evaluation of these algorithms.

**Continual Learning (CL) and RL.** In CL a learner faces non-stationary data distributions (commonly referred to as *tasks*) and aims at obtaining good performance (e.g. classification accuracy) on the newest task without hindering performance on previous tasks (De Lange et al., 2021; Wang et al., 2024). RL and CL connect in several ways: some continual supervised learning settings can be formulated as RL problems (Kumar et al., 2023), while RL can be extended to the CL setting by having non-stationary rewards or environment dynamics (Rusu et al., 2016; Khetarpal et al., 2020). Even when the rewards and dynamics are stationary—setting which we refer to as single task RL—data in RL can still arrive in a non-stationary fashion as in CL. As such, both RL and CL can end-up using surprisingly similar approaches, as discussed in the previous section. Nonetheless, not all research in CL might be relevant to single task RL and vice versa. For example, in RL, the learner can (indirectly) control the data distribution, providing additional mechanisms to reduce catastrophic forgetting which are not always possible in CL (see Sec. 6.3 for an example).

**Growing neural architectures in RL.** Saving past Q-functions was investigated in the context of policy evaluation. In Tosatto et al. (2017), a first Q-function is learned then frozen and a new network is added, learning the residual error. Anschel et al. (2017) extended DQN by saving the past 10 Q-functions, and using them to compute lower variance target values. Both Girgin & Preux (2008) and Della Vecchia et al. (2022) use a special neural architecture called the cascade-correlation network (Fahlman & Lebiere, 1989) to grow neural policies. The former work studies such policies in combination with LSPI (Lagoudakis & Parr, 2003), without entropy regularization. The latter work is closer to ours, as it uses a $D_{\mathrm{KL}}$-regularizer but does not include a deletion mechanism. As such the policy grows indefinitely, limiting the scaling of the method. Finally, Abbasi-Yadkori et al. (2019) save the past 10 Q-functions to compute the policy in Eq. 1 for the specific case of $\beta = 1$, but do not study the impact of deleting older Q-functions as we do in this paper. Growing neural architectures are more common in the neuroevolution community (Stanley & Miikkulainen, 2002). Such evolutionary approaches were already used for RL tasks but are beyond the scope of this paper.

## 3 PRELIMINARIES

Let a Markov Decision Problem (MDP) be defined by the tuple $(S, A, R, P, \gamma)$, such that $S$ and $A$ are finite state and action spaces, $R$ is a bounded reward function $R : S \times A \mapsto [-R_\mathrm{x}, R_\mathrm{x}]$ for some positive constant $R_\mathrm{x}$, $P$ defines the (Markovian) transition probabilities of the decision process and $\gamma$ is a discount factor. The algorithms presented in this paper can be extended to more general state spaces. However, the limitation to a finite $A$ is non-trivial to lift because of the sampling from softmax distributions as in Eq. 1. We discuss in Sec. 7 potential ways to address this limitation.

Let $\Delta(A)$ be the space of probability distributions over $A$, and $h$ be the negative entropy given by $h : \Delta(A) \mapsto \mathbb{R}$, $h(p) = p \cdot \log p$, where $\cdot$ is the dot product and the $\log$ is applied element-wise to the vector $p$. Let $\pi : S \mapsto \Delta(A)$ be a stationary stochastic policy mapping states to distributions over actions. We denote the entropy regularized V-function for policy $\pi$ and regularization weight $\tau > 0$ as $V_\tau^\pi : S \mapsto \mathbb{R}$, which is defined by:

$$V_\tau^\pi(s) = \mathbb{E}_\pi \left[ \sum_{t=0}^\infty \gamma^t \{ R(s_t, a_t) - \tau h(\pi(s_t)) \} \bigg| s_0 = s \right]. \tag{2}$$

In turn, the entropy regularized Q-function is given by $Q_\tau^\pi(s, a) = R(s, a) + \gamma \mathbb{E}_{s'} [V_\tau^\pi(s')]$. The V-function can be written as the expectation of the Q-function plus the current state entropy, i.e. $V_\tau^\pi(s) = \mathbb{E}_a [Q_\tau^\pi(s, a)] - \tau h(\pi(s))$ which leads to the Bellman equation:

$$Q_\tau^\pi(s, a) = R(s, a) + \gamma \mathbb{E}_{s', a'} [Q_\tau^\pi(s', a') - \tau h(\pi(s'))]. \tag{3}$$

In the following, we will write policies of the form $\pi(s) \propto \exp(Q(s, \cdot))$ for all $s \in S$ more succinctly as $\pi \propto \exp(Q)$. We define optimal V and Q functions by

$$\text{for all } s \in S, a \in A, \quad V_\tau^\star(s) := \max_\pi V_\tau^\pi(s), \quad Q_\tau^\star(s, a) := \max_\pi Q_\tau^\pi(s, a).$$

Moreover, the policy $\pi^\star \propto \exp\left(\frac{Q_\tau^\star}{\tau}\right)$ satisfies $Q_\tau^{\pi^\star} = Q_\tau^\star$ and $V_\tau^{\pi^\star} = V_\tau^\star$ simultaneously for all $s \in S$ (Zhan et al., 2023). In the following, we will overload notations of real functions defined on $S \times A$ and allow them to only take a state input and return a vector in $\mathbb{R}^{|A|}$. For example, $Q_\tau^\pi(s)$ denotes a vector for which the $i^\mathrm{th}$ entry $i \in \{1, \ldots, |A|\}$ is equal to $Q_\tau^\pi(s, i)$. Finally we define

$$\bar{R} := \frac{R_\mathrm{x} + \gamma \tau \log |A|}{1 - \gamma}, \tag{4}$$

as the finite upper-bound of $\|Q_\tau^\pi\|_\infty$ for any policy $\pi$, that can be computed by assuming the agent collects the highest reward and entropy possible at every step.

### 3.1 ENTROPY-REGULARIZED POLICY MIRROR DESCENT

To find $\pi^\star$, we focus on Entropy-regularized Policy Mirror Descent (EPMD) methods (Neu et al., 2017; Abbasi-Yadkori et al., 2019; Lazic et al., 2021) and notably on those that regularize both the policy update and the Q-function (Zhan et al., 2023; Lan, 2022). The PMD setting discussed here is also equivalent to the regularized natural policy gradient algorithm on softmax policies of Cen et al. (2022). Let $\pi_k$ be the policy at iteration $k$ of EPMD, and $Q_\tau^k := Q_\tau^{\pi_k}$ its Q-function. The next policy in EPMD is the solution of the following optimization problem:

$$\text{for all } s \in S, \pi_{k+1}(s) = \underset{p \in \Delta(A)}{\arg\max} \left\{ Q_\tau^k(s) \cdot p - \tau h(p) - \eta D_\mathrm{KL}(p; \pi_k(s)) \right\}, \tag{5}$$

$$\propto \pi_k(s)^{\frac{\eta}{\eta + \tau}} \exp\left( \frac{Q_\tau^k(s)}{\eta + \tau} \right), \tag{6}$$

where $D_\mathrm{KL}(p; p') = p \cdot (\log p - \log p')$ and $\eta > 0$ is the $D_\mathrm{KL}$ regularization weight. The closed form expression in Eq. 6 is well-known and its proof can be checked in Zhan et al. (2023) for instance. We let $\alpha = \frac{1}{\eta + \tau}$ and $\beta = \frac{\eta}{\eta + \tau}$, hereafter referred to as a step-size and a decay factor respectively. Our paper focuses on single task RL but sees the policy update in Eq. 5 through the lens of CL. Please see App. E for more details. Let $\xi_k$ be a real function of $S \times A$ for any positive integer $k$. We

assume as the initial condition that $\pi_0 \propto \exp(\xi_0)$ with $\xi_0 = 0$, i.e. $\pi_0$ is uniform over the actions. At every iteration of EPMD, the update in Eq. 6 yields the following logits update

$$\pi_{k+1} \propto \exp(\xi_{k+1}), \quad \xi_{k+1} = \beta\xi_k + \alpha Q_\tau^k. \tag{7}$$

From the recursive definition of $\xi_{k+1}$, it can easily be verified that $\xi_{k+1} = \alpha \sum_{i=0}^{k} \beta^{k-i} Q_\tau^i$. The convergence of EPMD is characterized by the following theorem

**Theorem 3.1** (Adapted from Zhan et al. (2023), Thm. 1). *At iteration $k$ of EPMD, the Q-function of $\pi_k$ satisfies* $\left\| Q_\tau^\star - Q_\tau^k \right\|_\infty \leq \gamma d^{k-1} \left( \left\| Q_\tau^\star - Q_\tau^0 \right\|_\infty + 2\beta \left\| Q_\tau^\star \right\|_\infty \right)$, *with $d = \beta + \gamma(1 - \beta) < 1$.*

The above theorem shows that by following EPMD, we have a linear convergence of $Q_\tau^k$ towards $Q_\tau^\star$, with a convergence rate of $d$. In the next section, we will be interested in an approximate version of EPMD, where the Q-function $Q_\tau^k$ is computed exactly but where $\xi_k$ is limited to summing at most $M$ Q-functions. We name this setting finite-memory EPMD. In the main paper, we only focus for clarity on error introduced in the policy update introduced by this deletion mechanism. Results for additional errors in the policy evaluation are deferred to App. A.

## 4 FINITE-MEMORY POLICY MIRROR DESCENT

Let $M > 0$ be a positive integer defining the maximum number of Q-functions we are allowed to store. As a warm-up, we first show in Sec. 4.1 a straightforward implementation of finite-memory EPMD, where we simply truncate the sum of the Q-functions in Eq. 1 to the last $M$ Q-functions. The convergence analysis of this algorithm is largely based on the analysis of the approximate EPMD of Zhan et al. (2023), which assumes a bounded sub-optimality of the policy update procedure (Eq. 5). Compared to the proof from the prior work, the main step in our setting consists in quantifying the effect of the finite-memory assumption to the policy improvement step. As in the class of approximate algorithms analyzed in Zhan et al. (2023), the algorithm in Sec. 4.1 always exhibits an irreducible error for a finite $M$. To address this issue, we introduce a weight corrected algorithm that rescales the policy in Eq. 1 to account for its finite-memory nature. This rescaling introduces long range dependencies that complicate the analysis, but can result in convergence to $Q_\tau^\star$, provided a large enough $M$.

### 4.1 VANILLA FINITE-MEMORY EPMD

Consider an approximate EPMD setting where the update to $\xi_k$ is given by

$$\xi_{k+1} = \beta\xi_k + \alpha \left( Q_\tau^k - \beta^M Q_\tau^{k-M} \right), \tag{8}$$

with $Q_\tau^{k-M} := 0$ whenever $k - M < 0$. Compared to $\xi_{k+1}$ in Eq. 7, we simultaneously add the new $Q_\tau^k$ and 'delete' an old Q-function by subtracting $Q_\tau^{k-M}$ in Eq. 8. As a result, $\xi_{k+1}$ can now be written as $\xi_{k+1} = \alpha \sum_{i=0}^{M-1} \beta^i Q_\tau^{k-i}$, and thus it is a finite-memory EPMD algorithm using at most $M$ Q-functions.

We now want to investigate if we have any convergence guarantees of $Q_\tau^k$ towards $Q_\tau^\star$ as for EPMD. Let the policy $\tilde{\pi}_k$ be defined by $\tilde{\pi}_k \propto \exp(\tilde{\xi}_k)$ with $\tilde{\xi}_k = \alpha \sum_{i=0}^{M-2} \beta^i Q_\tau^{k-1-i}$. Here, $\tilde{\xi}_k = \xi_k - \alpha\beta^{M-1}Q_\tau^{k-M}$, i.e. it is obtained by deleting the oldest Q-function from $\xi_k$ and thus is a sum of $M - 1$ Q-functions. The update in Eq. 8 can now be rewritten as $\xi_{k+1} = \beta\tilde{\xi}_k + \alpha Q_\tau^k$. From Sec. 3, we recognize this update as the result of the following optimization problem:

$$\text{for all } s \in S, \quad \pi_{k+1}(s) = \arg\max_{p \in \Delta(A)} \left\{ Q_\tau^k(s) \cdot p - \tau h(p) - \eta D_{\mathrm{KL}}(p; \tilde{\pi}_k(s)) \right\}. \tag{9}$$

In this approximate scheme, we compute the $D_{\mathrm{KL}}$ regularization w.r.t. $\tilde{\pi}_k$ instead of the previous policy $\pi_k$. This can negatively impact the quality of $\pi_{k+1}$ as it might force $\pi_{k+1}$ to stay close to the potentially bad policy $\tilde{\pi}_k$. In the following theorem, we provide a form of an approximate policy improvement of $\pi_{k+1}$ on $\pi_k$, that depends on how close $\tilde{\pi}_k$ is to $\pi_k$. This theorem applies to a generic policy $\tilde{\pi}_k$, therefore it can be of interest beyond the scope of this paper.

**Theorem 4.1** (Approximate policy improvement). *Let $\pi_k \propto \exp(\xi_k)$ be a policy with associated Q-function $Q_\tau^k$. Let $\tilde{\pi}_k \propto \exp(\tilde{\xi}_k)$ be an arbitrary policy. Let $\pi_{k+1}$ be the policy optimizing Eq. 9*

*w.r.t. the hereby defined $Q_\tau^k$ and $\tilde{\pi}_k$, then the Q-function $Q_\tau^{k+1}$ of $\pi_{k+1}$ satisfies*

$$Q_\tau^{k+1} \geq Q_\tau^k - \gamma\eta \frac{\max_{s\in S} \|(\pi_k - \tilde{\pi}_k)(s)\|_1 \left\|\xi_k - \tilde{\xi}_k\right\|_\infty}{1-\gamma}. \tag{10}$$

The proof of Thm. 4.1 and all future proofs are given in App. A. Applying Thm. 4.1 to our setting gives the following policy improvement lower bound

**Corollary 4.1.1.** *Let $\pi_k \propto \exp(\xi_k)$ be a policy with associated Q-function $Q_\tau^k$, such that $\xi_k = \alpha \sum_{i=0}^{M-1} \beta^i Q_\tau^{k-1-i}$. Let $\tilde{\pi}_k \propto \exp(\tilde{\xi}_k)$ be the policy such that $\tilde{\xi}_k = \alpha \sum_{i=0}^{M-2} \beta^i Q_\tau^{k-1-i}$. Let $\pi_{k+1}$ be the policy optimizing Eq. 9, then the Q-function $Q_\tau^{k+1}$ of $\pi_{k+1}$ satisfies*

$$Q_\tau^{k+1} \geq Q_\tau^k - \gamma\beta^M \frac{\min\left\{2, \alpha\beta^{M-1}\bar{R}\right\}\bar{R}}{1-\gamma}. \tag{11}$$

In vanilla EPMD, it is guaranteed that $Q_\tau^{k+1} \geq Q_\tau^k$ (Zhan et al., 2023). In this approximate setting, we can bring the error arbitrarily close to 0 through the term $\beta^M$ by choosing a large enough $M$, since $\beta < 1$.

Having quantified the error in the policy improvement step, we follow the general steps of the proof of approximate EPMD of Zhan et al. (2023) and come to the following convergence guarantees.

**Theorem 4.2** (Convergence of vanilla finite-memory EPMD). *After $k \geq 0$ iterations of Eq. 8, we have that $\left\|Q_\tau^\star - Q_\tau^k\right\|_\infty \leq \gamma d^k \left(\|Q_\tau^\star\|_\infty + \|Q_\tau^0\|_\infty\right) + \gamma\beta^k \|Q_\tau^0\|_\infty + \beta^M C_1$, with $d = \beta + \gamma(1 - \beta) < 1$ and $C_1 = \frac{\gamma\bar{R}}{1-\gamma} \left[\frac{\gamma \min\{2, \alpha\beta^{M-1}\bar{R}\}}{(1-\beta)(1-\gamma)} + 2\right]$.*

Since $\beta < d < 1$, the slowest term vanishes at a rate of $d$ as with exact EPMD. However, we eventually reach an error of $\beta^M C_1$, that does not decrease as $k$ increases, and that we can only control by increasing the memory size $M$. A problem with the current algorithm is that even if all past Q-functions are equal to $Q_\tau^\star$, then $\tau\xi_k = (1-\beta) \sum_{i=0}^{M-1} \beta^i Q_\tau^\star = (1-\beta^M)Q_\tau^\star$, whereas we know that asymptotically $\xi_k$ should converge to the logits of $\pi^\star$ (Sec. 3) which are $\frac{Q_\tau^\star}{\tau}$. This suggests a slightly modified algorithm that rescales $\xi_k$ by $1 - \beta^M$, which we analyze in the next section.

## 4.2 Weight corrected finite-memory EPMD

Consider now the alternative update to $\xi_k$ given by

$$\xi_{k+1} = \beta\xi_k + \alpha Q_\tau^k + \frac{\alpha\beta^M}{1-\beta^M}(Q_\tau^k - Q_\tau^{k-M}), \tag{12}$$

where $Q_\tau^{k-M} := 0$ whenever $k - M < 0$. In contrast to the vanilla algorithm in Sec. 4.1, we now delete the oldest Q-function in $\xi_k$ and also slightly overweight the most recent Q-function to ensure that the Q-function weights sum to 1. Indeed, assuming that $\xi_0 := 0$, we can show (see App. A.4.1 for a proof) for all $k \geq 0$ that the logits only use the past $M$ Q-functions and are given by

$$\xi_{k+1} = \frac{\alpha}{1-\beta^M} \sum_{i=0}^{M-1} \beta^i Q_\tau^{k-i}. \tag{13}$$

Similar to the previous section, we introduce a policy $\tilde{\pi}_k \propto \exp(\tilde{\xi}_k)$ with $\tilde{\xi}_k = \xi_k + \frac{\alpha\beta^{M-1}}{1-\beta^M}(Q_\tau^k - Q_\tau^{k-M})$ such that logits of $\pi_{k+1}$ are given by $\xi_{k+1} = \beta\tilde{\xi}_k + \alpha Q_\tau^k$. This form of $\xi_{k+1}$ implies that $\pi_{k+1}$ satisfies the policy update in Eq. 9, and thus Thm. 4.1 applies and we have

**Corollary 4.1.2.** *Let $\pi_k \propto \exp(\xi_k)$ be a policy with associated Q-function $Q_\tau^k$, such that $\xi_k = \frac{\alpha}{1-\beta^M} \sum_{i=0}^{M-1} \beta^i Q_\tau^{k-1-i}$. Let $\tilde{\pi}_k \propto \exp(\tilde{\xi}_k)$ be the policy such that $\tilde{\xi}_k = \xi_k + \frac{\alpha\beta^{M-1}}{1-\beta^M}(Q_\tau^k - Q_\tau^{k-M})$. Let $\pi_{k+1}$ be the policy optimizing Eq. 9 with the hereby defined $Q_\tau^k$ and $\tilde{\pi}_k$, then the Q-function $Q_\tau^{k+1}$ of $\pi_{k+1}$ satisfies*

$$Q_\tau^{k+1} \geq Q_\tau^k - 2\gamma\beta^M \frac{\left\|Q_\tau^k - Q_\tau^{k-M}\right\|_\infty}{(1-\gamma)(1-\beta^M)}. \tag{14}$$

Compared to the approximate policy improvement of Sec. 4.1, we see that the lower-bound in Eq. 14 depends on $\left\|Q_\tau^k - Q_\tau^{k-M}\right\|_\infty$ instead of just $\left\|Q_\tau^{k-M}\right\|_\infty$. Thus, we can expect that as the Q-functions converge to $Q_\tau^\star$, we get tighter and tighter guarantees on the policy improvement step, which in turn guarantees convergence to $Q_\tau^\star$ without the residual error of Sec. 4.1. The next two results show that indeed, for $M$ large enough, the finite-memory EPMD scheme defined by Eq. 12 leads to convergence to $Q_\tau^\star$. Lem. 4.3 provides an upper bounding sequence for $\left\|Q_\tau^\star - Q_\tau^k\right\|_\infty$.

**Lemma 4.3.** *Let $x_{k+1} = d_1 x_k + d_2 x_{k-M}$ be a sequence such that $\forall k < 0$, $x_k = \frac{\|Q_\tau^\star\|_\infty}{\gamma}$, $x_0 = \|Q_\tau^\star\|_\infty + \|Q_\tau^0\|_\infty$, $d_1 := \beta + \gamma \frac{1-\beta}{1-\beta^M} + \gamma c_2$, $d_2 := \frac{2c_1\gamma^2}{1-\gamma}$, $c_1 := \frac{\beta^M}{1-\beta^M}$, and $c_2 := \left(\frac{1+\gamma}{1-\gamma} - \beta\right) c_1$.*
*After $k \geq 0$ iterations of Eq. 12, we have that $\left\|Q_\tau^\star - Q_\tau^k\right\|_\infty \leq x_k$.*

Then, we compute values of $M$ for which the sequence $x_k$ converges to 0 and characterize the convergence rate of $\left\|Q_\tau^\star - Q_\tau^k\right\|_\infty$ through Thm. 4.4.

**Theorem 4.4** (Convergence of weight corrected finite-memory EPMD)**.** *With the definitions of Lemma 4.3, if $M > \log \frac{(1-\gamma)^2(1-\beta)}{\gamma^2(3+\beta)+1-\beta}(\log \beta)^{-1}$ then $\lim_{k \to \infty} x_k = 0$. Moreover, $\forall k \geq 0$,*
$$\left\|Q_\tau^\star - Q_\tau^k\right\|_\infty \leq (d_1 + d_2 d_3^{-1})^k \max\left\{\frac{\|Q_\tau^\star\|_\infty}{\gamma}, \|Q_\tau^\star\|_\infty + \|Q_\tau^0\|_\infty\right\}, \text{ where } d_3 := \left(d_1^M + d_2 \frac{1-d_1^M}{1-d_1}\right)$$
*and $\lim_{M \to \infty} d_1 + d_2 d_3^{-1} = \beta + \gamma(1-\beta)$.*

Thm. 4.4 defines a minimum memory size that guarantees convergence to $Q_\tau^\star$. This minimum $M$ depends only on $\beta$ and $\gamma$, and is usually within the range of practical values: for example, with $\gamma = 0.99$ and $\beta = 0.95$, the minimum $M$ suggested by Thm. 4.4 is 265, which is reasonable in terms of memory and computation with current GPUs (we used $M = 300$ in all our experiments). As can be expected, these values of $M$ are generally pessimistic and in practice we did not observe better performance when using as large $M$ as suggested by Thm. 4.4.

In terms of convergence rate, $d_1 + d_2 d_3^{-1}$ given in Thm 4.4 tends to $d$—the convergence rate of exact EPMD—as $M$ goes to infinity. Thus, it is slower than exact EPMD, and slower than the algorithm in Sec. 4.1, but unlike the latter it does not have an irreducible error and converges to $Q_\tau^\star$.

# 5 STACKED NEURAL NETWORKS

To study finite-memory EPMD in the memory regimes suggested by Thm. A.6, we propose the Stacked NN architecture (SNN, illustrated in Fig. 1) that makes efficient use of GPUs to compute multiple Q-values in parallel, and we call the resulting algorithm StaQ. An SNN is parameterized by $M$, the maximum number of networks kept in memory. The main operation of an SNN is a `push` function, that "stacks" an input NN into an SNN. The stacking consists in building tensors with one extra dimension for every weight matrix of the NN Q-function. For instance, let $A_k$ be a $256 \times 256$ weight matrix of a hidden layer of (the neural approximation of) $Q_\tau^k$. Let $\mathbf{A}_k$ be a $k \times 256 \times 256$ tensor of the SNN that stores all past $A_i$ weight matrices for $i \leq k$. In the forward pass of the SNN, to compute past Q-functions $Q_\tau^i$ for $i \leq k$, we use the batch matrix operation of PyTorch (Paszke et al., 2017) on $\mathbf{A}_k$ to make full use of the GPU parallel computing power. Finally, if the SNN contains more than $M$ NNs after a `push` operation, the oldest NN is deleted from the SNN in a "first in first out" fashion. This `push` operation of SNNs currently supports multi-layer Perceptron NN architectures only. We leave the extension to other neural architectures such as CNNs (LeCun et al., 2015) or LSTMs (Hochreiter & Schmidhuber, 1997) to future work.

To further reduce the impact of a large $M$, we pre-compute $\xi_k$ for all entries in the replay buffer[1] at the start of policy evaluation. The logits $\xi_k$ are used to sample on-policy actions when computing the targets for $Q_\tau^k$. As a result of the pre-computation, during policy evaluation, forward and backward passes only operate on the current Q-function and hence the impact of large $M$ is minimized. Beyond policy evaluation, computing $\xi_k$ is also used in data collection by the behavior policy $\pi_k^b$ that typically relies on computing $\pi_k$ (see Sec. 6). Conversely, the policy update consists only of the above `push` operation, and thus, is optimization free and (almost) instantaneous. Table 1 shows the training time of StaQ over 5 million steps as a function of $M$ for two environments. Varying $M$ (up to 500) or the size of the state space has little impact on the runtime of StaQ when running on GPU.

---

[1]Since we use small replay buffer sizes of 50K transitions, we are likely to process each transition multiple times (25.6 times in expectation in our experiments) making this optimization worthwhile.

| | Memory size $M$ | 1 | 50 | 100 | 300 | 500 |
|---|---|---|---|---|---|---|
| Hopper-v4 | Training time (hrs) | 9.8 | 10.1 | 10.3 | 10.3 | 10.9 |
| Ant-v4 | Training time (hrs) | 10.4 | 10.7 | 10.3 | 11 | 10.5 |

Table 1: Training times for StaQ (5 million steps), as a function of $M$, on Hopper-v4 (state dim.=11) and Ant-v4 (state dim. = 105), computed on an NVIDIA Tesla V100 and averaged over 3 seeds.

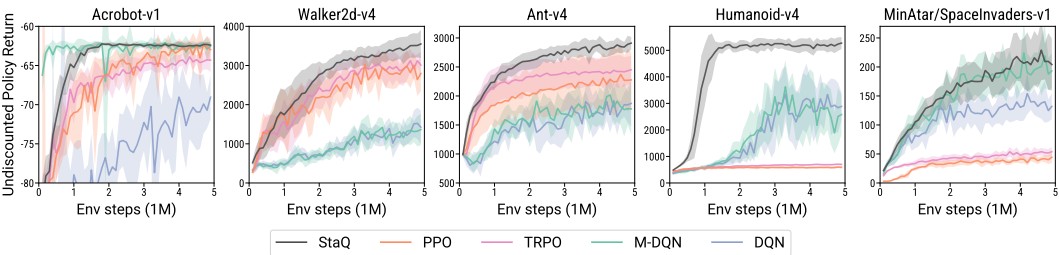

Figure 2: Policy return of StaQ, compared with deep RL baselines. Plots show mean and one standard deviation computed over 10 seeds. StaQ matches or exceeds state-of-the-art on several tasks and has generally lower variance. Even when the variance appears to be of the same order than the deep RL baselines, the oscillations within runs are generally reduced, see e.g. Fig. 3. See App. B.1 for all results.

## 6 EXPERIMENTS

In this section, we assess the empirical merits of StaQ, paying attention to both the performance and stability. Note that, in the context of Continual Learning, we can relate this to the stability-plasticity trade-off, since plasticity in this single-task setting is measured by the policy return. Regarding stability, we check if the averaging over the large number of Q-functions (several hundred) done by SNN improves the stability of the learning process, in terms of reducing performance oscillations across iterations of the same seed. Naturally, we also investigate the hyper-parameter $M$ to determine values for which the above stability might manifest. Finally, we discuss some of the limitations of our algorithm and how they open new perspectives towards a fully reliable deep RL solver.

**Environments.** We use all 9 environments suggested by Ceron & Castro (2021) for comparing deep RL algorithms with finite action spaces, comprising 4 classic control tasks from Gymnasium (Towers et al., 2023) and all MinAtar tasks (Young & Tian, 2019). To that we add 5 Mujoco tasks (Todorov et al., 2012), adapted to discrete action spaces by considering only extreme actions similarly to (Seyde et al., 2021). To illustrate, the discrete version of a Mujoco task with action space $A = [-1, 1]^d$ consists in several $d$ dimensional vectors that have zeroes everywhere except at entry $i \in \{1, \ldots, d\}$ that can either take a value of 1 or $-1$; to that we add the zero action, leading to a total of $2d + 1$ actions.

**Baselines.** We compare StaQ against the value iteration algorithm DQN (Mnih et al., 2015) and its entropy-regularized variant M-DQN (Vieillard et al., 2020), the policy gradient algorithm TRPO (Schulman et al., 2015) as it uses a $D_{\mathrm{KL}}$ regularizer and PPO (Schulman et al., 2017)[2]. SAC (Haarnoja et al., 2018) is another popular deep RL baseline that uses entropy regularization but is not adapted for discrete action environments, as discussed in App. D. Comparisons with baselines are averaged over 10 seeds and show the mean and standard deviation of the return. The return is computed by evaluating every 100K steps the current deterministic policy by averaging 50 rollouts. Hyperparameters for StaQ and the baselines are provided in App. F.

**StaQ's components.** We consider the behavior policy $\pi_k^b$ that is an $\epsilon$-*softmax* policy over $\pi_k$ — by analogy with $\epsilon$-greedy policies, which mixes the softmax policy $\pi_k$ and a uniform policy with probabilities $(1 - \epsilon)$ and $\epsilon$ respectively. We use the $\epsilon$-softmax policy instead of $\pi_k$ as we found that using only $\pi_k$ can cause catastrophic forgetting in the Q-function, as discussed in Sec. 6.3. For

---

[2]For TRPO and PPO, we use the implementation provided in `stable-baselines3` (Raffin et al., 2021), while we implemented our own PyTorch version of (M)-DQN.

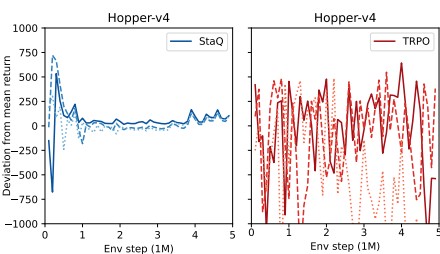
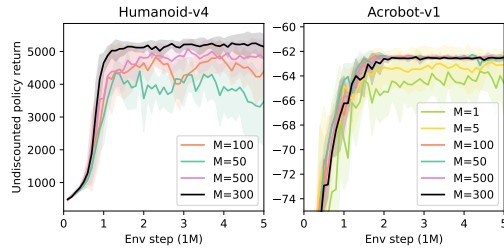

(a) Return of individual runs, centered at each step.

(b) Impact of memory size M.

Figure 3: **Left:** Deviation from mean policy returns for individual runs on Hopper, comparing StaQ and the closest performing baseline TRPO. Returns are centered at every timestep by subtracting the mean across 10 seeds. In general, individual runs of StaQ have significantly lower variance across timesteps compared to TRPO. For clarity, we only plot the first three seeds. See App B.2 for further environments and algorithms. **Right:** Policy returns under different choice of $M$. On the simpler Acrobot task, $M > 5$ seems sufficient but on Humanoid, even $M = 100$ seems insufficient. Plots showing mean and one standard deviation computed over 5 seeds.

learning $Q_\tau^k$, we used a similar approach to SAC (Haarnoja et al., 2018), sampling an action from the current policy and use an ensemble of two target Q-functions updated in a hard manner. At the start, the NN for $Q_\tau^0$ is initialized with the zero output, so that $\pi_0$ is a uniform policy, for all consecutive iterations the NN for $Q_\tau^k$ is initialized at the computed $Q_\tau^{k-1}$ (to make the transfer from $Q_\tau^k$ to $Q_\tau^{k-1}$ smoother). When computing the Q-values in the next states, we aggregate the target values by taking either the `min` or the `mean` of the two Q-functions. We discuss the impact of this choice in App. B.5. For the full set of hyperparameters consult App. F. To account for the varying action dimension $|A|$ of the environments, we set the *scaled entropy coefficient* $\bar\tau$ as a hyperparameter, defined by $\bar\tau = \tau \log |A|$, rather than directly setting $\tau$. Furthermore, the entropy weight is linearly annealed from its minimum and maximum values over the first 1 million timesteps.

## 6.1 STABILITY AND PERFORMANCE OF STAQ

A comparison of StaQ to deep RL baselines is shown for a selection of environments in Fig. 2, and for all environments in App. B.1. In general, the performance matches or exceeds previous state-of-the-art algorithms, with lower variance and less performance oscillation. The improved stability of StaQ is more clearly seen when we look at the variability within individual runs. An example of this is shown in Fig. 3 (**Left**) for Hopper, where we plot the return for each seed, centered by subtracting the mean return across all seeds at each evaluation timestep. More comparisons are provided in App. B.2. These experiments confirm the preliminary results of Abbasi-Yadkori et al. (2019) that a policy averaging over multiple Q-functions stabilizes learning. While prior work considered only saving the last 10 Q-functions, we show next that, on more complex tasks, saving an order of magnitude more Q-functions can still have positive effects on stability and performance.

## 6.2 THE IMPACT OF THE MEMORY-SIZE $M$

According to Sec. 4.2, $M$ is a crucial parameter that should be large enough to guarantee the convergence. The $M$ estimation obtained from Thm. A.6 may be very conservative in practice. In Fig. 3 (**Right**), we present the results for different choices of $M$ for Acrobot and Humanoid. Despite that some low $M \leq 100$ ($M \leq 10$ for Acrobot) can still give us decent average performance, stability is negatively affected, which is especially pronounced for more challenging environment such as Humanoid. Conversely, higher $M = 500$, while being more expensive to compute, does not lead to any improvement either in terms of performance or stability. Thus, $M = 300$ is the best compromise between stability and compute time. More environments are shown in App B.2.

## 6.3 CATASTROPHIC FORGETTING IN THE Q-FUNCTION

As illustrated in Fig. 4, when the behavior policy $\pi_k^b$ is the current soft-max policy $\pi_k$, there are more performance drops than when using an $\epsilon$-softmax policy introduced earlier. To understand this behavior, we have recorded the Q-values across 100 states for two variants of StaQ that only differ in whether or not the behavior policy adds a probability $\epsilon = 0.05$ of picking actions uniformly at random. The results are shown in App. B.4. On several states, the Q-values show larger variability when $\epsilon = 0.0$ than when $\epsilon = 0.05$, which itself creates instability in the policy as discussed in App. B.4. While $\epsilon = 0.05$ improves stability, the plot in the appendix still shows occurrences of large and sudden changes in the Q-values, which can still negatively impact performance. Moreover, $\epsilon$-softmax policies, similarly to $\epsilon$-greedy ones, might prevent deep explo-ration (Osband et al., 2016) and their off-policy nature might complicate learning (Kumar et al., 2020). In this paper, we focus on the instability of the policy; we believe that proper continual learning treatment of the Q-function, as discussed in Sec. 7, could potentially fix all remaining instabilities in deep RL.

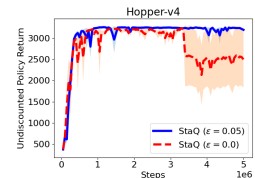

Figure 4: Policy return of StaQ for two different $\epsilon$-softmax behavior policies, with $\epsilon = 0.05$ or $\epsilon = 0$. Plot shows mean and one standard deviation over 5 seeds.

### 6.4 Entropy regularization does not solve exploration in RL

StaQ is generally competitive with the state-of-the-art, but it might fail on some environments such as MountainCar or a few MinAtar environments. These environments are considered as hard explo-ration ones by Ceron & Castro (2021), and the combination of a deterministic policy with $\epsilon$-greedy exploration seems to fare better in some tasks than the softmax policies of StaQ, PPO and TRPO, as also discussed by Vieillard et al. (2020), App. B.2. This is especially evident in MountainCar, where the initial maximum-entropy policy that picks an action uniformly at random always observes the same reward and remains close to the uniform policy throughout the learning process. To address this challenge, we have experimented with $\epsilon$-greedy behavior policies in MountainCar as discussed in App. B.5, which improves performance. Although we used a fixed entropy weight for all experi-ments, we found that on the harder MinAtar environments, tuning the values of the entropy weight can improve performance, as presented in App. B.5. Overall, we believe that exploration is an or-thogonal problem to the stability of the policy update we address in this paper, and while entropy regularization is often used as an exploration heuristic, we only use its stabilizing properties here, and leave the development of theoretically sound exploration strategies to future work.

## 7 Discussion and future work

In this paper, we have proposed to stabilize the policy update of approximate policy iteration by averaging over a very large number $M$ of past Q-functions. Surprisingly, even when $M$ is large, the final computational burden is small on modern hardware mostly thanks to the proposed SNN archi-tecture. The resulting policy update has a solid theoretical foundation and clear empirical benefits as it greatly reduces learning instability on several tasks. Yet, while the policy update is more stable, instability in learning the Q-function remains and would benefit from a more thorough continual learning treatment. To prevent the catastrophic forgetting of low probability actions, one promising idea would be to better manage the replay buffer to keep representative samples, as done in CL by Aljundi et al. (2019) for example. Another interesting line of research from the CL community is the dual architecture based on the complementary learning systems theory (e.g. Lee et al. (2016)). Adapted to RL, the Q-function would be split between a fast learning—but potentially unstable— component that adapts to the most recent samples and a slower learning component that is stabilized using, for example, growing neural architectures as in this work to avoid catastrophic forgetting. If the neural architecture of the Q-function is much larger than the one used in this paper, it might be relevant to compress periodically the policy during learning as done in CL (Schwarz et al., 2018). Finally, extending StaQ to continuous action domains could be done as in SAC (Haarnoja et al., 2018), using an extra actor network learned by minimizing the $D_{\mathrm{KL}}$ to a soft policy. This will lose the optimization-free and exact nature of the policy update but may still result in improved stabil-ity if we replace the soft policy $\exp(Q_\tau^k)$ used by SAC with $\exp(\xi_k)$ which stabilizes the target by averaging over a large number of past Q-functions.

## REPRODUCIBILITY STATEMENT

We provide the code of StaQ with the examples of launching files in the supplementary material. In addition, we carefully report the hyperparameters used for all runs of StaQ and baseline algorithms, see App. F. The code for reproducing the baselines results is withheld for anonymity reasons, as it uses a publicly available in-house library.

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

# A    PROOFS

This section includes proofs of the lemmas and theorems of the main paper.

## A.1    PROPERTIES OF ENTROPY REGULARIZED BELLMAN OPERATORS

We first start with a reminder of some basic properties of the (entropy regularized) Bellman operators, as presented in (Geist et al., 2019). Within the MDP setting defined in Sec. 3, let $T_\tau^\pi$ be the operator defined for any map $f : S \times A \mapsto \mathbb{R}$ by

$$(T_\tau^\pi f)(s,a) = R(s,a) + \gamma \mathbb{E}_{s',a'}[f(s',a') - \tau h(\pi(s'))], \qquad (15)$$

For this operator we will need the three following properties.

**Proposition A.1** (Contraction). *$T_\tau^\pi$ is a $\gamma$-contraction w.r.t. the $\|.\|_\infty$ norm, i.e. $\|T_\tau^\pi f - T_\tau^\pi g\|_\infty \leq \gamma \|f - g\|_\infty$ for any real functions $f$ and $g$ of $S \times A$.*

**Proposition A.2** (Fixed point). *$Q_\tau^\pi$ is the unique fixed point of the operator $T_\tau^\pi$, i.e. $T_\tau^\pi Q_\tau^\pi = Q_\tau^\pi$.*

Let $f, g$ be two real functions of $S \times A$. We say that $f \geq g$ iff $f(s, a) \geq g(s, a)$ for all $(s, a) \in S \times A$.

**Proposition A.3** (Monotonicity). *$T_\tau^\pi$ is monotonous, i.e. if $f \geq g$ then $T_\tau^\pi f \geq T_\tau^\pi g$.*

Let the Bellman optimality $T_\tau^\star$ operator be defined by

$$(T_\tau^\star f)(s, a) = R(s, a) + \gamma \mathbb{E}_{s'} \left[ \max_{p \in \Delta(A)} f(s') \cdot p - \tau h(p) \right]. \tag{16}$$

For the Bellman optimality operator we need the following two properties.

**Proposition A.4** (Contraction). *$T_\tau^\star$ is a $\gamma$-contraction w.r.t. the $\|.\|_\infty$ norm, i.e. $\|T_\tau^\star f - T_\tau^\star g\|_\infty \leq \gamma \|f - g\|_\infty$ for any real functions $f$ and $g$ of $S \times A$.*

**Proposition A.5** (Optimal fixed point). *$T_\tau^\star$ admits $Q_\tau^\star$ as a unique fixed point, satisfying $T_\tau^\star Q_\tau^\star = Q_\tau^\star$.*

Finally, we will make use of the well known property that the softmax distribution is entropy maximizing (Geist et al., 2019). Specifically, we know that the policy $\pi_k$ as defined in Eq. 7 satisfies the following property

$$\text{for all } s \in S, \quad \pi_k(s) = \arg\max_{p \in \Delta(A)} \xi_k(s) \cdot p - h(p), \tag{17}$$

## A.2 Proof of Theorem 4.1

We present in this appendix proofs for a more general setting where the Q-functions are inexact. Results with exact policy evaluation of the main paper can be recovered by simply setting the policy evaluation error $\epsilon_{eval}$, as defined below, to zero and by replacing $\tilde{Q}_\tau$ by $Q_\tau$.

**Assumption A.1.** *We assume that we can only compute $Q_\tau^k$ approximately, which is a Q-value function of $\pi_k$. We use $\tilde{Q}_\tau^k$ to denote the approximate $Q_\tau^k$ and we assume that there exists $\epsilon_{eval} < \infty$ such that the following holds for any $k$*

$$\left\| Q_\tau^k - \tilde{Q}_\tau^k \right\|_\infty \leq \epsilon_{eval}. \tag{18}$$

Note that Eq. 18 implies that for any $s, a$,

$$|Q_\tau^k(s, a) - \tilde{Q}_\tau^k(s, a)| \leq \epsilon_{eval} \tag{19}$$

or equivalently

$$-\epsilon_{eval} \leq Q_\tau^k(s, a) - \tilde{Q}_\tau^k(s, a) \leq \epsilon_{eval}. \tag{20}$$

As the exact $Q_\tau^k$ is no longer available, the policy update is done in the inexact policy evaluation with $\tilde{Q}_\tau^k$:

$$\text{for all } s \in S, \quad \pi_{k+1}(s) = \arg\max_{p \in \Delta(A)} \left\{ \tilde{Q}_\tau^k(s) \cdot p - \tau h(p) - \eta D_{\mathrm{KL}}(p; \tilde{\pi}_k(s)) \right\}. \tag{21}$$

We restate below the approximate policy improvement theorem in its more general form, and Theorem 4.1 can be recovered for $\epsilon_{eval} = 0$.

**Theorem A.1** (Approximate policy improvement with inexact policy evaluation). *Let $\pi_k \propto \exp(\xi_k)$ be a policy with associated evaluated Q-function $\tilde{Q}_\tau^k$. Let $\tilde{\pi}_k \propto \exp(\tilde{\xi}_k)$ be an arbitrary policy. Let $\pi_{k+1}$ be the policy optimizing Eq. 21 w.r.t. the hereby defined $\tilde{Q}_\tau^k$ and $\tilde{\pi}_k$, then the Q-function $Q_\tau^{k+1}$ of $\pi_{k+1}$ satisfies*

$$Q_\tau^{k+1} \geq \tilde{Q}_\tau^k - \gamma \eta \frac{\max_{s \in S} \|(\pi_k - \tilde{\pi}_k)(s)\|_1 \left\| \xi_k - \tilde{\xi}_k \right\|_\infty}{1 - \gamma} - \frac{\epsilon_{eval}}{1 - \gamma}. \tag{22}$$

*Proof.* Let $\pi_k \propto \exp(\xi_k)$ and let $\tilde{\pi}_k \propto \exp(\tilde{\xi}_k)$ with $X := \xi_k - \tilde{\xi}_k$. Define $\pi_{k+1}$ as in Eq. 21. From Sec. 3, we have that $\pi_{k+1} \propto \exp(\xi_{k+1})$ with the change that now an approximate $\tilde{Q}_\tau^k$ is used in the update:

$$\xi_{k+1} = \beta \tilde{\xi}_k + \alpha \tilde{Q}_\tau^k. \tag{23}$$

From the optimality of $\pi_{k+1}$ w.r.t. the policy update optimization problem in Eq. 21 we have

$$\tilde{Q}_\tau^k(s) \cdot \pi_k(s) - \tau h(\pi_k(s)) \leq \tilde{Q}_\tau^k(s) \cdot \pi_{k+1}(s) - \tau h(\pi_{k+1}(s))$$
$$- \eta D_{\text{KL}}(\pi_{k+1}(s); \tilde{\pi}_k(s)) + \eta D_{\text{KL}}(\pi_k(s); \tilde{\pi}_k(s)), \tag{24}$$

$$\leq \tilde{Q}_\tau^k(s) \cdot \pi_{k+1}(s) - \tau h(\pi_{k+1}(s)) + \eta D_{\text{KL}}(\pi_k(s); \tilde{\pi}_k(s)), \tag{25}$$

where the last inequality is due to the non-negativity of the $D_{\text{KL}}$. Let us try now to upper bound $D_{\text{KL}}(\pi_k(s); \tilde{\pi}_k(s))$ for any $s \in S$. For clarity, we will drop $s$ from the notations, and only write for e.g. $\xi(a)$ instead of $\xi_k(s, a)$. We define $Z = \sum_a \exp(\xi(a))$ and $\tilde{Z} = \sum_a \exp(\tilde{\xi}(a))$, where the sums are over all $a \in A$.

$$D_{\text{KL}}(\pi_k; \tilde{\pi}_k) = \mathbb{E}_\pi[\log \pi(a) - \log \tilde{\pi}(a)], \tag{26}$$

$$= \mathbb{E}_\pi \left[ \log \frac{\exp \xi(a)}{Z} - \log \frac{\exp \tilde{\xi}(a)}{\tilde{Z}} \right], \tag{27}$$

$$= \mathbb{E}_\pi \left[ \xi(a) - \tilde{\xi}(a) - \log \frac{Z}{\tilde{Z}} \right], \tag{28}$$

$$= \mathbb{E}_\pi \left[ X(a) - \log \frac{\sum_{a'} \exp(X(a')) \exp(\tilde{\xi}(a'))}{\sum_{a'} \exp(\tilde{\xi}(a'))} \right], \tag{29}$$

$$\overset{(i)}{\leq} \mathbb{E}_\pi \left[ X(a) - \mathbb{E}_{\tilde{\pi}} \left[ X(a') \right] \right], \tag{30}$$

$$= (\pi - \tilde{\pi}) \cdot X, \tag{31}$$

where (i) is due to Jensen's inequality. Replacing Eq. 31 into Eq. 25 yields

$$\tilde{Q}_\tau^k(s) \cdot \pi_k(s) - \tau h(\pi_k(s)) \leq \tilde{Q}_\tau^k(s) \cdot \pi_{k+1}(s) - \tau h(\pi_{k+1}(s)) + \eta(\pi - \tilde{\pi})(s) \cdot X(s). \tag{32}$$

For any $s \in S$, we have that

$$\eta(\pi - \tilde{\pi})(s) \cdot X(s) \leq \eta \max_{s \in S} |(\pi - \tilde{\pi})(s) \cdot X(s)|, \tag{33}$$

$$\overset{(i)}{\leq} \eta \max_{s \in S} \|(\pi - \tilde{\pi})(s)\|_1 \|X(s)\|_\infty, \tag{34}$$

$$= \eta \max_{s \in S} \|(\pi - \tilde{\pi})(s)\|_1 \|X\|_\infty, \tag{35}$$

$$:= \epsilon, \tag{36}$$

where we applied Hölder's inequality in (i). Combining Eq. 36 with Eq. 32 and using the definition of the operator $T_\tau^\pi$ as in Eq. 15 yields for any $s \in S$ and $a \in A$

$$R(s, a) + \gamma \mathbb{E}_{s'} \left[ \tilde{Q}_\tau^k(s') \cdot \pi_k - \tau h(\pi_k(s')) \right] \leq R(s, a) + \gamma \mathbb{E}_{s'}[\tilde{Q}_\tau^k(s') \cdot \pi_{k+1}(s')$$
$$- \tau h(\pi_{k+1}(s')) + \epsilon], \tag{37}$$

$$\implies (T_\tau^k \tilde{Q}_\tau^k)(s, a) \leq (T_\tau^{k+1} \tilde{Q}_\tau^k)(s, a) + \gamma \epsilon, \tag{38}$$

where $\epsilon = \eta \max_{s \in S} \|(\pi - \tilde{\pi})(s)\|_1 \|X\|_\infty$. Since Eq. 38 is valid for any $s$ and $a$, then

$$T_\tau^k \tilde{Q}_\tau^k \leq T_\tau^{k+1} \tilde{Q}_\tau^k + \gamma \epsilon, \tag{39}$$

Let us have a closer look at $T_\tau^k \tilde{Q}_\tau^k$, if we use Eq. 18 and by the fixed point property of Prop. A.2 we have

$$T_\tau^k \tilde{Q}_\tau^k = T_\tau^k Q_\tau^k + \gamma \mathbb{E}_{s'} \left[ (\tilde{Q}_\tau^k - Q_\tau^k) \cdot \pi_k \right] \geq Q_\tau^k - \gamma \epsilon_{eval}. \tag{40}$$

This implies

$$Q_\tau^k \le T_\tau^{k+1}\tilde{Q}_\tau^k + \gamma\epsilon + \gamma\epsilon_{eval}. \tag{41}$$

The addition of $\gamma(\epsilon + \epsilon_{eval})$ in the above expression is performed element-wise for all states and actions. Using the monotonicity property of Prop. A.3 on Eq. 41, we have

$$T_\tau^{k+1}Q_\tau^k \le T_\tau^{k+1}\left(T_\tau^{k+1}\tilde{Q}_\tau^k + \gamma(\epsilon + \epsilon_{eval})\right), \tag{42}$$

$$\le (T_\tau^{k+1})^2\tilde{Q}_\tau^k + \gamma^2(\epsilon + \epsilon_{eval}), \tag{43}$$

$$\implies Q_\tau^k \le (T_\tau^{k+1})^2\tilde{Q}_\tau^k + \gamma(\epsilon + \epsilon_{eval}) + \gamma^2(\epsilon + \epsilon_{eval}). \tag{44}$$

By repeating the same process one can easily show by induction that

$$Q_\tau^k \le (T_\tau^{k+1})^n\tilde{Q}_\tau^k + \sum_{i=1}^{n}\gamma^i(\epsilon + \epsilon_{eval}). \tag{45}$$

Taking the limit of Eq. 45 for $n \to \infty$ yields by the uniqueness of the fixed point of $T_\tau^{k+1}$

$$Q_\tau^k \le Q_\tau^{k+1} + \frac{\gamma(\epsilon + \epsilon_{eval})}{1 - \gamma}, \tag{46}$$

Finally,

$$\tilde{Q}_\tau^k \le Q_\tau^{k+1} + \frac{\gamma\epsilon}{1 - \gamma} + \frac{\epsilon_{eval}}{1 - \gamma} \tag{47}$$

$$= Q_\tau^{k+1} + \gamma\eta\frac{\max_{s \in S}\|(\pi - \tilde{\pi})(s)\|_1 \left\|\xi_k - \tilde{\xi}_k\right\|_\infty}{1 - \gamma} + \frac{\epsilon_{eval}}{1 - \gamma}. \tag{48}$$

$$\square$$

### A.3 APPROXIMATE FINITE-MEMORY EPMD

#### A.3.1 PROOF OF COROLLARY 4.1.1

Cor. 4.1.1 is a direct application of Thm. 4.1 with the specific values for $\xi_k$ and $\tilde{\xi}_k$ of finite-memory EPMD as defined in Sec. 4.1.

*Proof.* To prove Cor. 4.1.1, we will bound the two terms $\eta\left\|\xi_k - \tilde{\xi}_k\right\|_\infty$ and $\max_{s \in S}\|(\pi - \tilde{\pi})(s)\|_1$ individually, using the fact that

$$\xi_k - \tilde{\xi}_k = \alpha\beta^{M-1}\tilde{Q}_\tau^{k-M}. \tag{49}$$

Let us first start with the term

$$\eta\left\|\xi_k - \tilde{\xi}_k\right\|_\infty \overset{(i)}{=} \beta^M\left\|\tilde{Q}_\tau^{k-M}\right\|_\infty \tag{50}$$

$$\le \beta^M\bar{R}. \tag{51}$$

In $(i)$ we used the fact that $\eta\alpha = \beta$, whereas the second inequality comes from the bounded nature of $Q_\tau^\pi$ for any $\pi$, where $\bar{R}$ is defined in Eq. 4.

For $\max_{s \in S}\|(\pi - \tilde{\pi})(s)\|_1$, we can either upper-bound it by 2, or use the fact that $\pi$ and $\tilde{\pi}$ are close given large enough $M$. First, note that the gradient of the negative entropy is given by

$$\nabla h(p) = \nabla(p \cdot \log p), \tag{52}$$

$$= \log p + 1. \tag{53}$$

As the negative entropy is 1-strongly convex w.r.t. the $\|.\|_1$ norm (a.k.a. Pinsker's inequality), we have for all $s \in S$, where the $s$ dependency is dropped

$$\|\pi_k - \tilde{\pi}_k\|_1^2 \leq (\pi_k - \tilde{\pi}_k) \cdot (\nabla h(\pi_k) - \nabla h(\tilde{\pi}_k)), \tag{54}$$

$$= (\pi_k - \tilde{\pi}_k) \cdot (\log \pi_k - \log \tilde{\pi}_k), \tag{55}$$

$$\stackrel{(i)}{=} (\pi_k - \tilde{\pi}_k) \cdot (\xi_k - \tilde{\xi}_k), \tag{56}$$

$$\leq \|\pi_k - \tilde{\pi}_k\|_1 \left\| \xi_k - \tilde{\xi}_k \right\|_\infty, \tag{57}$$

$$= \|\pi_k - \tilde{\pi}_k\|_1 \, \alpha\beta^{M-1} \left\| \tilde{Q}_\tau^{k-M} \right\|_\infty, \tag{58}$$

$$\implies \|\pi_k - \tilde{\pi}_k\|_1 \leq \alpha\beta^{M-1} \left\| \tilde{Q}_\tau^{k-M} \right\|_\infty, \tag{59}$$

$$\leq \alpha\beta^{M-1}(\bar{R} + \epsilon_{eval}). \tag{60}$$

In $(i)$, the normalizing constants $\log Z = \log \sum_a \exp(\xi(a))$ and $\log \tilde{Z} = \log \sum_a \exp(\tilde{\xi}(a))$ do not appear because their dot product with $\pi - \tilde{\pi}_k$ is equal to 0, as they have constant values for all actions. Combining both results, we have

$$\|\pi_k - \tilde{\pi}_k\|_1 \leq \min \left\{ 2, \alpha\beta^{M-1}(\bar{R} + \epsilon_{eval}) \right\}. \tag{61}$$

which holds for all $s \in S$ and thus also for the state $\arg\max_{s \in S} \|(\pi - \tilde{\pi})(s)\|_1$. $\qquad\square$

In the case of an update

$$\xi_{k+1} = \beta\xi_k + \alpha \left( \tilde{Q}_\tau^k - \beta^M \tilde{Q}_\tau^{k-M} \right), \tag{62}$$

we get that for any $k \geq 0$ holds

$$Q_\tau^{k+1} \geq \tilde{Q}_\tau^k - \min \left\{ 2, \alpha\beta^{M-1}(\bar{R} + \epsilon_{eval}) \right\} \gamma\beta^M \frac{\bar{R} + \epsilon_{eval}}{1 - \gamma} - \frac{\epsilon_{eval}}{1 - \gamma}. \tag{63}$$

For simplicity, we will further analyse the case of

$$Q_\tau^{k+1} \geq \tilde{Q}_\tau^k - 2\gamma\beta^M \frac{\bar{R} + \epsilon_{eval}}{1 - \gamma} - \frac{\epsilon_{eval}}{1 - \gamma}. \tag{64}$$

Note that for $k \leq M$, Eq. 64 can be replaced by a stronger $Q_\tau^{k+1} \geq \tilde{Q}_\tau^k - \frac{\epsilon_{eval}}{1-\gamma}$ as $\xi_k - \tilde{\xi}_k = 0$, but for simplicity we only consider Eq. 64.

### A.3.2 PROOF OF THEOREM 4.2

To prove Thm. 4.2, we first need the following Lemma, that uses the approximate policy improvement bounds of vanilla finite-memory EPMD in Cor. 4.1.1, to show a relation between the Q-function $Q_\tau^k$ and the sum of Q-functions $\xi_k$. Further, we show the final error that is introduced by having an approximate policy evaluation and how it affects the final convergence results.

**Lemma A.2.** *After $k \geq 0$ iterations of Eq. 62, we have* $\left\| Q_\tau^\star - \tilde{Q}_\tau^{k+1} \right\|_\infty \leq \gamma \|Q_\tau^\star - \tau\xi_{k+1}\|_\infty +$ $\frac{1-\beta^M}{1-\beta}\gamma\epsilon + \frac{1-\beta^{M+1}}{1-\beta}\epsilon_{eval} + \gamma\beta^M \bar{R}$, *where* $\epsilon = 2\gamma\beta^M \frac{\bar{R}+\epsilon_{eval}}{1-\gamma} + \frac{\epsilon_{eval}}{1-\gamma}$.

*Proof.* For all $s \in S$ and $a \in A$

$$(Q_\tau^\star - Q_\tau^{k+1})(s,a) = (T_\tau^\star Q_\tau^\star)(s,a) - \left( R(s,a) + \gamma\mathbb{E}_{s',a'}[Q_\tau^{k+1}(s',a') - \tau h(\pi_{k+1}(s'))] \right) \tag{65}$$

$$= (T_\tau^\star Q_\tau^\star)(s,a) - \Big( R(s,a) + \gamma\mathbb{E}_{s',a'}[\tau\xi_{k+1}(s',a') - \tau h(\pi_{k+1}(s'))]$$
$$+ \gamma\mathbb{E}_{s',a'}[Q_\tau^{k+1}(s',a') - \tau\xi_{k+1}(s',a')] \Big). \tag{66}$$

Looking at the first inner term, using the entropy maximizing nature of $\pi_{k+1}$ as defined in Eq. 17, and using the definition of the Bellman optimality operator $T_\tau^\star$ gives

$$R(s,a) + \gamma\mathbb{E}_{s'}[\tau\xi_{k+1}(s') \cdot \pi_{k+1}(s') - \tau h(\pi_{k+1}(s'))] = R(s,a)$$
$$+ \gamma\mathbb{E}_{s'}[\max_{p\in\Delta(A)} \tau\xi_{k+1}(s') \cdot p - \tau h(p)] \tag{67}$$

$$= (T_\tau^\star \tau\xi_{k+1})(s,a) \tag{68}$$

For the second inner term, using the definition of $\xi_{k+1}$, the fact that $\tau\alpha = 1 - \beta$ and the definition of $\epsilon$, we have for all $s \in S$ and $a \in A$

$$Q_\tau^{k+1} - \tau\xi_{k+1} = Q_\tau^{k+1} - (1-\beta)\sum_{i=0}^{M-1}\beta^i\tilde{Q}_\tau^{k-i} \tag{69}$$

$$= \sum_{i=0}^{M}\beta^i Q_\tau^{k+1-i} - \sum_{i=1}^{M}\beta^i Q_\tau^{k+1-i} + \sum_{i=0}^{M-1}\beta^{i+1}\tilde{Q}_\tau^{k-i} - \sum_{i=0}^{M-1}\beta^i\tilde{Q}_\tau^{k-i} \tag{70}$$

$$= \sum_{i=0}^{M-1}\beta^i(Q_\tau^{k+1-i} - \tilde{Q}_\tau^{k-i}) + \sum_{i=1}^{M}\beta^i(\tilde{Q}_\tau^{k+1-i} - Q_\tau^{k+1-i}) + \beta^M Q_\tau^{k+1-M} \tag{71}$$

$$\geq -\sum_{i=0}^{M-1}\beta^i\epsilon - \beta^M\bar{R} + \sum_{i=1}^{M}\beta^i(\tilde{Q}_\tau^{k+1-i} - Q_\tau^{k+1-i}) \tag{72}$$

$$= -\frac{1-\beta^M}{1-\beta}\epsilon - \beta^M\bar{R} + \sum_{i=1}^{M}\beta^i(\tilde{Q}_\tau^{k+1-i} - Q_\tau^{k+1-i}). \tag{73}$$

Using successively Eq. 68 and Eq. 73 back into Eq. 66 yields

$$(Q_\tau^\star - Q_\tau^{k+1})(s,a) = (T_\tau^\star Q_\tau^\star)(s,a) - (T_\tau^\star\tau\xi_{k+1})(s,a)$$
$$- \gamma\mathbb{E}_{s',a'}[Q_\tau^{k+1}(s',a') - \tau\xi_{k+1}(s',a')], \tag{74}$$
$$\leq (T_\tau^\star Q_\tau^\star)(s,a) - (T_\tau^\star\tau\xi_{k+1})(s,a)$$
$$+ \frac{\gamma(1-\beta^M)}{1-\beta}\epsilon + \gamma\beta^M\bar{R} - \gamma\mathbb{E}_{s',a'}\sum_{i=1}^{M}\beta^i(\tilde{Q}_\tau^{k+1-i} - Q_\tau^{k+1-i}). \tag{75}$$

Since $Q_\tau^\star - Q_\tau^{k+1} \geq 0$ and using the triangle inequality, the fact that $\mathbb{E}_{s,a}[X] \leq \|X\|_\infty$ and the contraction property of $T_\tau^\star$ completes the proof

$$\left\|Q_\tau^\star - \tilde{Q}_\tau^{k+1}\right\|_\infty \leq \left\|Q_\tau^\star - Q_\tau^{k+1}\right\|_\infty + \left\|Q_\tau^{k+1} - \tilde{Q}_\tau^{k+1}\right\|_\infty \tag{76}$$

$$\overset{(i)}{\leq} \|T_\tau^\star Q_\tau^\star - T_\tau^\star\tau\xi_{k+1}\|_\infty + \frac{\gamma(1-\beta^M)}{1-\beta}\epsilon + \gamma\beta^M\bar{R}$$
$$+ \sum_{i=0}^{M}\beta^i\left\|\tilde{Q}_\tau^{k+1-i} - Q_\tau^{k+1-i}\right\|_\infty, \tag{77}$$

$$\leq \gamma\|Q_\tau^\star - \tau\xi_{k+1}\|_\infty + \frac{\gamma(1-\beta^M)}{1-\beta}\epsilon + \gamma\beta^M\bar{R} + \frac{1-\beta^{M+1}}{1-\beta}\epsilon_{eval}. \tag{78}$$

Here $(i)$ is due to Eq. 75 and $\gamma < 1$. This completes the proof.

$\square$

**Theorem A.3** (Convergence of approximate vanilla finite-memory EPMD). *After $k \geq 0$ iterations of Eq. 8, we have that* $\left\|Q_\tau^\star - \tilde{Q}_\tau^k\right\|_\infty \leq \gamma d^k\|Q_\tau^\star\|_\infty + C_1\beta^M + \frac{\epsilon_{eval}}{(1-\gamma)^2(1-\beta)}$, *with* $d = \beta + \gamma(1-\beta) < 1$, $C_1 = \frac{2\gamma\bar{R}}{1-\gamma}\left(1 + \frac{\gamma(1-\beta^M)}{(1-\beta)(1-\gamma)}\right) + \frac{(2-\gamma)\gamma\epsilon_{eval}}{(1-\gamma)^2(1-\beta)}$.

This theorem states that the approximate vanilla finite-memory EPMD algorithm converges to an error that consists of two components: the first one scales with $\beta^M$ and thus should become negligible for large enough $M$ and the second one fully depends on $\epsilon_{eval}$ and is small only if $\epsilon_{eval}$ is small too.

*Proof.* From the definition of $\xi_{k+1}$ in Eq. 62 and the triangle inequality we get

$$\left\|Q_\tau^\star - \tau\xi_{k+1}\right\|_\infty = \left\|Q_\tau^\star - \beta\tau\xi_k - (1-\beta)\tilde{Q}_\tau^k + (1-\beta)\beta^M\tilde{Q}_\tau^{k-M}\right\|_\infty, \tag{79}$$

$$\leq \beta\left\|Q_\tau^\star - \tau\xi_k\right\|_\infty + (1-\beta)\left\|Q_\tau^\star - \tilde{Q}_\tau^k\right\|_\infty + (1-\beta)\beta^M\left\|\tilde{Q}_\tau^{k-M}\right\|_\infty, \tag{80}$$

$$\begin{aligned}\leq \beta\left\|Q_\tau^\star - \tau\xi_k\right\|_\infty \\ + (1-\beta)\gamma\left\|Q_\tau^\star - \tau\xi_k\right\|_\infty + \gamma(1-\beta^M)\epsilon + (1-\beta^{M+1})\epsilon_{eval} \\ + (1-\beta)\gamma\beta^M\bar{R} + (1-\beta)\beta^M\left\|\tilde{Q}_\tau^{k-M}\right\|_\infty,\end{aligned} \tag{81}$$

$$\begin{aligned}\leq (\beta + \gamma(1-\beta))\left\|Q_\tau^\star - \tau\xi_k\right\|_\infty + \gamma(1-\beta^M)\epsilon \\ + (1+\gamma)(1-\beta)\beta^M\bar{R} + (1+\beta^M)\epsilon_{eval}.\end{aligned} \tag{82}$$

Where in the last inequality we used the fact that $\left\|\tilde{Q}_\tau^{k-M}\right\|_\infty \leq \bar{R} + \epsilon_{eval}$ and $1 - \beta^{M+1} + (1-\beta)\beta^M = 1 + \beta^M - 2\beta^{M+1} \leq 1 + \beta^M$. Letting

$$d := \beta + \gamma(1-\beta), \tag{83}$$

one can show by induction, using the fact that $\xi_0 = 0$, that

$$\begin{aligned}\left\|Q_\tau^\star - \tau\xi_{k+1}\right\|_\infty &\leq d^{k+1}\left\|Q_\tau^\star\right\|_\infty \\ &+ \sum_{i=0}^k d^i\left[\gamma(1-\beta^M)\epsilon + (1+\gamma)(1-\beta)\beta^M\bar{R} + (1+\beta^M)\epsilon_{eval}\right],\end{aligned} \tag{84}$$

$$\leq d^{k+1}\left\|Q_\tau^\star\right\|_\infty + \frac{\gamma(1-\beta^M)\epsilon + (1+\gamma)(1-\beta)\beta^M\bar{R} + (1+\beta^M)\epsilon_{eval}}{1-d}, \tag{85}$$

$$= d^{k+1}\left\|Q_\tau^\star\right\|_\infty + \frac{(1+\gamma)\beta^M\bar{R}}{1-\gamma} + \frac{\gamma(1-\beta^M)\epsilon + (1+\beta^M)\epsilon_{eval}}{(1-\gamma)(1-\beta)}. \tag{86}$$

For Eq. 85, we used the fact that $\sum_{i=0}^k d^i = \frac{1-d^{k+1}}{1-d} \leq \frac{1}{1-d}$. Using Eq. 86 in Eq. 78 finally gives

$$\begin{aligned}\left\|Q_\tau^\star - \tilde{Q}_\tau^{k+1}\right\|_\infty &\leq \gamma d^{k+1}\left\|Q_\tau^\star\right\|_\infty \\ &+ \left[\gamma\beta^M + \frac{(1+\gamma)\gamma\beta^M}{1-\gamma}\right]\bar{R} \\ &+ \left[\frac{\gamma^2(1-\beta^M)}{(1-\gamma)(1-\beta)} + \frac{\gamma(1-\beta^M)}{1-\beta}\right]\epsilon \\ &+ \left[\frac{\gamma(1+\beta^M)}{(1-\gamma)(1-\beta)} + \frac{1-\beta^{M+1}}{1-\beta}\right]\epsilon_{eval}\end{aligned} \tag{87}$$

Now, let us analyse more closely the constants in front of $\bar{R}$ and $\epsilon_{eval}$. First, let us simplify the constant in front of $\epsilon$, we get $\frac{\gamma^2(1-\beta^M)}{(1-\gamma)(1-\beta)} + \frac{\gamma(1-\beta^M)}{1-\beta} = \frac{\gamma(1-\beta^M)}{1-\beta}\left(\frac{\gamma}{1-\gamma} + 1\right) = \frac{\gamma(1-\beta^M)}{(1-\gamma)(1-\beta)}$. By inserting the value of $\epsilon$ from Lemma A.2 we obtain the following coefficient for $\bar{R}$

$$\gamma\beta^M + \frac{(1+\gamma)\gamma\beta^M}{1-\gamma} + \frac{\gamma(1-\beta^M)}{(1-\gamma)(1-\beta)}\frac{2\gamma\beta^M}{1-\gamma} = \frac{2\gamma\beta^M}{1-\gamma}\left(1 + \frac{\gamma(1-\beta^M)}{(1-\beta)(1-\gamma)}\right) \tag{88}$$

and $\epsilon_{eval}$

$$
\begin{aligned}
&\frac{\gamma(1+\beta^M)}{(1-\gamma)(1-\beta)} + \frac{1-\beta^{M+1}}{1-\beta} + \frac{\gamma(1-\beta^M)}{(1-\gamma)(1-\beta)}\frac{2\gamma\beta^M+1}{1-\gamma} \\
&\overset{(i)}{<} \frac{\gamma(1+\beta^M)}{(1-\gamma)(1-\beta)} + \frac{1-\beta^{M+1}}{1-\beta} + \frac{\gamma(1+\beta^M)}{(1-\gamma)^2(1-\beta)} \\
&\overset{(ii)}{\leq} \frac{\gamma+\gamma\beta^M+1-\gamma}{(1-\gamma)(1-\beta)} + \frac{\gamma(1+\beta^M)}{(1-\gamma)^2(1-\beta)} \\
&= \frac{1}{(1-\gamma)(1-\beta)}\left(1+\frac{\gamma}{1-\gamma}\right) + \frac{(1-\gamma+1)\gamma\beta^M}{(1-\gamma)^2(1-\beta)} \\
&= \frac{1}{(1-\gamma)^2(1-\beta)} + \frac{(2-\gamma)\gamma\beta^M}{(1-\gamma)^2(1-\beta)},
\end{aligned}
\tag{89}
$$

where in $(i)$ we use $\gamma(1-\beta^M)(2\gamma\beta^M+1) = 2\gamma^2\beta^M - 2\gamma^2\beta^{2M} + \gamma - \gamma\beta^M < 2\gamma\beta^M - \gamma\beta^M + \gamma = \gamma(1+\beta^M)$ and in $(ii)$ we cancel the negative components. Combining Eq. 87, Eq. 88 and Eq. 89, we complete the proof. $\qquad\square$

### A.4 APPROXIMATE WEIGHT-CORRECTED FINITE-MEMORY EPMD

#### A.4.1 PROOF OF THE LOGITS EXPRESSION IN SEC. 4.2

*Proof.* For $k = 0$,

$$\xi_1 = \beta 0 + \alpha \tilde{Q}_\tau^0 + \frac{\alpha \beta^M}{1 - \beta^M}(\tilde{Q}_\tau^0 - 0), \tag{90}$$

$$= \alpha \left(1 + \frac{\beta^M}{1 - \beta^M}\right) \tilde{Q}_\tau^0, \tag{91}$$

$$= \frac{\alpha}{1 - \beta^M} \tilde{Q}_\tau^0. \tag{92}$$

If it is true for $k$, then

$$\xi_{k+1} = \beta \frac{\alpha}{1 - \beta^M} \sum_{i=0}^{M-1} \beta^i \tilde{Q}_\tau^{k-1-i} + \alpha \tilde{Q}_\tau^k + \frac{\alpha \beta^M}{1 - \beta^M}(\tilde{Q}_\tau^k - \tilde{Q}_\tau^{k-M}), \tag{93}$$

$$= \frac{\alpha}{1 - \beta^M} \sum_{i=0}^{M-2} \beta^{i+1} \tilde{Q}_\tau^{k-1-i} + \frac{\alpha \beta^M}{1 - \beta^M}(\tilde{Q}_\tau^{k-M} - \tilde{Q}_\tau^{k-M}) + \frac{\alpha}{1 - \beta^M} \tilde{Q}_\tau^k, \tag{94}$$

$$= \frac{\alpha}{1 - \beta^M} \sum_{i=0}^{M-1} \beta^i \tilde{Q}_\tau^{k-i} \tag{95}$$

$\square$

#### A.4.2 PROOF OF COROLLARY 4.1.2

*Proof.* The proof is immediate from Thm. 4.1, upper-bounding $\max_{s \in S} \|(\pi_k - \tilde{\pi}_k)(s)\|_1$ by 2, using the definition of $\tilde{\xi}_k - \xi_k = \frac{\alpha \beta^{M-1}}{1 - \beta^M}(\tilde{Q}_\tau^k - \tilde{Q}_\tau^{k-M})$ in $\left\|\xi_k - \tilde{\xi}_k\right\|_\infty$ and using the fact that $\alpha \eta = \beta$. $\square$

Note that, as in Cor. 4.1.1, we could have used the expression of the logits of $\pi$ and $\tilde{\pi}$ to have a bound of $\|(\pi_k - \tilde{\pi}_k)(s)\|_1$ that depends on $\left\|\xi_k - \tilde{\xi}_k\right\|_\infty$ and ultimately on $\left\|\tilde{Q}_\tau^k - \tilde{Q}_\tau^{k-M}\right\|_\infty$. This bound becomes tighter as $k$ goes to infinity for $M$ large enough, since we show below that $\tilde{Q}_\tau^k$ converges to $Q_\tau^\star$ and thus $\max_{s \in S} \|(\pi_k - \tilde{\pi}_k)(s)\|_1$ converges to 0. Nonetheless, using this tighter bound would introduce quadratic terms $\left\|\tilde{Q}_\tau^k - \tilde{Q}_\tau^{k-M}\right\|_\infty^2$ that would complicate the overall analysis of the algorithm, and thus we use the more crude bound of 2 for $\max_{s \in S} \|(\pi_k - \tilde{\pi}_k)(s)\|_1$ in the remainder of the proofs for Sec. 4.2.

Given Theorem A.1 and Corollary 4.1.2, and in case of an update

$$\xi_{k+1} = \beta \xi_k + \alpha \tilde{Q}_\tau^k + \frac{\alpha \beta^M}{1 - \beta^M}(\tilde{Q}_\tau^k - \tilde{Q}_\tau^{k-M}), \tag{96}$$

we get

$$Q_\tau^{k+1} \geq \tilde{Q}_\tau^k - 2\gamma \beta^M \frac{\left\|\tilde{Q}_\tau^k - \tilde{Q}_\tau^{k-M}\right\|_\infty}{(1 - \gamma)(1 - \beta^M)} - \frac{\epsilon_{eval}}{1 - \gamma}. \tag{97}$$

### A.5 PROOF OF LEMMA 4.3

As with Thm. 4.2, we first need an intermediary Lemma connecting $\left\|Q_\tau^\star - Q_\tau^{k+1}\right\|_\infty$ and $\left\|Q_\tau^\star - \tau \xi_{k+1}\right\|_\infty$ before proving Lem. 4.3.

**Lemma A.4.** *After $k \geq 0$ iterations of Eq. 96, we have* $\left\|Q_\tau^\star - \tilde{Q}_\tau^{k+1}\right\|_\infty \leq \gamma \left\|Q_\tau^\star - \tau \xi_{k+1}\right\|_\infty + \gamma \beta^{k+1} \left\|Q_\tau^0\right\|_\infty + \gamma \sum_{i=0}^k \beta^i \epsilon'_{k-i} + \frac{\epsilon_{eval}}{(1-\gamma)(1-\beta)}$, *with* $\epsilon'_k = \left(\frac{1+\gamma}{1-\gamma} - \beta\right) \frac{\beta^M}{1-\beta^M} \left\|\tilde{Q}_\tau^{k-M} - \tilde{Q}_\tau^k\right\|_\infty$.

*Proof.* Define $\epsilon_k$ as

$$\epsilon_k := \frac{2\beta^M}{1-\beta^M}\left\|\tilde{Q}_\tau^{k-M} - \tilde{Q}_\tau^k\right\|_\infty, \tag{98}$$

For all $s \in S$ and $a \in A$

$$(Q_\tau^\star - Q_\tau^{k+1})(s,a) = (T_\tau^\star Q_\tau^\star)(s,a) - \Big(R(s,a) + \gamma\mathbb{E}_{s',a'}[Q_\tau^{k+1}(s',a') - \tau h(\pi_{k+1}(s'))]\Big) \tag{99}$$

$$= (T_\tau^\star Q_\tau^\star)(s,a) - \Big(R(s,a) + \gamma\mathbb{E}_{s',a'}[\tau\xi_{k+1}(s',a') - \tau h(\pi_{k+1}(s'))]+ \tag{100}$$

$$\gamma\mathbb{E}_{s',a'}[Q_\tau^{k+1}(s',a') - \tau\xi_{k+1}(s',a')]\Big)$$

Looking at the first inner term and using the entropy maximizing nature of $\pi_{k+1}$ as defined in Eq. 17 gives

$$R(s,a) + \gamma\mathbb{E}_{s'}[\tau\xi_{k+1}(s')\cdot\pi_{k+1}(s') - \tau h(\pi_{k+1}(s'))] \tag{101}$$

$$= R(s,a) + \gamma\mathbb{E}_{s'}[\max_{p\in\Delta(A)}\tau\xi_{k+1}(s')\cdot p - \tau h(p)] = (T_\tau^\star\tau\xi_{k+1})(s,a) \tag{102}$$

For the second inner term, using the recursive definition of $\xi_{k+1}$ in Eq. 96 gives

$$Q_\tau^{k+1} - \tau\xi_{k+1} = Q_\tau^{k+1} - \left(\beta\tau\xi_k + (1-\beta)\tilde{Q}_\tau^k + \frac{(1-\beta)\beta^M}{1-\beta^M}(\tilde{Q}_\tau^k - \tilde{Q}_\tau^{k-M})\right), \tag{103}$$

$$= \beta(\tilde{Q}_\tau^k - \tau\xi_k) + Q_\tau^{k+1} - \tilde{Q}_\tau^k - \frac{(1-\beta)\beta^M}{1-\beta^M}(\tilde{Q}_\tau^k - \tilde{Q}_\tau^{k-M}), \tag{104}$$

$$\geq \beta(Q_\tau^k - \tau\xi_k) - \frac{\gamma\epsilon_k}{1-\gamma} - \frac{(1-\beta)\epsilon_k}{2} - \frac{\epsilon_{eval}}{1-\gamma} + \beta\left(\tilde{Q}_\tau^k - Q_\tau^k\right). \tag{105}$$

Letting

$$\epsilon_k' := \frac{\gamma\epsilon_k}{1-\gamma} + \frac{(1-\beta)\epsilon_k}{2} = \left(\frac{1+\gamma}{1-\gamma} - \beta\right)\frac{\beta^M}{1-\beta^M}\left\|\tilde{Q}_\tau^{k-M} - \tilde{Q}_\tau^k\right\|_\infty, \tag{106}$$

one can easily show by induction that

$$Q_\tau^{k+1} - \tau\xi_{k+1} \geq \beta^{k+1}Q_\tau^0 - \sum_{i=0}^k \beta^i\epsilon_{k-i}' - \sum_{i=0}^k \beta^i\frac{\epsilon_{eval}}{1-\gamma} - \frac{\beta(1-\beta^{k+1})}{1-\beta}\epsilon_{eval} \tag{107}$$

$$\geq \beta^{k+1}Q_\tau^0 - \sum_{i=0}^k \beta^i\epsilon_{k-i}' - \frac{\epsilon_{eval}}{(1-\gamma)(1-\beta)} - \frac{\beta\epsilon_{eval}}{1-\beta}. \tag{108}$$

The inequality uses the fact that $\xi_0 = 0$. Using successively Eq. 102 and Eq. 108 back into Eq. 100 yields

$$(Q_\tau^\star - Q_\tau^{k+1})(s,a) = (T_\tau^\star Q_\tau^\star)(s,a) - (T_\tau^\star\tau\xi_{k+1})(s,a) - \gamma\mathbb{E}_{s',a'}[Q_\tau^{k+1}(s',a') - \tau\xi_{k+1}(s',a')], \tag{109}$$

$$\leq (T_\tau^\star Q_\tau^\star)(s,a) - (T_\tau^\star\tau\xi_{k+1})(s,a) - \gamma\mathbb{E}_{s',a'}[\beta^{k+1}Q_\tau^0(s',a')]$$

$$+ \gamma\sum_{i=0}^k \beta^i\epsilon_{k-i}' + \frac{\gamma\beta}{1-\beta}\epsilon_{eval} + \frac{\gamma\epsilon_{eval}}{(1-\gamma)(1-\beta)}. \tag{110}$$

Since $Q_\tau^\star - Q_\tau^{k+1} \geq 0$ and using the triangle inequality, the fact that $\mathbb{E}_{s,a}[Q_\tau^0(s,a)] \leq \|Q_\tau^0\|_\infty$, and the contraction property of $T_\tau^\star$ gives us

$$\|Q_\tau^\star - Q_\tau^{k+1}\|_\infty \leq \|T_\tau^\star Q_\tau^\star - T_\tau^\star\tau\xi_{k+1}\|_\infty + \gamma\beta^{k+1}\|Q_\tau^0\|_\infty$$

$$+ \gamma\sum_{i=0}^k \beta^i\epsilon_{k-i}' + \frac{\gamma\beta}{1-\beta}\epsilon_{eval} + \frac{\gamma\epsilon_{eval}}{(1-\gamma)(1-\beta)} \tag{111}$$

$$\leq \gamma\|Q_\tau^\star - \tau\xi_{k+1}\|_\infty + \gamma\beta^{k+1}\|Q_\tau^0\|_\infty$$

$$+ \gamma\sum_{i=0}^k \beta^i\epsilon_{k-i}' + \frac{\gamma\beta}{1-\beta}\epsilon_{eval} + \frac{\gamma\epsilon_{eval}}{(1-\gamma)(1-\beta)}. \tag{112}$$

Finally, using

$$\left\|Q_\tau^\star - \tilde{Q}_\tau^{k+1}\right\|_\infty \le \left\|Q_\tau^\star - Q_\tau^{k+1}\right\|_\infty + \left\|Q_\tau^{k+1} - \tilde{Q}_\tau^{k+1}\right\|_\infty \le \left\|Q_\tau^\star - Q_\tau^{k+1}\right\|_\infty + \epsilon_{eval} \quad (113)$$

and also simplifying the constants $\frac{\gamma\beta}{1-\beta} + 1 = \frac{1-\beta+\gamma\beta}{1-\beta} \le \frac{1}{1-\beta}$ and $\frac{1}{1-\beta} + \frac{\gamma}{(1-\gamma)(1-\beta)} = \frac{1}{1-\beta}(1 + \frac{\gamma}{1-\gamma}) = \frac{1}{(1-\gamma)(1-\beta)}$, we obtain the statement of the lemma. $\qquad\square$

We now state a more general form of Lemma 4.3 and prove it.

**Lemma A.5.** *Let* $x_{k+1} = d_1 x_k + d_2 x_{k-M} + \frac{\epsilon_{eval}}{1-\gamma}$ *be a sequence such that* $\forall k < 0$, $x_k = \frac{\|Q_\tau^\star\|_\infty}{\gamma}$, $x_0 = \|Q_\tau^\star\|_\infty + \left\|\tilde{Q}_\tau^0\right\|_\infty + \frac{\epsilon_{eval}}{(1-\gamma)(1-\beta)}$, $d_1 := \beta + \gamma\frac{1-\beta}{1-\beta^M} + \gamma c_2$, $d_2 := \frac{2c_1\gamma^2}{1-\gamma}$, $c_1 := \frac{\beta^M}{1-\beta^M}$, *and* $c_2 := \left(\frac{1+\gamma}{1-\gamma} - \beta\right) c_1$. *After* $k \ge 0$ *iterations of Eq. 12, we have that* $\left\|Q_\tau^\star - \tilde{Q}_\tau^k\right\|_\infty \le x_k$.

*Proof.* Define the following constants

$$c_1 := \frac{\beta^M}{1-\beta^M}, \text{ and } c_2 := \left(\frac{1+\gamma}{1-\gamma} - \beta\right) c_1. \quad (114)$$

Let a sequence $x_k$ defined by $\forall k < 0$, $x_k = \frac{\|Q_\tau^\star\|_\infty}{\gamma}$ and let

$$x_0 = \|Q_\tau^\star\|_\infty + \left\|Q_\tau^0\right\|_\infty + \frac{\epsilon_{eval}}{(1-\gamma)(1-\beta)}. \quad (115)$$

For subsequent terms, we define $x_k$ by the recursive definition, $\forall k \ge 0$

$$x_{k+1} = \gamma \left(\frac{1-\beta}{1-\beta^M} \sum_{i=0}^{M-1} \beta^i x_{k-i} + \beta^{k+1}\frac{\left\|Q_\tau^0\right\|_\infty}{\gamma} + c_2 \sum_{i=0}^{k} \beta^i (x_{k-i} + x_{k-i-M})\right) + \frac{\epsilon_{eval}}{(1-\gamma)(1-\beta)}. \quad (116)$$

Note that $x_0$ can also be recovered by Eq. 116, for $k = -1$. Now, let us simplify Eq. 116. Using this recursive definition, we have $\forall k \ge 0$

$$x_{k+1} = \gamma\frac{1-\beta}{1-\beta^M} \sum_{i=0}^{M-1} \beta^i x_{k-i} + \beta^{k+1} \left\|Q_\tau^0\right\|_\infty + c_2\gamma \sum_{i=0}^{k} \beta^i (x_{k-i} + x_{k-i-M}) \\ + \frac{\epsilon_{eval}}{(1-\gamma)(1-\beta)}, \quad (117)$$

$$= \beta \left(\frac{\gamma(1-\beta)}{1-\beta^M} \sum_{i=0}^{M-1} \beta^i x_{k-1-i} + \beta^k \left\|Q_\tau^0\right\|_\infty + \gamma c_2 \sum_{i=0}^{k-1} \beta^i (x_{k-1-i} + x_{k-1-i-M}) \right. \\ \left. + \frac{\epsilon_{eval}}{(1-\gamma)(1-\beta)}\right) + \frac{\gamma(1-\beta)}{1-\beta^M}\left(x_k - \beta^M x_{k-M}\right) + \gamma c_2 \left(x_k + x_{k-M}\right) \\ + \frac{\epsilon_{eval}}{1-\gamma}, \quad (118)$$

$$\overset{(i)}{=} \beta x_k + \gamma\left(\frac{1-\beta}{1-\beta^M}\left(x_k - \beta^M x_{k-M}\right) + c_2\left(x_k + x_{k-M}\right)\right) + \frac{\epsilon_{eval}}{1-\gamma}, \quad (119)$$

$$= \left(\beta + \gamma\frac{1-\beta}{1-\beta^M} + \gamma c_2\right) x_k + \gamma\left(c_2 - \frac{\beta^M(1-\beta)}{1-\beta^M}\right) x_{k-M} + \frac{\epsilon_{eval}}{1-\gamma} \quad (120)$$

$$= \left(\beta + \gamma\frac{1-\beta}{1-\beta^M} + \gamma c_2\right) x_k + \frac{2c_1\gamma^2}{1-\gamma} x_{k-M} + \frac{\epsilon_{eval}}{1-\gamma} \quad (121)$$

In (i) we used the recursive definition of $x_k$ which is also valid for $x_0$. Letting

$$d_1 := \beta + \gamma\frac{1-\beta}{1-\beta^M} + \gamma c_2, \text{ and } d_2 := \frac{2c_1\gamma^2}{1-\gamma}, \quad (122)$$

$x_{k+1}$ for all $k \geq 0$ can be more compactly defined by

$$x_{k+1} = d_1 x_k + d_2 x_{k-M} + \frac{\epsilon_{eval}}{1 - \gamma}. \tag{123}$$

Let us now prove that $\left\| Q_\tau^\star - \tilde{Q}_\tau^k \right\|_\infty \leq x_k$ by induction. For $k = 0$, we have that

$$\left\| Q_\tau^\star - \tilde{Q}_\tau^0 \right\|_\infty \leq \left\| Q_\tau^\star - Q_\tau^0 \right\|_\infty + \left\| Q_\tau^0 - \tilde{Q}_\tau^0 \right\|_\infty \tag{124}$$

$$\leq \left\| Q_\tau^\star \right\|_\infty + \left\| Q_\tau^0 \right\|_\infty + \epsilon_{eval}, \tag{125}$$

$$\leq x_0. \tag{126}$$

and for $k < 0$, we have that

$$\left\| Q_\tau^\star - \tilde{Q}_\tau^k \right\|_\infty = \left\| Q_\tau^\star \right\|_\infty, \tag{127}$$

$$\leq \frac{\left\| Q_\tau^\star \right\|_\infty}{\gamma}, \tag{128}$$

$$= x_k. \tag{129}$$

Now assume that $\left\| Q_\tau^\star - \tilde{Q}_\tau^i \right\|_\infty \leq x_i$ is true for all $i \leq k$ and let us prove that $\left\| Q_\tau^\star - \tilde{Q}_\tau^{k+1} \right\|_\infty \leq x_{k+1}$. First, we note that

$$\left\| Q_\tau^\star - \tau \xi_{k+1} \right\|_\infty = \left\| Q_\tau^\star - \tau \frac{\alpha}{1 - \beta^M} \sum_{i=0}^{M-1} \beta^i \tilde{Q}_\tau^{k-i} \right\|_\infty, \tag{130}$$

$$= \left\| \frac{1 - \beta}{1 - \beta^M} \sum_{i=0}^{M-1} \beta^i Q_\tau^\star - \frac{1 - \beta}{1 - \beta^M} \sum_{i=0}^{M-1} \beta^i \tilde{Q}_\tau^{k-i} \right\|_\infty, \tag{131}$$

$$\leq \frac{1 - \beta}{1 - \beta^M} \sum_{i=0}^{M-1} \beta^i \left\| Q_\tau^\star - \tilde{Q}_\tau^{k-i} \right\|_\infty, \tag{132}$$

$$\leq \frac{1 - \beta}{1 - \beta^M} \sum_{i=0}^{M-1} \beta^i x_{k-i}. \tag{133}$$

We also have that

$$\epsilon_k' = c_2 \left\| \tilde{Q}_\tau^{k-M} - \tilde{Q}_\tau^k \right\|_\infty, \tag{134}$$

$$\leq c_2 \left( \left\| Q_\tau^\star - \tilde{Q}_\tau^{k-M} \right\|_\infty + \left\| Q_\tau^\star - \tilde{Q}_\tau^k \right\|_\infty \right) \tag{135}$$

$$\leq c_2 (x_k + x_{k-M}). \tag{136}$$

Finally, using Eq. 133, Eq. 136 and $\left\| Q_\tau^0 \right\|_\infty \leq \frac{\left\| Q_\tau^\star \right\|_\infty}{\gamma}$ into Lemma A.4 completes the proof

$$\left\| Q_\tau^\star - \tilde{Q}_\tau^{k+1} \right\|_\infty \leq \gamma \left( \left\| Q_\tau^\star - \tau \xi_{k+1} \right\|_\infty + \beta^{k+1} \left\| Q_\tau^0 \right\|_\infty + \sum_{i=0}^k \beta^i \epsilon_{k-i}' \right) + \frac{\epsilon_{eval}}{(1 - \gamma)(1 - \beta)},$$

$$\tag{137}$$

$$\leq \gamma \left( \frac{1 - \beta}{1 - \beta^M} \sum_{i=0}^{M-1} \beta^i x_{k-i} + \beta^{k+1} \frac{\left\| Q_\tau^0 \right\|_\infty}{\gamma} + c_2 \sum_{i=0}^k \beta^i (x_{k-i} + x_{k-i-M}) \right)$$

$$+ \frac{\epsilon_{eval}}{(1 - \gamma)(1 - \beta)}, \tag{138}$$

$$= x_{k+1}. \tag{139}$$

$\square$

### A.6 PROOF OF THEOREM 4.4

We state a more general form for Theorem 4.4 that includes policy evaluation error and prove it below.

**Theorem A.6** (Convergence of approximate weight corrected finite-memory EPMD). *With the definitions of Lemma 4.3, if $M > \log \frac{(1-\gamma)^2(1-\beta)}{\gamma^2(3+\beta)+1-\beta}(\log \beta)^{-1}$ then $\lim_{k\to\infty} x_k \leq \frac{\epsilon_{eval}}{(1-\gamma)(1-d_1-d_2)}$. Moreover, $\forall k \geq 0$, $\left\|Q_\tau^\star - \tilde{Q}_\tau^k\right\|_\infty \leq (d_1 + d_2 d_3^{-1})^k \max\left\{\frac{\|Q_\tau^\star\|_\infty}{\gamma}, \|Q_\tau^\star\|_\infty + \|Q_\tau^0\|_\infty\right\} + \frac{\epsilon_{eval}}{(1-\gamma)(1-d_1-d_2)}$, where $d_3 := \left(d_1^M + d_2 \frac{1-d_1^M}{1-d_1}\right)$ and $\lim_{M\to\infty} d_1 + d_2 d_3^{-1} = \beta + \gamma(1-\beta)$.*

*Proof.* Let us find a value of $M$ such that

$$d_1 + d_2 < 1, \tag{140}$$

$$\Leftrightarrow \beta(1-\beta^M) + \gamma(1-\beta) + \gamma\left(\frac{1+\gamma}{1-\gamma} - \beta\right)\beta^M + \frac{2\gamma^2\beta^M}{1-\gamma} < 1 - \beta^M, \tag{141}$$

$$\Leftrightarrow \beta - \beta^{M+1} - \gamma\beta + \gamma\frac{1+\gamma}{1-\gamma}\beta^M - \gamma\beta^{M+1} + \beta^M + \frac{2\gamma^2\beta^M}{1-\gamma} < 1 - \gamma, \tag{142}$$

$$\Leftrightarrow (1-\gamma)\beta + \frac{\gamma^2(3+\beta)+1-\beta}{1-\gamma}\beta^M < 1 - \gamma, \tag{143}$$

$$\Leftrightarrow \beta^M < \frac{(1-\gamma)^2(1-\beta)}{\gamma^2(3+\beta)+1-\beta}, \tag{144}$$

$$\Leftrightarrow M \log \beta < \log \frac{(1-\gamma)^2(1-\beta)}{\gamma^2(3+\beta)+1-\beta}, \tag{145}$$

$$\Leftrightarrow M > \log \frac{(1-\gamma)^2(1-\beta)}{\gamma^2(3+\beta)+1-\beta}(\log \beta)^{-1}. \tag{146}$$

We will now show that for the values of $M$ that satisfy Eq. 146, the sequence $x_k$ converges to some finite error that depends on $\epsilon_{eval}$ as $k$ goes to infinity. To simplify the analysis of $x_k$ we study a slightly modified version thereof that has the same recursive definition $x_{k+1} = d_1 x_k + d_2 x_{k-M} + \frac{\epsilon_{eval}}{1-\gamma}$ but replaces the terms $x_{-k}, \forall k \geq 0$ with $x_{-k} = \max\left\{\frac{\|Q_\tau^\star\|_\infty}{\gamma}, \|Q_\tau^\star\|_\infty + \|Q_\tau^0\|_\infty + \frac{\epsilon_{eval}}{(1-\gamma)(1-\beta)}\right\}$. Clearly, this modified sequence upper-bounds the previous sequence.

To simplify the analysis, we first analyse another sequence $y_k$ that for $k \leq 0$ is identical to $x_k$, but for $k \geq 0$ it evolves following the next law $y_{k+1} = d_1 y_k + d_2 y_{k-M}$. Now, if $M$ is such that $d_1 + d_2 < 1$, then the sequence $y_k$ is constant from $y_{-M}$ to $y_0$ and is strictly decreasing thereafter, since for $y_1$ we have

$$y_1 = d_1 y_0 + d_2 y_{-M}, \tag{147}$$

$$= (d_1 + d_2)y_0, \tag{148}$$

$$< y_0. \tag{149}$$

Then, $\forall k \geq 1$

$$y_{k+1} = d_1 y_k + d_2 y_{k-M}, \tag{150}$$

$$< d_1 y_{k-1} + d_2 y_{k-M-1}, \tag{151}$$

$$= y_k. \tag{152}$$

Since the sequence is decreasing and lower bounded by 0, it has a limit due to the monotone convergence theorem. Let us study the convergence of a sub-sequence. Let for any integer $a > 0$

$$y_{aM+a} = d_1 y_{aM+a-1} + d_2 y_{aM+a-1-M}, \tag{153}$$

$$< (d_1 + d_2)y_{aM+a-1-M}, \tag{154}$$

$$= (d_1 + d_2)y_{(a-1)M+(a-1)}, \tag{155}$$

$$< (d_1 + d_2)^a y_0. \tag{156}$$

Thus, $\lim_{a \to \infty} y_{aM+a} = 0$, which implies that $\lim_{k \to \infty} y_k = 0$.

Further, let us show that for all $k$,

$$x_k \leq y_k + \sum_{i=0}^{k-1} (d_1 + d_2)^i \frac{\epsilon_{eval}}{1 - \gamma} = y_k + C(k) \frac{\epsilon_{eval}}{1 - \gamma}, \tag{157}$$

and therefore, if we simplify the above expression, then for all $k$

$$x_k \leq y_k + \frac{\epsilon_{eval}}{(1 - \gamma)(1 - d_1 - d_2)}. \tag{158}$$

We do it by mathematical induction. First,

$$x_1 = d_1 x_0 + d_2 x_{-M} + \frac{\epsilon_{eval}}{1 - \gamma} = d_1 y_0 + d_2 y_{-M} + \frac{\epsilon_{eval}}{1 - \gamma} = y_1 + \frac{\epsilon_{eval}}{1 - \gamma}. \tag{159}$$

Then, let us assume that Eq. 157 holds for any $i \leq k$, now we show that it also holds for $k + 1$

$$x_{k+1} = d_1 x_k + d_2 x_{k-M} + \frac{\epsilon_{eval}}{1 - \gamma} \tag{160}$$

$$\leq d_1 \left( y_k + C(k) \frac{\epsilon_{eval}}{1 - \gamma} \right) + d_2 \left( y_{k-M} + C(k - M) \frac{\epsilon_{eval}}{1 - \gamma} \right) + \frac{\epsilon_{eval}}{1 - \gamma} \tag{161}$$

$$\leq y_{k+1} + \max \{ C(k), C(k - M) \} (d_1 + d_2) \frac{\epsilon_{eval}}{1 - \gamma} + \frac{\epsilon_{eval}}{1 - \gamma} \tag{162}$$

$$\overset{(i)}{=} y_{k+1} + C(k)(d_1 + d_2) \frac{\epsilon_{eval}}{1 - \gamma} + \frac{\epsilon_{eval}}{1 - \gamma} \tag{163}$$

$$= y_{k+1} + \sum_{i=0}^{k} (d_1 + d_2)^i \frac{\epsilon_{eval}}{1 - \gamma}. \tag{164}$$

Here, in $(i)$ we use the definition of $C(k)$ from Eq. 157 and its monotonicity that comes out of it. Therefore, we get that $\lim_{k \to \infty} x_k \leq \lim_{k \to \infty} \left( y_k + \sum_{i=0}^{k-1} (d_1 + d_2)^i \frac{\epsilon_{eval}}{1 - \gamma} \right) = 0 + \sum_{i=0}^{\infty} (d_1 + d_2)^i \frac{\epsilon_{eval}}{1 - \gamma} = \frac{\epsilon_{eval}}{(1 - \gamma)(1 - (d_1 + d_2))}$, which completes the first part of our proof. Now, let us have a closer look on the convergence speed.

To better characterize the convergence of $x_k$, we again analyse the sequence $y_k$. First, we note that the constant $d_1 \geq \beta + \gamma(1 - \beta)$ is typically very close to 1, whereas $d_2 \to 0$ as $M \to \infty$. The sequence $y_k$ thus behaves almost as $d_1^k y_0$. A much tighter upper-bounding sequence than that of Eq. 156 can be obtained using the following inequalities

$$y_k = d_1 y_{k-1} + d_2 y_{k-1-M}, \tag{165}$$

$$= d_1^M y_{k-M} + d_2 \sum_{i=0}^{M-1} d_1^i y_{k-1-M-i}, \tag{166}$$

$$\geq \left( d_1^M + d_2 \frac{1 - d_1^M}{1 - d_1} \right) y_{k-M}, \tag{167}$$

where we have used in the last inequality the fact that $y_k$ is a decreasing sequence. Let

$$d_3 := \left( d_1^M + d_2 \frac{1 - d_1^M}{1 - d_1} \right), \tag{168}$$

then we can upper bound the sequence $y_k$ by

$$y_{k+1} = \left( d_1 + d_2 d_3^{-1} \right) y_k + d_2 (y_{k-M} - d_3^{-1} y_k), \tag{169}$$

$$\leq \left( d_1 + d_2 d_3^{-1} \right) y_k + d_2 (y_{k-M} - d_3^{-1} d_3 y_{k-M}), \tag{170}$$

$$= \left( d_1 + d_2 d_3^{-1} \right) y_k, \tag{171}$$

$$\leq \left( d_1 + d_2 d_3^{-1} \right)^{k+1} y_0. \tag{172}$$

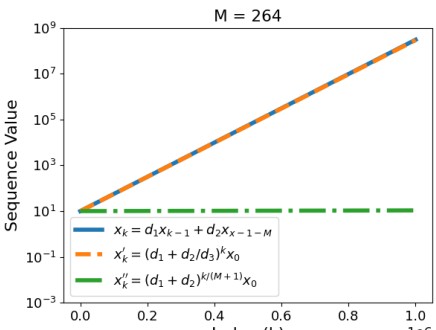 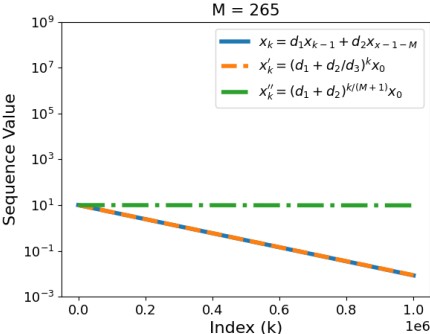

Figure 5: Evolution of $x_k$ for two successive values of $M$, one being large enough for $x_k$ to converge. The plot additionally shows the sequence $x_k'$ introduced by Thm. 4.4 that closely follows the behavior of $x_k$. See text for more details.

Now to study the limit $\lim_{M \to \infty} d_1 + d_2 d_3^{-1}$, let us first start with the rightmost term

$$d_2 d_3^{-1} \leq \frac{d_2}{d_1^M}, \tag{173}$$

$$\leq \frac{d_2}{(\beta + \gamma(1-\beta))^M}, \tag{174}$$

$$= \frac{1}{(\beta + \gamma(1-\beta))^M} \frac{2\beta^M \gamma^2}{(1-\gamma)(1-\beta^M)}, \tag{175}$$

$$= \left(\frac{\beta}{\beta + \gamma(1-\beta)}\right)^M \frac{2\gamma^2}{(1-\gamma)(1-\beta^M)}. \tag{176}$$

Since $\beta < \beta + \gamma(1-\beta)$, then clearly $\lim_{M \to \infty} d_2 d_3^{-1} = 0$, and from the definition of $d_1$ one can see that $\lim_{M \to \infty} d_1 = \beta + \gamma(1-\beta)$. Combining the result above with Eq. 158, we obtain the statement of the theorem. $\qquad\square$

To illustrate how close the sequence $\left(d_1 + d_2 d_3^{-1}\right)^k x_0$ is to $x_k$, let us take a numerical example with $\gamma = 0.99$ and $\beta = 0.95$. In this case, we have that $d_1 + d_2 < 1$ whenever $M \geq 265$. At $M = 265$ we have that $d_1 \approx 0.9997$ and $d_2 \approx 0.0002$. In Fig. 5 we plot the three sequences $x_k$, $x_k' = \left(d_1 + d_2 d_3^{-1}\right)^k x_0$ and $x_k'' = (d_1 + d_2)^{k/(M+1)} x_0$ for $M = 264$ and $M = 265$ and we see that $x_k'$ converges to zero for the same $M$ as $x_k$ and is almost indistinguishable from the latter, whereas $x_k''$ is a much more loose upper-bounding sequence at $M = 265$.

# B    ADDITIONAL EXPERIMENT RESULTS

## B.1    COMPARISON WITH DEEP RL BASELINES

We summarize all performance comparisons in Fig. 6 and Table 2.

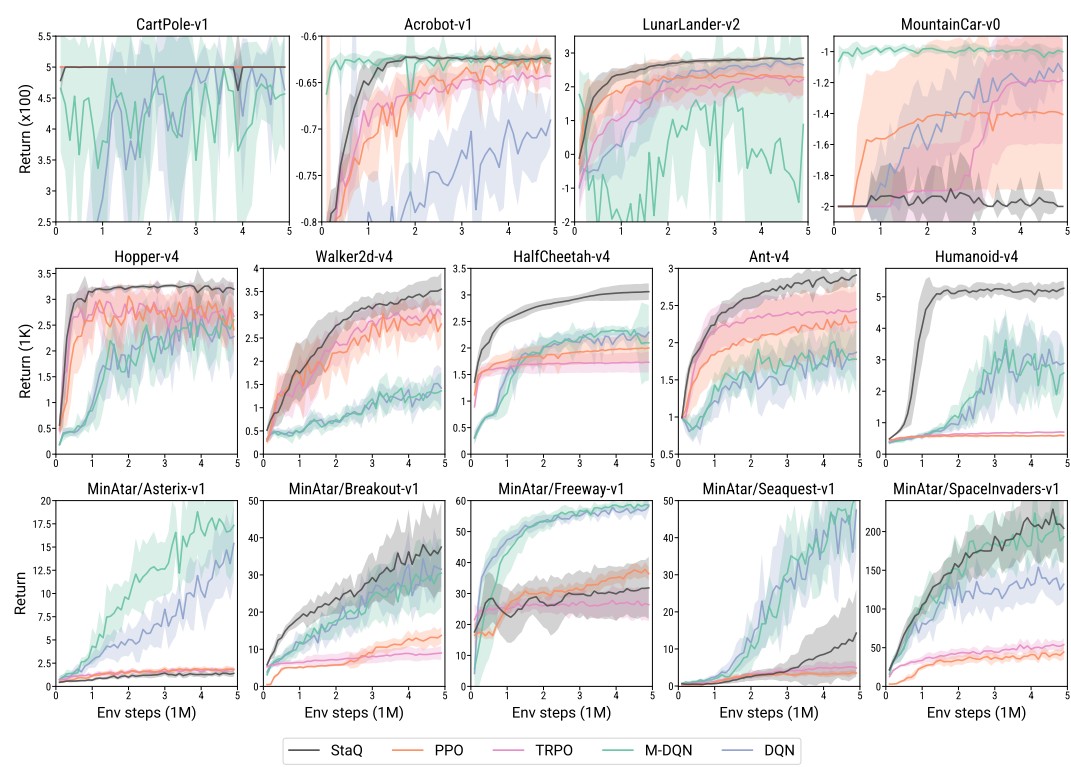

Figure 6: Policy performance across all environments.

|  | StaQ | PPO | TRPO | M-DQN | DQN |
|---|---|---|---|---|---|
| CartPole-v1 | **500** | **500** | **500** | 457 | 463 |
| Acrobot-v1 | **-62** | -63 | -64 | -63 | -69 |
| LunarLander-v2 | **285** | 227 | 222 | 88 | 265 |
| MountainCar-v0 | -200 | -141 | -118 | **-100** | -113 |
| Hopper-v4 | **3196** | 2411 | 2672 | 2600 | 2279 |
| Walker2d-v4 | **3550** | 2799 | 3010 | 1364 | 1424 |
| HalfCheetah-v4 | **3061** | 2001 | 1731 | 2098 | 2294 |
| Ant-v4 | **2910** | 2277 | 2452 | 1776 | 1871 |
| Humanoid-v4 | **5273** | 588 | 700 | 2580 | 2887 |
| MinAtar/Asterix-v1 | 1 | 2 | 2 | **17** | 15 |
| MinAtar/Breakout-v1 | **37** | 14 | 9 | 30 | 32 |
| MinAtar/Freeway-v1 | 32 | 36 | 26 | **59** | 58 |
| MinAtar/Seaquest-v1 | 14 | 3 | 5 | **51** | 47 |
| MinAtar/SpaceInvaders-v1 | **204** | 44 | 54 | 193 | 132 |

Table 2: Final performance on all environments.

## B.2    STABILITY PLOTS (VARIATION WITHIN INDIVIDUAL RUNS)

In this section we provide further stability plots to complement Fig. 3 (**Left**). In Fig. 7-8 we plot the returns of the first three seeds of the full results (shown in Fig. 6). At each timestep, the returns for

each individual seed are normalised by subtracting and then dividing by the mean across all seeds. We only include environments where all algorithms learn a decent policy to enable a fair comparison. We can see from Fig. 7-8 that Approximate Policy Iteration (API) algorithms (StaQ, TRPO, PPO) generally exhibit less variation within runs than Approximate Value Iteration (AVI) ones (DQN, M-DQN). In simple environments, such as CartPole, all three API algorithms have stable performance, but on higher dimensional tasks, only StaQ retains a similar level of stability while maintaining good performance. This is especially striking on Hopper, where runs show comparatively little variation within iterations while having the highest average performance, as shown in Fig. 6. We attribute this improved stability in the performance of the evaluation policy by the averaging over a very large number of Q-functions ($M = 300$) of StaQ, which reduces the infamous performance oscillation of deep RL algorithms in many cases.

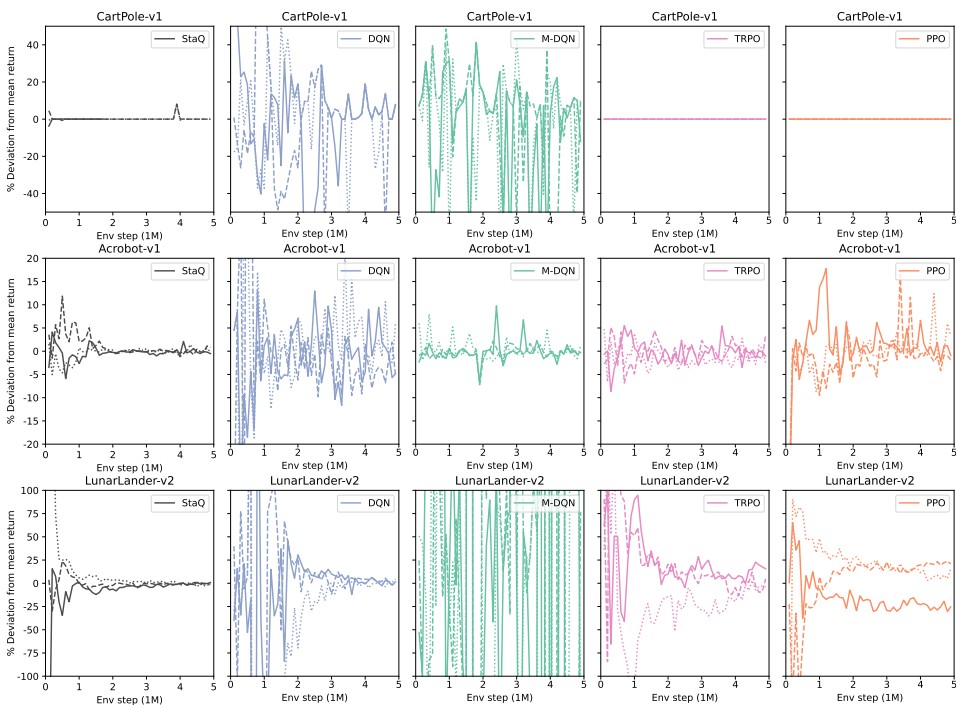

Figure 7: Stability plots showing normalized performance of the first three individual runs for each algorithm. See text for more details. Figures continue on the next page.

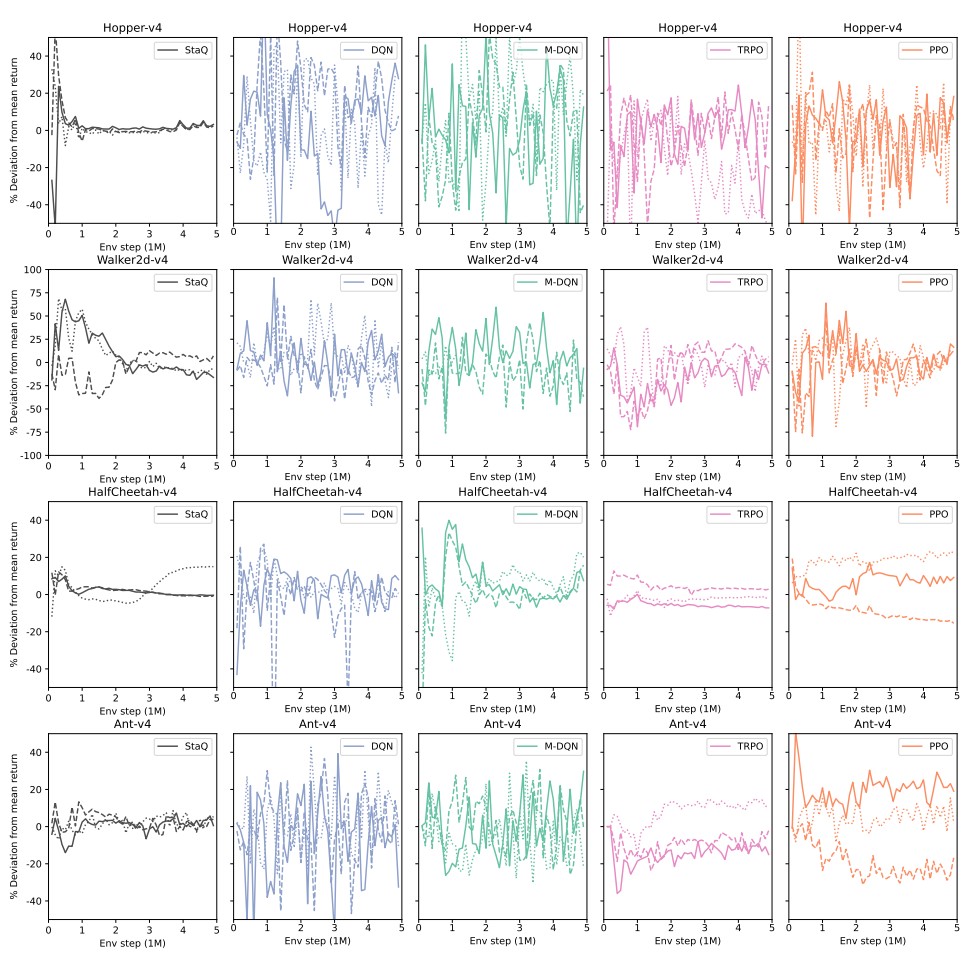

Figure 8: Stability plots showing normalized performance of the first three individual runs for each algorithm. See text for more details.

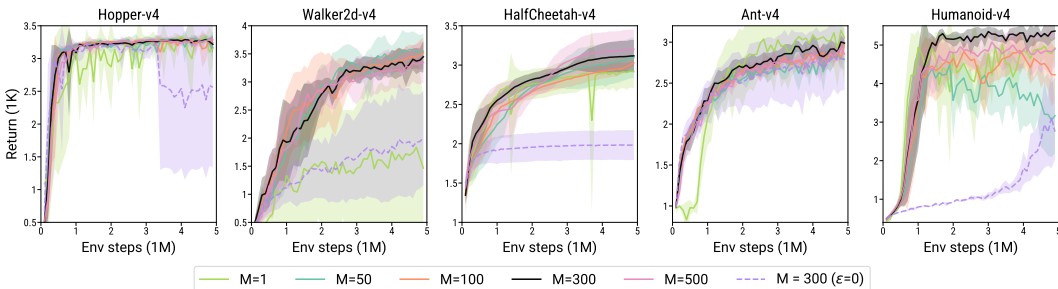

Figure 9: Ablation study for different memory sizes $M$ for $\epsilon = 0$, on all MuJoCo environments. Results showing then mean and one standard deviation averaged over 5 seeds.

### B.3 THE IMPACT OF THE MEMORY-SIZE M AND VALUE OF $\epsilon$

Figure 9 shows the performance of StaQ for different choices of $M$ and for the hyper-parameter $\epsilon = 0$ instead of $\epsilon = 0.05$ on additional MuJoCo tasks. Setting $M = 1$ corresponds to no KL-regularization as discussed in App. D and can be seen as an adaptation of SAC to discrete action spaces. $M = 1$ is unstable on both Hopper and Walker, in addition to Acrobot as shown in Fig. 3 in the main paper. Adding KL-regularization and averaging over at least 50 Q-functions greatly helps to stabilize performance except on the Humanoid task, as shown in Fig. 3, where $M = 50$ was still unstable compared to $M = 300$. Finally, the default setting of $\epsilon = 0.05$ outperforms a pure softmax policy with $\epsilon = 0$ on all but the Ant environment. We discuss some of the likely reasons for the need of $\epsilon$-softmax exploration in App. B.4.

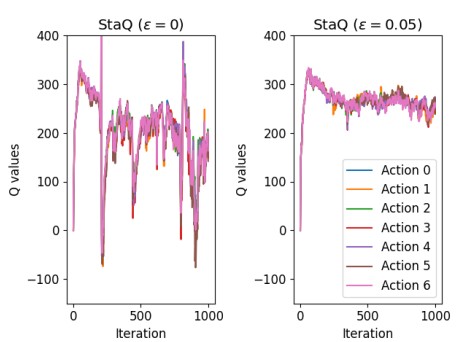 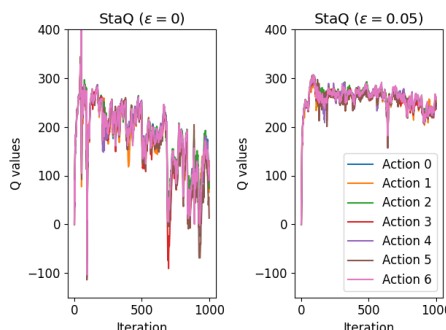

(a) Q-functions recorded at a given state across 1000 iterations for StaQ's behavior policy hyper-parameter $\epsilon = 0$ and $\epsilon = 0.05$. Seed 0.

(b) Q-functions recorded at a given state across 1000 iterations for StaQ's behavior policy hyper-parameter $\epsilon = 0$ and $\epsilon = 0.05$. Seed 1.

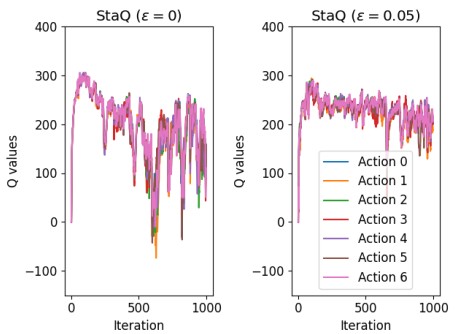 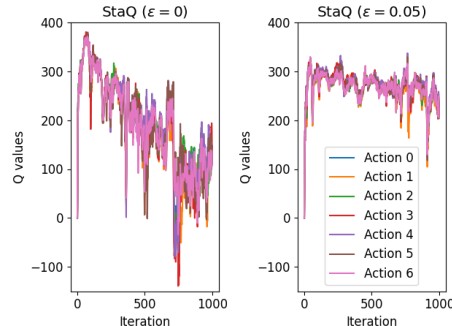

(c) Q-functions recorded at a given state across 1000 iterations for StaQ's behavior policy hyper-parameter $\epsilon = 0$ and $\epsilon = 0.05$. Seed 2.

(d) Q-functions recorded at a given state across 1000 iterations for StaQ's behavior policy hyper-parameter $\epsilon = 0$ and $\epsilon = 0.05$. Seed 3.

Figure 10: Q-values at some fixed states across 1000 iterations of StaQ, using an $\epsilon$-softmax behavior policy to collect data in the replay, with $\epsilon = 0.05$ or $\epsilon = 0$. With $\epsilon = 0$, we noticed very large variations in the Q-function between iterations that are reduced when using $\epsilon = 0.05$.

### B.4 CATASTROPHIC FORGETTING IN THE Q-FUNCTION

To understand why adding an $\epsilon$-softmax policy on top of the softmax policy $\pi_k$ stabilizes performance on Hopper-v4 as shown in Fig. 4, we have conducted the following experiment. We first launched two runs of StaQ with an $\epsilon$-softmax policy on top of $\pi_k$, with $\epsilon$ being either $0.05$ or $0$. From these two runs, we collected 100 states spread along both training processes. We then launched 5 independent runs for each value of $\epsilon$, and recorded for these 100 states the learned Q-values at each iteration. Upon manual inspection of the Q-values, we immediately notice that when $\epsilon = 0$, the Q-values vary more wildly across time for all the actions. Fig. 10 shows a few examples for four different seeds. To understand whether these variations have any tangible impact on the instability of the policy, we have performed the following test: we compute the logits $\xi_k$ at every iteration following the EPMD formula (Eq. 7) and rank the actions according to $\xi_k$. At each iteration $k$, we then compute the proportion of states, out of 100 reference states, in which an action has both the highest and the lowest rank in the next iterations $k' \geq k$. The results are shown in Fig. 11, where we can see that when $\epsilon = 0$, the fraction of states in which an action is considered as either being the best or the worst remains higher than when $\epsilon = 0.05$, which might result in performance drops across iterations. Thus the observed Q-function oscillations that appear more pronounced for $\epsilon = 0$ have a quantifiable impact on the stability of the policy, resulting in more states seeing actions switching from best to worst or vice versa.

It is hard to know exactly what causes the Q-values to oscillate more when $\epsilon = 0$. On the one hand, as these instabilities generally happen after the policy reached its peak performance, they could be because of some actions having very low probability of being selected in some states thus becoming

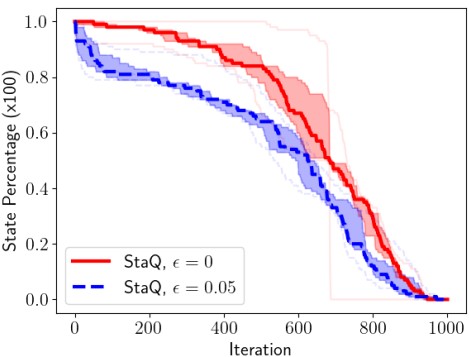

Figure 11: The percentage of states (out of 100 states) in which from iteration $k$ and onward, an action was considered both the best and the worst according to $\xi_k$ of EPMD. The difference of stability in the Q-values between $\epsilon = 0.05$ and $\epsilon = 0$ noted in Fig. 10 causes a difference in stability of policies, where actions switch more frequently from being worst to best when $\epsilon = 0$. The comparison is performed over 5 seeds showing the median and interquartile range.

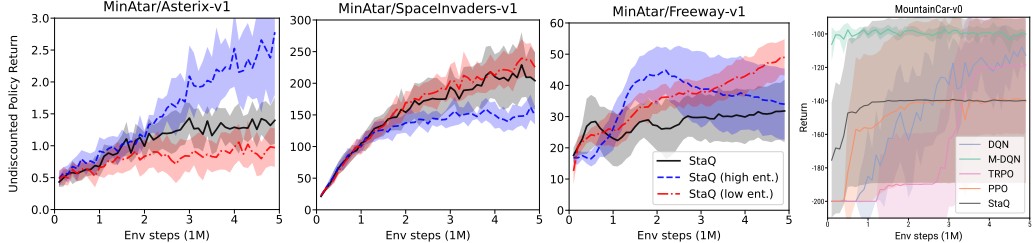

Figure 12: **Left three panels:** Effect of different entropy weights on the MinAtar environments. **Right panel:** Policy returns for StaQ (with additional tweaks to StaQ , see text for details) and deep RL baselines on MountainCar.

under-represented in the replay buffer $\mathcal{D}_k$. Setting $\epsilon > 0$ ensures that all actions have a non-zero probability of being sampled at any given state. On the other hand, due to the convexity[3] of $D_{\mathrm{KL}}$, i.e. $D_{\mathrm{KL}}\left((1-\epsilon)\pi + \epsilon p, (1-\epsilon)\pi' + \epsilon p'\right) \leq (1-\epsilon)D_{\mathrm{KL}}(\pi, \pi') + \epsilon D_{\mathrm{KL}}(p, p')$, if $\pi_k^b$ is an $\epsilon$-softmax strategy of $\pi_k$, then $D_{\mathrm{KL}}(\pi_k^b, \pi_{k+1}^b) \leq (1-\epsilon)D_{\mathrm{KL}}(\pi_k, \pi_{k+1})$ for any $\epsilon > 0$. This implies that successive replay buffers should be more similar when $\epsilon > 0$, which stabilizes the learning due to smoother transfer from $Q_\tau^k$ to $Q_\tau^{k+1}$. Nonetheless, a case of $\epsilon > 0$ is not without its own challenges as discussed in Sec. 6.3, and we can see in Fig. 10 that it still exhibits sudden changes in the Q-function which could harm stability. While the averaging over past Q-functions of an EPMD policy can stabilize learning to some extent, we believe that the catastrophic forgetting in the Q-function itself should be addressed to fully stabilize deep RL.

### B.5 Entropy regularization does not solve exploration in RL

Entropy regularization is a common technique to improve exploration and robustness of the policy (see e.g. Ziebart (2018); Haarnoja et al. (2017)), however overly-strong regularization can harm learning. In our main experimental results we fix the initial and final (scaled) entropy weight $\bar{\tau}$ across all environments, however we find that certain environments benefit from adjusting the entropy weight. This is illustrated in the left three panels of Fig. 12, where we vary the entropy weights, linearly annealed from 5 to 1 (1 to 0.05) for "high ent" ("low ent") over the first 1 million timesteps. In addition to environments such as Asterix and SpaceInvaders that absolutely prefer high or low entropy, some environments such as Freeway have complex dependence on the entropy that

---

[3]See e.g. https://statproofbook.github.io/P/kl-conv.html for the proof.

suggests that a constant or linearly-decaying entropy is insufficient. Future work could find best $\bar{\tau}$ automatically, for example as in SAC (Haarnoja et al., 2018).

For MountainCar, we find that having overly stable updates at the start of the training can prevent it from finding the sparse reward signal. StaQ can learn on such an environment (see right panel of Fig. 12), without tuning the entropy or $D_{\mathrm{KL}}$ weights, but with additional tweaks, such as increasing a scale of NN weights upon initialization (by $\times 20$), decreasing the policy update interval (from 5000 to 500 steps) and using an $\epsilon$-greedy strategy on top of $\xi_k$ as a behavior policy, rather than an $\epsilon$-softmax policy used for the main experiments. The $\epsilon$-greedy strategy is also used in DQN and M-DQN that outperform other baselines on MountainCar. Note that even though StaQ still performs worse than most other baselines, this is because some runs do not see a reward signal early enough in training and the inherent stability of StaQ prevents them from seeing it after the initial exploration period is over.

**Policy evaluation and exploration.** We find that using the `min` of the two Q-functions to compute the target values results in more stable training. `min` gives a more conservative target that is robust to overestimation bias in the Q-functions, and this allows us to reduce the KL weight. However, such a strategy struggles with sparse rewards, and hard exploration problems such as MountainCar-v0 and MinAtar/Seaquest-v1. Therefore we instead use the `mean` in Classic/MinAtar environments. Future work could use a more sophisticated approach that is both robust to overestimation bias and yet sensitive to weak reward signals.

## C TRAINING AND INFERENCE TIME COMPARISONS

| | Memory size $M$ | 1 | 50 | 100 | 300 | 500 |
|---|---|---|---|---|---|---|
| Hopper-v4 | Training time (hrs) | 9.8 | 10.1 | 10.3 | 10.3 | 10.9 |
| | Inference speed (steps/s) | 610 | 610 | 620 | 640 | 600 |
| Ant-v4 | Training time (hrs) | 10.4 | 10.7 | 10.3 | 11 | 10.5 |
| | Inference speed (steps/s) | 540 | 570 | 560 | 540 | 560 |

Table 3: Training and inference times for StaQ, as a function of $M$, on Hopper-v4 (state dim.=11) and Ant-v4 (state dim. = 105), computed on an NVIDIA Tesla V100 and averaged over 3 seeds.

| | | StaQ | PPO | TRPO | M-DQN | DQN |
|---|---|---|---|---|---|---|
| Hopper-v4 | Training time (hrs) | 10.3 | 3.7 | 3.2 | 5.6 | 4.9 |
| | Inference speed (steps/s) | 640 | 1040 | 1020 | 1550 | 1460 |
| Ant-v4 | Training time (hrs) | 11 | 4.3 | 3.6 | 6.1 | 5.3 |
| | Inference speed (steps/s) | 540 | 830 | 850 | 1110 | 1040 |

Table 4: Training and inference times for StaQ ($M = 300$) vs baselines, on the Hopper-v4 and Ant-v4 environments. Timings are computed on an NVIDIA Tesla V100, averaged over 3 seeds.

In this section, we report the training time and inference speed of StaQ, as a function of memory size $M$ and state space dimension. We also compare it to the deep RL baselines. All timing experiments were computed on an NVIDIA Tesla V100, and averaged over 3 seeds. The training time is defined as the time required to train StaQ for 5 million timesteps, while the inference speed is measured by the number of environment steps per second that can be evaluated during inference. Table 3 shows that memory size and dimension of the state space have a negligible impact on training and inference times, as discussed in Sec 6. Table 4 compares the training and inference time of StaQ ($M = 300$) with the baselines.

# D COMPARISON WITH SOFT ACTOR-CRITIC

In this appendix, we explain why Soft Actor-Critic (SAC, Haarnoja et al. (2018)) was not used as a baseline and how SAC relates to M-DQN and StaQ with $M = 1$. SAC is not used as a baseline because StaQ is a discrete action algorithm evaluated on discrete action environments, while SAC is not compatible with discrete action spaces. However, M-DQN can be seen as an adaptation of SAC to discrete action spaces with an additional KL-divergence regularizer. Please see the discussion in Vieillard et al. (2020) on page 3, between Eq. (1) and (2). Vieillard et al. (2020) also describe Soft-DQN in Eq. (1) as a straightforward discrete-action version of SAC, that can be obtained from M-DQN by simply setting the KL-divergence regularization weight to zero. Soft-DQN was not included as a baseline because the results of Vieillard et al. (2020) suggest that M-DQN generally outperforms Soft-DQN.

We also note that by setting $M = 1$ in StaQ, we remove the KL-divergence regularization and only keep the entropy bonus. This baseline can also be seen as an adaptation of SAC to discrete action spaces: indeed, if we set $M = 1$ in Eq. (13) we recover the policy logits

$$\xi_{k+1} = \frac{\alpha}{1 - \beta^M} \sum_{i=0}^{M-1} \beta^i Q_\tau^{k-i}$$

$$= \frac{\alpha}{1 - \beta} Q_\tau^k$$

$$= \frac{Q_\tau^k}{\tau},$$

where the last line is due to $\alpha\tau = 1 - \beta$. This results in a policy of the form $\pi_{k+1} \propto \exp\left(\frac{Q_\tau^k}{\tau}\right)$. Meanwhile, for SAC, the actor network is obtained by minimizing the following problem (Eq. 14 in Haarnoja et al. (2018))

$$\pi_{k+1} = \arg\min \text{KL}\left(\pi \middle| \frac{\exp\left(\frac{Q_\tau^k}{\tau}\right)}{Z_{\text{norm.}}}\right).$$

However, in the discrete action setting, we can sample directly from $\exp\left(\frac{Q_\tau^k}{\tau}\right)$—which is the minimizer of the above KL-divergence term—and we do not need an explicit actor network. As such StaQ with $M = 1$ could be seen as an adaptation of SAC to discrete action spaces.

# E CONTINUAL LEARNING VIEW OF POLICY UPDATE

In Continual RL (Lesort et al., 2020), a learner is presented with a sequence of MDPs and a one evaluates whether the learner is able learn on the new MDPs while retaining information of older ones. In our setting, a learner sees only one MDP and its performance is only measured on this one MDP. Studying if knowledge is retained on older tasks or whether knowledge transfers among tasks is beyond the scope of this paper.

Despite being limited to a single MDP, single-task RL has still strong ties with CL because of the sequential nature in which data arrives. As mentioned in Sec. 1, we are not the first to draw this parallel, but drawing this connection is interesting because it opens up a plethora of CL methods that are not well researched in the deep RL context, but are applicable even in a single task setting. Specifically, in this paper we focus on the entropy regularized policy update problem described below (Eq. 5 of the paper)

$$\text{for all } s \in S, \pi_{k+1}(s) = \arg\max_{p \in \Delta(A)} \left\{ Q_\tau^k(s) \cdot p - \tau h(p) - \eta D_{\text{KL}}(p; \pi_k(s)) \right\}.$$

The objective of this update can be seen as CL, as we receive a new "task" which is to find $p$ a maximum entropy distribution over actions that puts its largest mass on actions with high Q-values, yet, through the KL-divergence term above, we do not want to differ too much from $\pi_k$, and forget the solution of the previous "task". Because of this similarity with CL, existing methods to solve this problem can be categorized with the CL literature lens, for example: Lazic et al. (2021)

used a rehearsal method (replay buffer/experience replay in deep RL terminology) to tackle the above policy update, while Schulman et al. (2015) uses a parameter regularization approach. These methods cover two of the three main classes of CL methods De Lange et al. (2021), and the novelty of this paper is in investigating a method pertaining to the third class (parameter isolation) to tackle this problem, as this class of methods has strong performance in CL benchmarks (See Sec. 6 of De Lange et al. (2021)), yet remains largely understudied in deep RL.

## F  HYPERPARAMETERS

Here, we provide the full list of hyperparameters used in our experiments. StaQ's hyperparameters are listed in Table 5, while the hyperparameters for our baselines are provided in Tables 6-8. In all environments, we enforce a time limit of 5000 steps. This is particularly useful for Seaquest-v1, since an agent can get stuck performing an infinitely long rollout during data collection. Furthermore, to account for different scales of reward signals in different environments, we apply different reward scales between Classic/MuJoCo environments and MinAtar. Note that this is equivalent to inverse-scaling the entropy weight $\tau$ and KL weight $\eta$, ensuring that $\xi_k$ is of the same order of magnitude for all environments.

| Hyperparameter | Classic | MuJoCo | MinAtar |
|---|---|---|---|
| Discount ($\gamma$) | 0.99 | 0.99 | 0.99 |
| Memory size ($M$) | 300 | 300 | 300 |
| Policy update interval | 5000 | 5000 | 5000 |
| Ensembling mode | **mean** | **min** | **mean** |
| Target type | hard | hard | hard |
| Target update interval | 200 | 200 | 200 |
| Epsilon | 0.05 | 0.05 | 0.05 |
| Reward scale | **10** | **10**[*] | **200** |
| KL weight ($\eta$) | **20** | **10** | **20** |
| Initial scaled ent. weight | 2.0 | 2.0 | 2.0 |
| Final scaled ent. weight | 0.4 | 0.4 | 0.4 |
| Decay steps | **500K** | **1M** | **1M** |
| Architecture | $256 \times 2$ | $256 \times 2$ | $256 \times 2$ |
| Activation function | ReLu | ReLu | ReLu |
| Learning rate | 0.0001 | 0.0001 | 0.0001 |
| Replay capacity | 50K | 50K | 50K |
| Batch size | 256 | 256 | 256 |

Table 5: StaQ hyperparameters, with parameters which vary across environment types in bold. [*]Hopper-V4 uses a reward scale of 1.

| Hyperparameter | Classic | MuJoCo | MinAtar |
|---|---|---|---|
| Discount factor ($\gamma$) | 0.99 | 0.99 | 0.99 |
| Horizon | 2048 | 2048 | 1024 |
| Num. epochs | 10 | 10 | 3 |
| Learning starts | 5000 | 20000 | 20000 |
| GAE parameter | 0.95 | 0.95 | 0.95 |
| VF coefficient | 0.5 | 0.5 | 1 |
| Entropy coefficient | 0 | 0 | 0.01 |
| Clipping parameter | 0.2 | 0.2 | $0.1 \times \alpha$ |
| Architecture | $64 \times 2$ | $64 \times 2$ * | $256 \times 2$ |
| Activation function | Tanh | Tanh | Tanh |
| Learning rate | $3 \times 10^{-4}$ | $3 \times 10^{-4}$ | $2.5 \times 10^{-4} \times \alpha$ |
| Replay capacity | 1M | 1M | 50K |
| Batch size | 64 | 64 | 32 |

Table 6: PPO hyperparameters, based on (Schulman et al., 2017). In the MinAtar environments $\alpha$ is linearly annealed from 1 to 0 over the course of learning. *Humanoid-v4 uses a hidden layer size of 256.

| Hyperparameter | Classic | MuJoCo | MinAtar |
|---|---|---|---|
| Discount factor ($\gamma$) | 0.99 | 0.99 | 0.99 |
| Horizon | 2048 | 2048 | 2048 |
| Learning starts | 5000 | 20000 | 20000 |
| GAE parameter | 0.95 | 0.95 | 0.95 |
| Stepsize | 0.01 | 0.01 | 0.01 |
| Architecture | $64 \times 2$ | $64 \times 2$ * | $256 \times 2$ |
| Activation function | Tanh | Tanh | Tanh |
| Learning rate | $3 \times 10^{-4}$ | $3 \times 10^{-4}$ | $3 \times 10^{-4}$ |
| Replay capacity | 1M | 1M | 1M |
| Batch size | 64 | 64 | 64 |

Table 7: TRPO hyperparameters, based on (Schulman et al., 2015). *Humanoid-v4 uses a hidden layer size of 256.

| Hyperparameter | Classic | MuJoCo | MinAtar |
|---|---|---|---|
| Discount factor ($\gamma$) | 0.99 | 0.99 | 0.99 |
| Target update interval | 100 | 8000 | 8000 |
| Epsilon | 0.1 | 0.1 | 0.1 |
| Decay steps | 20K | 20K | 20K |
| M-DQN temperature | 0.03 | 0.03 | 0.03 |
| M-DQN scaling term | 1.0 | 0.9 | 0.9 |
| M-DQN clipping value | -1 | -1 | -1 |
| Architecture | $512 \times 2$ | $128 \times 2$ | $128 \times 2$ |
| Activation function | ReLu | ReLu | ReLu |
| Learning rate | $1 \times 10^{-3}$ | $5 \times 10^{-5}$ | $5 \times 10^{-5}$ |
| Replay capacity | 50K | 1M | 1M |
| Batch size | 128 | 32 | 32 |

Table 8: MDQN and DQN hyperparameters, based on (Vieillard et al., 2020; Ceron & Castro, 2021)

# G   PSEUDOCODE OF STAQ

We provide in this section the pseudocode of StaQ in Alg. 1. As an approximate policy itera-
tion algorithm, StaQ comprises three main steps: i) data collection, ii) policy evaluation iii) policy
improvement. Data collection (Line 4-5) consist in interacting with the environment to collect tran-
sitions of type (state, action, reward, next state) that are stored in a replay buffer. A policy evaluation
algorithm is then called to evaluation the current Q-function $Q_\tau^k$ using the replay buffer. Finally, the
policy update is optimization-free and simply consists in stacking the Q-function in the SNN policy
as discussed in Sec. 5. After $K$ iterations, the last policy is returned.

---

**Algorithm 1** StaQ (Finite-memory entropy regularized policy mirror descent)

---

1: **Input:** An MDP $\mathcal{M}$, a memory-size $M$, Number of samples per iteration $N$, Replay buffer size
   $D$, Initial behavior policy $\pi_0^b$, entropy weight $\tau$, $\epsilon$-softmax exploration parameter
2: **Output:** Policy $\pi_K \propto \exp(\xi_K)$
3: **for** $k = 0$ **to** $K - 1$ **do**
4:    Interact with $\mathcal{M}$ using the behavior policy $\pi_k^b$ for $N$ times steps
5:    Update replay buffer $\mathcal{D}_k$ to contain the last $D$ transitions
6:    Learn $Q_\tau^k$ from $\mathcal{D}_k$ using a policy evaluation algorithm
7:    Obtain logits $\xi_{k+1}$ by stacking the last $M$ Q-functions (see Sec. 5) following the finite-
      memory EPMD update of Eq. 12.
8:    Set $\pi_{k+1} \propto \exp(\xi_{k+1})$ and $\pi_{k+1}^b$ to an $\epsilon$-softmax policy over $\pi_{k+1}$
9: **end for**

---

