# OpenReview forum: "A Continual Learning Perspective to Entropy Regularized Deep Reinforcement Learning"
_ICLR.cc/2025/Conference — Submitted to ICLR 2025_

### Official Review · Reviewer_b4CY · 2024-10-23

**Soundness:** 3
**Presentation:** 3
**Contribution:** 2
**Rating:** 5
**Confidence:** 5

**Summary:**

This paper introduces StaQ, an algorithm based on Policy Mirror Descent that uses a growing neural architecture to retain Past Q-functions. By limiting memory to the last M Q-functions, StaQ balances between scalability and stability. The authors experiments are evaluated on Classic Control, Mujoco, and the Minatar environment. StaQ shows strong performance in the Mujoco domain, and is compared against PPO, TRPO, DQN and Munchausen DQN.

**Strengths:**

- Well written.

- Extensive Related Work.

- Interesting approach of stacking neural networks.

**Weaknesses:**

- Evaluation: The classic control environments are, in my opinion, not informative when comparing RL algorithms. The same holds for the Minatar environments. The only baseline that is representative for RL algorithms in this paper therefore are the Mujoco environments. I would advise the authors to refrain from using Minatar and Classic control in their submissions.

- In line 393-396, you draw comparisons between your algorithms value functions and computational efficiency with those of SAC. However, what is the reason you do not compare to SAC in your evaluations, but instead to TRPO and PPO?

- Figure 3a:  *"Individual runs of StaQ show a clearer trend (2 runs plateau and one continues increasing),
while M-DQN, the closest performing baseline oscillates more around zero"*. I do not understand the reason why the authors would show this figure, as well as the caption. How is two runs plateauing and one increasing superior over another oscillating around zero? The first shows a 33% success rate while the second is oscillating a little more. I believe this figure and the connected conclusions around it are not contributing to the paper. What was the intended idea of this figure?

- Ablations: In the main paper, only Fig. 3b and Fig. 4 show some ablations of which 1 represents ablations on Acrobot. It would be much more informative if the authors would run a variety of ablations on all of the Mujoco environments, and normalize them into 1 score. As said in my first point, results and ablations on Acrobot are not informative to the reader.

**Questions:**

-

---

> ### Author Response · Authors · 2024-11-19
> **Response 1/2**
>
> Dear Reviewer b4CY,
>
> Thank you for your valuable feedback. We provide a preliminary answer to your comments below, and will complement our answer as soon as we have the additional ablation experiments you requested.
>
> **Q1. Relevance of the tasks**
>
> **A1.** In your review, you claim that
> >The classic control environments are, in my opinion, not informative when comparing RL algorithms. The same holds for the Minatar environments.
>
> In our paper, we cite the prior work [Ceron et al. 2021](https://arxiv.org/pdf/2011.14826) whose claims appear to be in contradiction to your statement above. They show that the Classic Control + MinAtar environments are sufficiently rich to reach comparable conclusions when comparing RL algorithms as with a more complex testbed. We would appreciate if you could clarify whether you disagree with the claims of [Ceron et al. 2021](https://arxiv.org/pdf/2011.14826) and your reasoning for this.
>
> ---
>
> **Q2. Lacking comparison with SAC**
>
> **A2.** Reiterating our answer to reviewer U8vT, we would like first of all to remind the reviewer that StaQ is only compatible with discrete action spaces and hence we only use discrete-action deep RL baselines. In the paper, we consider four baselines: TRPO, PPO, DQN and Munchausen DQN (M-DQN).
>
> Soft Actor Critic (SAC) is not compatible with discrete action spaces, but M-DQN can be seen as an adaptation of SAC to discrete action spaces with an additional KL-divergence regularizer. Please see the discussion in the M-DQN paper ([Vieillard et al. 2020](https://proceedings.neurips.cc/paper/2020/file/2c6a0bae0f071cbbf0bb3d5b11d90a82-Paper.pdf)) on page 3, between Eq. (1) and (2). The M-DQN paper also discusses Soft-DQN in Eq. (1), which they call the
> "most straightforward discrete-actions version of Soft Actor-Critic (SAC)", and which can be obtained from M-DQN by simply setting the KL-divergence regularization weight to zero. However, we did not consider Soft-DQN as a baseline because the M-DQN paper shows that M-DQN generally outperforms Soft-DQN.
>
> We also note that by setting $M=1$ in StaQ, we remove the KL-divergence regularization and only keep the entropy bonus. This baseline can also be seen as an adaptation of SAC to discrete action spaces: indeed, if we set $M=1$ in Eq. (14) we recover the policy logits
> $\xi_{k+1} = \frac{\alpha}{1-\beta^M}\sum_{i=0}^{M-1}\beta^i Q_{\tau}^{k-i} = \frac{\alpha}{1-\beta}Q_{\tau}^{k} = \frac{Q_{\tau}^{k}}{\tau}$ (because $\alpha \tau = 1-\beta$), such that $\pi_{k+1} \propto \exp\left(\frac{Q_{\tau}^k}{\tau}\right)$. While for SAC, the actor network is obtained by minimizing this problem (Eq. 14 in [Haarnoja et al. 2019](https://arxiv.org/pdf/1812.05905)) $\pi_{k+1} = \arg\min \text{KL}\left(\pi \Big| \frac{\exp\left(\frac{Q_{\tau}^k}{\tau}\right)}{Z_{\text{norm.}}}\right)$, but in the discrete action setting, we can sample directly from $\exp\left(\frac{Q_{\tau}^k}{\tau}\right)$---which is the minimizer of the above KL-divergence term---and we do not need an explicit actor network. As such StaQ with $M=1$ could be seen as an adaptation of SAC to discrete action spaces. Please note that we have already included $M=1$ in the ablation study for some environments.
>
> In summary, because M-DQN outperforms Soft-DQN and because StaQ with $M=1$ resembles SAC (as it uses only soft Q-functions and no KL-divergence regularization), we do not believe that Soft-DQN would be useful as an additional baseline, however we would be open to feedback on this.
>
> ---
>
> **Q3. Intention behind Fig. 3a**
>
> **A3.** The main claim of this paper is that we are able to implement a Policy Mirror Descent like algorithm with strong theoretical guarantees. The theory guarantees monotonic improvements and therefore we wanted to see if in practice we have more stable monotonically increasing performance compared to baselines. For many tasks, this can be observed at a glance, as StaQ runs demonstrate the lower variance in the returns. However, in tasks such as SpaceInvaders in Fig.2, we can see a comparable variance between StaQ and M-DQN. Fig. 3a zooms into the performance of StaQ and M-DQN on SpaceInvaders by plotting the centered returns of the individual runs. It shows that the variance in StaQ is not caused by large performance oscillation within runs, and in fact we still see the same smooth return curves as observed on other environments in Fig. 7; but simply the variance is due to one seed of StaQ that was more succesful than the others, likely due to better exploration, which is a component beyond the scope of this paper. We also note that the other two seeds did not fail in this plot, and in fact the average performance of StaQ on SpaceInvaders is slightly higher than that of M-DQN. It just so happens that one seed performed better than all other seeds, including the runs of M-DQN.

---

> > ### Comment · Reviewer_b4CY · 2024-11-22
> > **Reponse to Authors**
> >
> > Q1:
> >
> > From what I have seen, the author of the paper that is mentioned has not used any of the Classic Control or Minatar environments in any of his Reinforcement Learning papers following the claims in [Ceron et al, 2021](https://arxiv.org/pdf/2011.14826). Coming back to your quote:
> >
> > *They show that the Classic Control + MinAtar environments are sufficiently rich to reach comparable conclusions when comparing RL algorithms as with a more complex testbed*
> >
> > Then I don't see any reason as to why the whole RL community would not use these environments.
> >
> > Furthermore, this manuscript itself validates my observation. If you look at Fig.6: The Classic Control environment scores are so inconsistent for any algorithm that a reader couldn't possibly create an expectation of the algorithm's overall performance. The same holds for the Minatar environment. Looking at the Minatar environment, the reader might argue that vanilla DQN outperforms StaQ. So please answer this question: If the authors believe that the Classic Control and Minatar environments are sufficient to reach comparable decisions, and the authors would have left the Mujoco benchmark out of the paper, then what would the manuscript's conclusion be?
> >
> > However, as I take the Mujoco tasks more seriously, I still believe the StaQ algorithm to probably be stronger than the baselines.
> >
> > Q2:
> >
> > Fair enough, I see that as a good reason. It might be good to mention this more prominently in the paper to avoid these questions from other readers.
> >
> > Q3:
> >
> > Although I see the point, it is still very confusing. One could even argue that M-DQN runs perform closer to the mean, albeit in a more intra-run oscillating matter. Why would the variance in the StaQ runs be explained by 'likely due to better exploration', while M-DQN does not seem to have these runs with better or worse exploration? As I have said before, I feel like this figure does not offer a contribution in its current form, and is confusing to the reader.

---

> ### Author Response · Authors · 2024-11-19
> **Response 2/2**
>
> **Q4. Ablation on more tasks**
>
> **A4.** We are currently running the ablation regarding the choice of $M$ on all Mujoco tasks. We thank you for your suggestion and will complement our answer as soon as we have more results.

---

> ### Author Response · Authors · 2024-11-22
>
> Dear Reviewer,
>
> Thank you for your detailed response and for articulating so clearly your point of view. Even if we may not agree on all points, it at least leads to an interesting discussion. Please see below for our comments.
>
> ---
>
> **Q1: Relevance of the tasks and challenges of exploration**
>
> **A1.** We would like to point out that the authors of [Ceron et al. 2021](https://arxiv.org/abs/2011.14826) do in fact use the MinAtar environments in several subsequent publications, see for example [Willi et al. 2024](https://arxiv.org/pdf/2406.18420), [Araújo et al. 2021](https://openreview.net/pdf?id=3UK39iaaVpE) and [Ostrovski et al. 2021](https://proceedings.neurips.cc/paper/2021/file/c3e0c62ee91db8dc7382bde7419bb573-Paper.pdf). More generally, the Atari environments are a widely-used standard benchmark for discrete-action RL, and MinAtar is designed to replicate those environments except with a more compact feature representation and faster runtime.
>
> If we were to exclude the MuJoCo environments, we would still be able to see the same overall picture: that StaQ is capable of achieving performance comparable to or above the baselines, and with better stability, but it struggles with hard exploration problems. Indeed, out of the 9 Classic Control+Minatar environments, 4 of them are classified as hard exploration problems by [Ceron et al. 2021](https://arxiv.org/abs/2011.14826) (see page 8), and 3 out of the 4 problems where StaQ doesn't outperform baselines are on these hard exploration tasks. It seems that on these hard exploration problems, $\epsilon$-greedy exploration used by DQN and M-DQN is a better exploration heuristic than sampling from the stochastic policy. This is not true only for StaQ but also for PPO/TRPO that similarly use a stochastic policy and struggle on these tasks. We do not know exactly why is this the case, but perhaps using deterministic policies is more likely to achieve ''deep exploration'' as discussed in [Osband et al. 2016](https://arxiv.org/pdf/1602.04621). [Fortunato et al. 2018](https://arxiv.org/pdf/1706.10295) also show that exploration with deterministic policies generally works better than stochastic ones on several benchmarks.
>
> Nonetheless, StaQ can be combined with any behavior policy for exploration, but exploration is a challenge of its own that is beyond the scope of this paper. Since our focus is rather on the stability of deep RL, we added the 5 MuJoCo tasks as an additional testbed since they have dense rewards, simplifying exploration, and leaving the proper use of function approximators as the only challenge. We agree with you that the results on MuJoCo strengthen the claim of the paper, but in no way do they negate the results on other environments. They rather complement them since if we remove the 4 hard exploration problems from [Ceron et al. 2021](https://arxiv.org/abs/2011.14826), we are only left with 5 environments, which is perhaps too few.
>
>
> ---
>
> **Q2: Adding discussion on SAC baseline**
>
> **A2.** We agree on this and we have updated the manuscript to make the connection between M-DQN/Soft-DQN and StaQ more explicit. Please see the mention (red text) in the Baselines paragraph of Sec. 6 and the full discussion in Appendix D. Thank you for this feedback.
>
> ---
>
> **Q3: Intention behind Fig. 3a**
>
> **A3.** Regarding why M-DQN has better exploration, we refer to A1 above. Nevertheless, we agree that Fig. 3a may be confusing to the reader. We plan to update that figure to clarify this point. We will update you when the manuscript has been updated to incorporate your feedback.
>
> ---
>
> **Q4. Ablation on more tasks**
>
> **A4.** Following up from our previous response, we have now updated the manuscript to include ablation studies for a range $M$, and have also ablated the choice of $\epsilon=0.05$ on more MuJoCo tasks. Please see Fig. 9 in App. B.3.

---

> > ### Comment · Reviewer_b4CY · 2024-11-26
> >
> > Q1: To clarify, he does not use these environments in any of his own first-author RL papers. Furthermore, out of the 3 papers you cited, there is only one top-tier conference paper ([Ostrovski et al. 2021](https://proceedings.neurips.cc/paper/2021/file/c3e0c62ee91db8dc7382bde7419bb573-Paper.pdf)), which uses original Atari games for 200M frames in the evaluations. They only use the Cartpole environment for 1 ablation figure. I think this further proves my point.
> >
> > Q2: Thanks. This improves the paper.
> >
> > Q3: This removes the confusion in the figure. Thanks.
> >
> > Furthermore, I want to reply to the quote from the Authors:
> >
> > - "More generally, the Atari environments are a widely-used standard benchmark for discrete-action RL, and MinAtar is designed to replicate those environments except with a more compact feature representation and faster runtime."*
> >
> > I do not feel like this is a valid argument for switching Atari for MinAtar. Just because MinAtar was designed to resemble Atari, does not mean it comes close to it in terms of algorithm evaluation quality. The Atari domain is the primary evaluation used for testing discrete algorithms such as DQN and M-DQN.
> >
> > Furthermore, I see that the authors use the same arguments for the reviewer oaMK, where the authors also cite [Ceron et al, 2021](https://arxiv.org/abs/2011.14826) as the reason to not do additional experiments or use different environments. Citing this paper to refrain from using valid benchmarks is in my opinion not a strong argument.
> >
> > Lastly, I am still concerned that the authors keep advocating for the Classic Control environments as valid evaluation benchmarks for new algorithms in ICLR submissions.
> >
> > I therefore will keep my score unchanged. I will however, raise my confidence score.

---

> ### Author Response · Authors · 2024-11-25
>
> Dear Reviewer,
>
> Following up on **Q3**, we have now updated the manuscript to incorporate your feedback regarding the clarity of Fig. 3a. The related changes to the text is in green. We now plot the Hopper environment rather than SpaceInvaders, as we believe this more clearly demonstrates the improved stability of StaQ, and is a more representative example. Note that, as before, we perform the same stability comparisons for a range of Classic Control + MuJoCo environments in App. B.2.
>
> We hope that this addresses your concerns, and we welcome any further feedback on this.

---

> ### Author Response · Authors · 2024-11-29
>
> Dear Reviewer,
>
> We thank you again for following up on the discussion, and we are happy to see that all your questions except the choice of environments have found a positive resolution. Let us point out that the experimental part is only one aspect of our paper, and that our contributions comprise a theoretical aspect, that extends the analysis of approximate PMD to a more practical finite-memory setting. In the context of a theoretically grounded setting, we believe that evaluating our algorithm on Classic Control tasks is relevant. Please see the discussion below.
>
> ---
>
> **Validity of Classic Control benchmarks.** We believe it is insightful to confirm our theoretical results on tasks where exploration and policy evaluation are easier, in order to leave policy update as the only differentiating factor between StaQ and baselines. The Classic Control tasks fulfill this need and we see greatly improved performance and stability over baselines. Of course, if our empirical evaluation was limited to these smaller dimensional problems, one would naturally wonder how StaQ would scale to larger environments. Luckily, this is not the case in this paper and we also evaluate StaQ on larger testbeds, namely MuJoCo and MinAtar, comparing favourably with the baselines.
>
> ---
>
> **Feedback on the theoretical contributions.** Our exchange has only focused on the empirical merits of our contributions. While we are grateful for your feedback on this aspect, as it helped us improve the paper, we still hope that you have assessed the merits of our paper with all of our contributions in mind. In the present state however, your review does not mention in any capacity our theoretical results, neither in good or bad, and we would really appreciate your feedback on this aspect. If some aspects of our contributions, or their relation to existing literature are unclear, we would be happy to use the extended discussion period to clarify them.
>
> As a reminder, we have analyzed two entropy regularized Policy Mirror Descent (PMD) like algorithms that only keep in memory the last $M$ Q-functions. The exact entropy regularized PMD form of a policy at iteration $k$ is given  by
>
> $$\pi_k\propto \exp\left(\alpha \sum_{i=0}^k \beta^i Q^{k-i}\right)$$
> with $\beta < 1$ a discount factor. The intuition is that since $\beta^i$ vanishes for large $i$, deleting a very old Q-function will have little impact on the policy. However, in our theoretical analysis we show that naively deleting the last Q-function (Vanilla Finite-Memory EPMD, Sec. 4.1) results in an irreducible error that prevents convergence. A significant theoretical contribution of our paper is proving that there exists a modified policy update rule (Weight-corrected Finite-Memory EPMD, Sec 4.2) for which exact convergence is guaranteed, provided the memory size $M$ is sufficiently large. We also derive an expression of a minimal $M$ as of function of $\beta$ and the MDP's discount factor $\gamma$. We believe this is a strong contribution to the community, as it leads to practical algorithms with clear empirical benefits and a solid theoretical framework to build on, and we would be happy to hear your thoughts on this aspect of our work.

---

> > ### Comment · Reviewer_b4CY · 2024-12-02
> > **Final comment.**
> >
> > I would like to respond to some of the quotes made by the Authors. I also recommend the other reviewers to look at this.
> >
> > "*The Classic Control tasks fulfill this need and we see greatly improved performance and stability over baselines.*"
> >
> > - I have made a table, normalizing all scores according to the Normalization Formula in the Appendix below. The table below does, in no way, correlate with the Authors' claim of "*greatly improved performance*". On average, StaQ is the **worst** performer in the Classic Control benchmark.
> >
> > | Category        |     StaQ (Authors) |      PPO |     TRPO |    M-DQN |      DQN |
> > |:----------------|---------:|---------:|---------:|---------:|---------:|
> > | Classic Control | 0.718   | 0.824   | **0.866**    | 0.814   | 0.801   |
> >
> > "*Of course, if our empirical evaluation was limited to these smaller dimensional problems, one would naturally wonder how StaQ would scale to larger environments. Luckily, this is not the case in this paper and we also evaluate StaQ on larger testbeds, namely MuJoCo and MinAtar, comparing favourably with the baselines.*"
> >
> > - Again, the authors claim a "*favourable*" comparison with baselines. In Mujoco, yes. On the MinAtar testbed however, Vanilla DQN and M-DQN vastly outperform StaQ:
> >
> > | Category        |     StaQ (Authors) |      PPO |     TRPO |    M-DQN |      DQN |
> > |:----------------|---------:|---------:|---------:|---------:|---------:|
> > | MinAtar         | 0.498 | 0.246 | 0.210 | **0.858** | 0.774 |
> >
> >
> > Since the authors claim that MinAtar is a valid replacement for the Atari benchmark, but the StaQ algorithm significantly underperforms DQN and M-DQN on MinAtar, can we conclude that StaQ will also underperform DQN and M-DQN on the Atari benchmark? This, together with the persistent reluctance of the Authors to test on Atari leads me to the conclusion that the evaluation done on StaQ is not sufficient. I therefore cannot recommend it for acceptance. I am lowering my score.
> >
> > My score does not reflect my concerns over the theoretical contribution, but it reflects my concerns over the evaluation method. Furthermore, it also reflects my concerns about the Authors' repeated incorrect statements on performance.
> >
> > ### Appendix: Normalization Formula
> >
> > To normalize scores for each environment:
> >
> > \[
> > Normalized Score = (Value - y_min) / (y_max - y_min)
> > \]
> >
> > Where:
> > - **Value**: The raw score for an algorithm.
> > - **y_min**: The minimum score for the environment.
> > - **y_max**: The maximum score for the environment.
> >
> > **Special Cases:**
> > - For environments where a lower score is better (e.g., MountainCar-v0), the formula is adjusted by flipping the score:
> >
> > \[
> > Normalized Score = (- Value - y_min) / (y_max - y_min)
> > \]
> >
> > ---
> >
> > ### Normalization Ranges
> >
> > | Environment              | \( y_min \) | \( y_max \) |
> > |--------------------------|---------------------:|---------------------:|
> > | **CartPole-v1**          | 250                 | 500                 |
> > | **Acrobot-v1**           | -80                 | -60                 |
> > | **LunarLander-v2**       | -200                | 300                 |
> > | **MountainCar-v0**       | -200                | -100                |
> > | **MinAtar/Breakout**     | 0                   | 50                  |
> > | **MinAtar/Asterix**      | 0                   | 20                  |
> > | **MinAtar/Freeway**      | 0                   | 60                  |
> > | **MinAtar/Seaquest**     | 0                   | 50                  |
> > | **MinAtar/SpaceInvaders**| 0                   | 230                 |
> >
> > ### Author Scores:
> > | Environment                 | StaQ |  PPO | TRPO | M-DQN |  DQN |
> > |-----------------------------|------|------|------|-------|------|
> > | **CartPole-v1**             |  500 |  500 |  500 |   457 |  463 |
> > | **Acrobot-v1**              |  -62 |  -63 |  -64 |   -63 |  -69 |
> > | **LunarLander-v2**          |  285 |  227 |  222 |    88 |  265 |
> > | **MountainCar-v0**          | -200 | -141 | -118 |  -100 | -113 |
> > | **Hopper-v4**               | 3196 | 2411 | 2672 |  2600 | 2279 |
> > | **Walker2d-v4**             | 3550 | 2799 | 3010 |  1364 | 1424 |
> > | **HalfCheetah-v4**          | 3061 | 2001 | 1731 |  2098 | 2294 |
> > | **Ant-v4**                  | 2910 | 2277 | 2452 |  1776 | 1871 |
> > | **Humanoid-v4**             | 5273 |  588 |  700 |  2580 | 2887 |
> > | **MinAtar/Asterix-v1**      |    1 |    2 |    2 |    17 |   15 |
> > | **MinAtar/Breakout-v1**     |   37 |   14 |    9 |    30 |   32 |
> > | **MinAtar/Freeway-v1**      |   32 |   36 |   26 |    59 |   58 |
> > | **MinAtar/Seaquest-v1**     |   14 |    3 |    5 |    51 |   47 |
> > | **MinAtar/SpaceInvaders-v1**|  204 |   44 |   54 |   193 |  132 |

---

> > > ### Author Response · Authors · 2024-12-02
> > > **Response 1/2**
> > >
> > > Dear Reviewer,
> > >
> > > We thank you again for the follow up response. We believe you are responding to quotes from our rebuttal without taking into account their context, including our previous interactions with you. We answer in the hope that it clears up the remaining misunderstandings regarding the experiments in our paper, especially, regarding your repeated queries to benchmark on Atari, which is not currently supported by our algorithm.
> > >
> > > **StaQ performance on Classic Control/MinAtar hard exploration tasks.** As we have already discussed with you and other reviewers, as well as in the paper through a dedicated experiments section (Sec. 6.4), StaQ underperforms on hard exploration tasks. This underperformance seems to extend generally to algorithms exploring with softmax policies since PPO/TRPO have similar struggles. Some discussion of the impact of softmax vs deterministic policies on exploration is given in App. B2 of the M-DQN paper ([Vieillard et al. 2020](https://arxiv.org/pdf/2007.14430)). In the Classic Control testbed, this includes MountainCar which is known to be a hard exploration task (please see for example discussion in page 8 of [Ceron et al. 2021](https://arxiv.org/abs/2011.14826)) due to its sparse reward structure.
> > >
> > >
> > >
> > > In Sec 6.4 and Appendix B.5 of the paper, we performed additional experiments on MountainCar by switching to a deterministic $\epsilon$-greedy policy for exploration, resulting in improved performance (about half the seeds can now solve the task, up from none), although not reaching the best baseline on this task (M-DQN). We believe the remaining difference is explained by the rate at which we change the deterministic policy (once every 5000 steps), which perhaps does not sweep through sufficiently many deterministic policies compared to baselines (e.g. M-DQN which updates every four steps). Nonetheless, although exploration is an interesting challenge, we believe properly addressing it with a theoretically sound method is outside the scope of this paper. On tasks with easier exploration, our goal was to investigate whether the theoretically sound policy update of StaQ would lead to practical benefits in terms of performance and stability. As such, our quote
> > >
> > > > "The Classic Control tasks fulfill this need and we see greatly improved performance and stability over baselines."
> > >
> > > was to be understood in the context of tasks that are not hard-exploration (i.e. excluding MountainCar), and this context was provided in the sentence *just* before the quote you extracted. Excluding MountainCar, we still stand by this conclusion, since M-DQN/DQN can exhibit large instabilities even on these smaller dimensional tasks, and PPO/TRPO while being relatively stable do not perform as well as StaQ. In contrast, StaQ is both better performing (Fig. 6 and Table 2) and has better stability (see Appendix B.2 and Fig. 7), supporting the monotonic improvements of StaQ suggested by theory.
> > >
> > > These conclusions extend gracefully as the dimensionality of the tasks increases: on the smallest MuJoCo task (Hopper), StaQ behaves very similarly as in the Classic Control testbed. On higher dimensional tasks, we see less of a concentration of the ten seeds around the same performing policies. This can be explained by the hard exploration nature of some of the tasks (especially MinAtar ones) and the increased error in policy evaluation which adds an error floor---please see our last response to Reviewer 8HXA and our updated manuscript for more details on this topic.
> > >
> > > In terms of **aggregated performance**, we beat all baselines on 10 out of 14 tasks. Of the 4 tasks where StaQ underperforms, 3 are classified as hard exploration by [Ceron et al. 2021](https://arxiv.org/abs/2011.14826). For the remaining task, namely Asterix-v1, PPO/TRPO perform similarly badly and as such we suspect that although this task is not classified as a hard exploration one, exploration with softmax policies seems to still be problematic here. If we would only look at the MuJoCo tasks, as you seem to suggest, StaQ would outperform baselines on 5 out of 5 tasks, and although we agree that this would put our algorithm in the best of light, we believe the current choice of tasks is a more balanced selection to both demonstrate the unique aspects of StaQ and its limitations, which we believe is equally important to the research community. Especially since we put effort into understanding the failure cases through dedicated sections on both the shortcomings of policy evaluation (Sec. 6.3) and exploration with softmax policies (Sec. 6.4).

---

> > > > ### Author Response · Authors · 2024-12-02
> > > > **Response 2/2**
> > > >
> > > > ---
> > > >
> > > > **"Can we conclude that StaQ will also underperform DQN and M-DQN on the Atari benchmark?"** Our expectation for Atari and on any other MDP/POMDP not evaluated in this paper is that if the task requires ''deep exploration'' [(Osband et al. 2016)](https://arxiv.org/pdf/1602.04621), then the softmax exploration will likely lead StaQ to underperform compared to M-DQN/DQN and more generally to algorithms exploring with deterministic policies. Otherwise, we will likely see improved performance due to the averaging over past Q-functions that stabilizes learning.
> > > >
> > > > ---
> > > >
> > > >
> > > > **"Persistent reluctance of the Authors to test on Atari".** As stated in the paper (Line 366), the Stacked NN architecture that allows us to evaluate StaQ in the high $M$ regimes with reasonable scaling is only compatible with MLPs but not with CNNs for now. As such, it is not a reluctance on our part, but testing on Atari tasks, or on tasks requiring a CNN is simply not possible. While demonstrating scaling to higher and higher dimensionality tasks is an interesting research area, we believe improving the stability of deep RL on small to medium sized tasks is an equally valid research topic and many real life decision problems are not necessarily at the scale of Atari games. Overall, we believe that the experiments we have done adequately support our theoretical results, with sufficient quantity on the number of environments solvable by an MLP, and show clearly both the merits of our contributions and their limitations.
> > > >
> > > > ---
> > > >
> > > > **"My score does not reflect my concerns over the theoretical contribution".** We are glad that you have no concerns regarding our theoretical contributions. Our concern however by bringing up this point, was whether your review fairly evaluated all aspects of our work, and in which theoretical results are an important part. The fact that (i) you do not mention the theoretical contributions in your review, (ii) your focus on the Atari benchmark, which is not supported by our algorithm, and (iii) the fact that you have lowered your score post-rebuttal because of the experiments despite us *adding* ablation studies you requested, further reinforces our feeling of an incomplete review.
> > > >
> > > > Nonetheless, we are sincerely grateful for the time you have spent reviewing our work, and we are glad we have addressed most of your concerns.

---

> ### Comment · Reviewer_b4CY · 2024-12-03
>
> Thank you for your reply.
>
> *"As stated in the paper (Line 366), the Stacked NN architecture that allows us to evaluate StaQ in the high regimes with reasonable scaling is only compatible with MLPs but not with CNNs for now."*
>
> I did not note this line when reading through the paper. I believe it is very important to mention this in abstract as well as the experiments section. In the experiments section, link it to the reason for not conducting experiments on complex testbeds, instead of only referring to [Obando-Ceron et al. 2021](https://arxiv.org/pdf/2011.14826). Since you have found a mistake in my message, I believe it is only fair that I undo my action on yesterday.
>
> Besides this, I still keep my initial concerns.
>
> "Of the 4 tasks where StaQ underperforms, 3 are classified as hard exploration by Ceron et al. 2021"
> "StaQ performance on Classic Control/MinAtar hard exploration tasks"
>
> They actually state: " It is worth highlighting that both Seaquest and Freeway appear to lend themselves well for research on exploration methods, due to their partial observability and reward sparsity."  There is no statement of a hard exploration task. It is not common practice to name a task that vanilla DQN can solve a "hard" exploration task. Hard exploration tasks are [Minigrid](https://minigrid.farama.org/index.html) or Montezuma's revenge or Vizdoom. This has been the problem from the beginning in this discussion, where you overstate both the significance of the evaluation environments and of StaQ's performance. I see that this is further done in Section 6.4:
>
> "StaQ is generally competitive with the state-of-the-art, but it might fail on some environments such
> as MountainCar or a few MinAtar environments. "
>
> Could the authors please explain which algorithms are state-of-the-art? For Mujoco, [TRPO](https://arxiv.org/abs/1502.05477) is from 2015, and [PPO](https://arxiv.org/abs/1707.06347) is from 2017. How did the authors reach the conclusion that StaQ is competitive with the state-of-the-art? Also the phrase "but it might fail on some environments such as MountainCar or a **few** MinAtar envronments" is quite confusing. You test on 5 MinAtar environments, where StaQ significantly underperforms on 3/5 environments. In other words, it fails in the MinAtar domain. As I said before, the performance and evaluation claims do not make sense to me.

---

> ### Author Response · Authors · 2024-12-03
> **Response 1/2**
>
> Dear Reviewer, thank you for your response. We really appreciated the continued discussion on this.
>
> **Hardness of exploration.** Our use of the terminology "Hard Exploration" is relative to the full set of tasks that we consider, to contrast MountainCar/Freeway/Seaquest with e.g. the MuJoCo tasks, and to understand why StaQ (and PPO/TRPO) struggle on those environments. We could have simply called them "harder exploration" tasks rather than "hard exploration", if you will. Of course, there are tasks that are much harder in terms of exploration than MountainCar or these MinAtar tasks, and you named a few of them.
>
> ---
>
> **Exploration challenges specific to StaQ and Max Ent algorithms.** As you noted, M-DQN/DQN do not struggle on the MountainCar/MinAtar tasks, but they are still problematic for StaQ. Although these problems are not the hardest, they still have a sparse reward structure, which makes it challenging in terms of exploration for StaQ and any algorithm following the maximum entropy principle: if a reward cannot be observed from the initial uniform policy, then this initial uniform policy acts as an attractor because the agent observes no reward and the initial uniform policy is the one maximizing entropy.
>
> This is precisely what happens in MountainCar. One can easily check on this problem that if you sample actions uniformly at random you can do millions of steps and you might never observe a reward. As such, due to the maximum entropy principle, StaQ prematurely converges to the uniform policy and never leaves this local optimum. In contrast, algorithms that explore with deterministic policies seem to still eventually achieve some rewards that enable learning (completely by chance, as none of the baselines implement a theoretically sound exploration method).
>
> ---
>
> **Why we include harder exploration tasks while it is not the focus of the paper.** As we already said, any behavior policy can be combined to StaQ to properly solve the exploration problem, but since exploration in deep RL is still pretty much an open question we thought it better to focus only on our main contribution which is the policy update. We believe this is valid as there are still many tasks that have dense rewards (e.g. MuJoCo tasks), and thus easier exploration, but in which existing deep RL algorithm can still show unstable learning dynamics which is the main point we want to address in this paper. Nonetheless, we thought it good to still have a section (Sec 6.4 + App B.5) dedicated to exploration to emphasize, perhaps contrary to popular belief, that the maximum entropy principle is not a proper answer to the exploration/exploitation trade-off in RL, and that our paper only sees entropy regularization as a way to stabilize learning, not improve exploration.
>
> ---
>
> **Failure on a few MinAtar tasks.** If you see Fig. 12 in App. B.5, where we have tried several entropy bonus weights, you can see that with a lower entropy weight we can reach a return of about 50 on Freeway, not too far off from strongest baselines. So talking about failure of StaQ in this case is perhaps a bit strong as it seems to be more of a hyper-parameter tuning problem. But we agree with the general sentiment that StaQ underperforms in this set of tasks, and we do not see the same high mean low variance return plots as in the other testbeds.

---

> > ### Author Response · Authors · 2024-12-03
> > **Response 2/2**
> >
> > **Choice of baselines and their strength.** Our baselines were thoughtfully chosen to ablate various aspects of our policy update contribution. We first describe our motivations below, and then address the other potential baselines.
> > - Munchausen DQN (**M-DQN**): this is the strongest baseline of the bunch and a relatively recent algorithm (2020). Importantly, it uses very similar principles as StaQ, as it adds both an entropy bonus and KL-divergence regularization. The main difference with StaQ is that in M-DQN, KL-divergence is implicitly added as a bonus term when learning the Q-function, whereas StaQ performs a nearly exact (up to the deletion of the old Q-function) KL regularized update. When adding the log probability term (stemming from the KL) to the target Q-function, the authors of M-DQN were careful in clipping the value of the log probability to avoid numerical problems as it can be unbounded (see page 6 of [Vieillard et al. 2020](https://arxiv.org/pdf/2007.14430)), and this trick is not covered by their theory. In contrast, although we do the KL regularized update approximately (due to the deletion of the old Q-function), we do a thorough analysis of this deletion and show that it is perfectly sound to do so. The algorithm we evaluated is thus very close to its theoretical version, and we thought it would be interesting to compare these two approaches to see if the theoretical guarantees translate in noticeable practical differences.
> > - **DQN**: This baseline is interesting because it shows the impact of entropy/KL regularization of M-DQN over the vanilla version. We can see that while M-DQN generally outperforms DQN, the regularization does not seem to stabilize learning all that much and large performance oscillations are present in both DQN and M-DQN, whereas the stabilizing effect of entropy/KL regularization is much more noticeable with the implementation of StaQ.
> > - **TRPO/PPO**: As you said these are older baselines, but they are state-of-the-art in terms of stability as can be seen in Fig. 7, where they are generally much more stable than M-DQN/DQN. TRPO in particular is a state-of-the-art way of performing KL regularization through a very clever computation of the natural gradient that is scalable to deep RL, and we thought it would be extremely relevant to add it as a baseline. PPO is more heuristic but is both popular and considered a lightweight, sometimes better performing, version of TRPO.
> >
> > Overall, we believe our choice of baselines covers a wide range of approaches to entropy/KL regularization in deep RL: through a bonus term (M-DQN), following the natural gradient (TRPO) or with a clipping loss (PPO). If we missed important baselines in this domain we would be happy to take suggestions. Of course these baselines can be extended by mixing in various components such as Noisy Nets ([Fortunato et al. 2017](https://arxiv.org/pdf/1706.10295)) and distributional RL ([Bellemare et al. 2017](https://arxiv.org/pdf/1707.06887)) leading into, for example Rainbow ([Hessel et al. 2017](https://arxiv.org/pdf/1710.02298)) and its derivatives. But these components are orthogonal to our proposed policy update method and could be readily integrated into StaQ. Since the focus of our paper is a policy update with strong theoretical motivations, we believe mixing in these components would obscure the empirical analysis of the policy update.
> >
> > As stated throughout the paper and the discussions, we believe that some of these components, especially those focused on exploration, could be implemented in a theoretically sound way into StaQ, and that would be an interesting direction for future work.

---

### Official Review · Reviewer_oaMK · 2024-10-28

**Soundness:** 3
**Presentation:** 2
**Contribution:** 2
**Rating:** 6
**Confidence:** 3

**Summary:**

The authors proposed Stacked NNs (SNNs), stacking weights with respect to iterations and theoretically analized the convergence resultsof Policy Mirror Descent (PMD) algorithms with SNNs. They proposed StaQ using these concepts (PMD with SNNs) and validated its effectiveness by experimental results.

**Strengths:**

1. Theoretically Solid Approach
2. Simple Algorithm
3. Outperforming Experimental Results

**Weaknesses:**

1. Only valid for discrete actions
2. More Weights (More memory)
3. InComplete Experiemtal Design Yet

**Questions:**

**Q1:** There are numerous tasks, such as Meta-world (https://arxiv.org/abs/1910.10897), to validate the stability-plasticity dilemma. Thus, I am curious whether the MuJoCo tasks chosen by the authors are truly appropriate for studying this dilemma. For instance, in my personal opinion, many of the MuJoCo environments exhibit flat regions and necessitate the acquisition of new skills, as like Acrobat and CartPole. Could the authors elaborate on why these tasks are deemed suitable for investigating the stability-plasticity dilemma?

**Q2:** Given that Table 2 in the supplementary material indicates a memory size of 300, it would be beneficial to specify the inference and training speed of the proposed method in comparison to other Q-learning algorithms.

**Q3:** While the authors analyzed stability throughout the paper to prevent catastrophic forgetting, it appears that plasticity has not been adequately addressed. Someone may think that the remarkable performance may be an effect by emphasizing stability. Indeed, I think that some tasks (e.g., MuJoCo, Acrobat, CartPole) used are relatively static and do not require the development of new skills. I would appreciate the authors’ insights on this matter.

**Q4:**  The paper referenced in (http://proceedings.mlr.press/v139/peer21a/peer21a.pdf) considered the ensemble of $Q$-functions, and reported a reduction in bias as illustrated in Figure 5. It may be beneficial for the authors to compare the proposed StaQ approach with it.

---

> ### Author Response · Authors · 2024-11-19
>
> Dear Reviewer oaMK,
>
> Thank you for your valuable feedback. We address each comment in detail to the best of our ability but would like some clarification on Q1 to make sure we are providing an adequate answer.
>
> ---
>
> **Q1/Q3. Plasticity vs stability and suitability of the RL tasks**
>
> **A1.** StaQ is composed of an explicit Q-function approximated by a neural network, and an optimization-free actor repesented by an average of past Q-functions. First of all, we note that there is no loss in plasticity compared to the state-of-the-art for the Q-function in StaQ. Each $Q_{\tau}^k$ is trained independently, thus we introduce no restriction or change to the learning procedure and use a standard MLP architecture for the Q-function. Stability is introduced in the actor, through the averaging over past Q-functions, which is a standard entropy-regularized Policy Mirror Descent (PMD) operation that is well understood and justified theoretically. While it will slow down the rate of change of the policy depending on the weight of the KL-regularizer that acts as the learning rate, in no way does it compromise plasticity or performance and wouldn't prevent convergence to an optimal policy.
>
> We do not understand exactly what you mean when stating that the tasks are
> >relatively static and do not require the development of new skills
>
> To us, this sounds like an exploration problem that is conducted by the behavior policy that ensures that all state-action pairs have been sufficiently sampled. It is an orthogonal challenge to the stability of the policy update tackled in this paper (see also the discussion in A3 for Reviewer U8vT). Provided that exploration is handled well, and since we impose no restriction to the learning of the Q-function, we do not believe there is any conceptual hurdle preventing StaQ from ''developing new skills''.
>
> However, we might be misunderstanding your question, especially since you cite Meta-World. Perhaps you mean the development of new skills in the continual RL sense, as in a reward function that changes with time? That would still not alter the convergence of PMD, although it may slow it down if the new reward function is too different from previous one. Nonetheless, our paper only covers single-task RL with a fixed reward function. We chose the tasks in our paper based on suggestions and common usage in prior work, as discussed in Line 414. Namely, we have used all 9 tasks suggested by [Ceron et al. 2021](https://arxiv.org/abs/2011.14826) for comparing discrete action deep RL algorithms, and have included the MuJoCo environments as additional standard RL tasks. What new insights do you believe the Meta-World tasks specifically would add to our experimental evaluation?
>
> ---
>
> **Q2. Inference and training speed.**
>
> **A2.** We will update the paper and add a table detailing the inference and training speeds. Thank you for the suggestion.
>
>
> ---
>
> **Q4. Ensemble Bootstrapping for Q-Learning.**
>
> **A3.** The referenced paper ([Peer et al. 2021](https://arxiv.org/pdf/2103.00445)) describes EBQL, an extension of Double Q-Learning ([Hasselt et al. 2015](https://arxiv.org/pdf/1509.06461)) to an ensemble of arbitrary size. We would like to emphasize that EBQL is an ensemble over the current Q-functions, to reduce bias in policy evaluation, whereas the policy update step of StaQ is averaging over past Q-functions, to stabilize policy update. These components are independent, and as mentioned in A1, any policy evaluation algorithm can be used in conjunction with StaQ, including EBQL.
>
> In our policy evalution we do use an ensemble of two (current) Q-functions in StaQ, but it is not sufficient to stabilize learning. This instability can be seen e.g. in the ablation of Fig. 3b with $M=1$, which removes our contribution to the policy update and only leaves the effect of the ensembling over Q-functions. Currently, we have only presented the $M=1$ ablation for Acrobot, however we plan to add these plots for all MuJoCo tasks in the Appendix.
>
> It would be interesting to explore the impact of different policy evaluation mechanisms, including [EBQL](https://arxiv.org/pdf/2103.00445), [CrossQ](https://arxiv.org/pdf/1902.05605), [EMIT](https://openreview.net/pdf/128a9d983bfa0d3001505e1f06d240b75230d1ad.pdf) and many more on the policy update of StaQ. The stability of the policy update in StaQ makes it a good testbed, as it should reduce learning noise and leave only differences in policy evaluation algorithms. However, we leave this to future work as we believe that these improvements are independent of the policy update proposed here.

---

> ### Comment · Reviewer_oaMK · 2024-11-20
>
> Thank you for your comments.
>
> **The reason I highlighted Meta-World tasks** was that the plasticity-stability dilemma is frequently studied in the tasks (e.g., https://arxiv.org/pdf/2209.13900). Developing a method to tackle Meta-World tasks inherently requires addressing both plasticity and stability. Similarly, since your manuscript explicitly mentions the plasticity-stability dilemma, I expected that both aspects—plasticity and stability—would be carefully addressed throughout the work. However, from my perspective, the paper primarily focuses on stability in a single task, as the term "plasticity" is mentioned only in the Introduction, and it is unclear how plasticity is addressed in the context of a single task. Although the authors used the tasks proposed by Ceron et al. 2021, it is necessary to explain whether these tasks are truly suitable for studying the dilemma. I am not insisting that experiments should be conducted specifically in the Meta-World environment but I think it is important to clarify this point.
>
> Please let me know if there is any part I may have missed.

---

> ### Author Response · Authors · 2024-11-20
>
> Dear reviewer,
>
> Thank you for your quick reply.
>
> **Is stability of StaQ at the expense of performance?** Plasticity in the Continual Learning (CL) literature is measured by performance on the task [(De Lange et al., 2021)](https://arxiv.org/pdf/1909.08383). We completely agree with you that stability could have been at the expense of performance, but clearly this is not the case for StaQ as we have more stable learning and good performance compared to baselines. This good trade-off is not surprising for parameter isolation methods in the CL literature (please see Sec. 6 in [De Lange et al., 2021](https://arxiv.org/pdf/1909.08383) for comparison of CL approaches), and the cost of parameter isolation methods is rather computational.
>
> Now, if the question is whether the RL tasks are challenging enough to be used as RL baselines, we point out again the work of [Ceron et al. 2021](https://arxiv.org/abs/2011.14826) that proposes a set of 9 tasks to compare discrete action deep RL algorithms, which we have all used. To that we have added 5 Mujoco tasks (Hopper, HalfCheetah, Walker, Ant and Humanoid) which are common in the deep RL literature. We believe that results on these 14 tasks support our claim with a certain level of confidence that the stability of StaQ is not at the expense of its performance, and complement our theoretical contributions that showed that the Policy Mirror Descent-like algorithm that we proposed does not limit performance as it converges to the optimal policy.

---

> ### Comment · Reviewer_oaMK · 2024-11-21
>
> Thank you for your detailed explanation.
>
> I agree that the StaQ method has successfully improved performance per each single task. However, since the tasks used in the study are not frequently utilized in Continual RL, as far as I know, I believe it is necessary to justify why these specific tasks were chosen to address the dilemma that commonly arise in Continual RL. Although the paper by Ceron et al. (2021) mentions that these tasks provide valuable scientific insights, it does not mean that these tasks are suitable for validating each algorithm  in terms of the dilemma.
>
> I am curious to hear the authors' opinions on my thoughts above.

---

> ### Author Response · Authors · 2024-11-24
>
> Dear Reviewer,
>
> Thank you so much for following up on the discussion, as we believe we have now identified the source of misunderstanding. In short, this paper is not about Continual RL but about looking at the policy update step in single-task deep RL as a Continual Learning (CL) problem, which opens up a range of novel approaches in the deep RL context. We hope the detailed answer below alleviates any confusion regarding this point. We have also updated the paper (see blue text) to clarify this distinction for the readers.
>
>
> **Are tasks suitable for Continual RL?** This paper is *not* about Continual RL. For instance, in the paper you cited above [(Wołczyk et al., 22)](https://arxiv.org/pdf/2209.13900), a learner is presented a sequence of Markov Decision Processes (MDPs) and the authors investigate if training on one MDP could lead to knowledge transfer to the next MDP. In our setting, a learner sees only one MDP and its performance is only measured on this one MDP. Studying whether knowledge transfers among tasks and if knowledge is retained on older tasks is beyond the scope of this paper.
>
> Despite being limited to a single MDP, single-task RL has still strong ties with CL because of the sequential nature in which data arrives. As mentioned in our paper, we are not the first to draw this parallel, the CL survey of [Lesort et al. 2020](https://arxiv.org/pdf/1907.00182) discusses this similarity (please see specifically Sec. 2.4.3). Drawing this connection is interesting because it opens up a plethora of CL methods that are not well researched in the deep RL context, but are applicable *even in a single task setting*. Specifically, in this paper we focus on the entropy regularized policy update problem described below (Eq. 6 of the paper)
>
> $$
> \text{for all } s\in S, \quad \pi_{k+1}(s) =\underset{p\in \Delta(A)}{\arg\max} \bigg[ Q_\tau^k(s) \cdot p - \tau h(p) -\eta \text{KL}(p;\pi_k(s)) \bigg].
> $$
>
> The objective of the update can be seen as CL, as we receive a new ''task'' which is to find $p$, a maximum entropy distribution over actions that puts its largest mass on actions with high Q-values, yet, through the KL-divergence term above, we do not want $p$ to differ too much from $\pi_k$, and forget the solution of the previous ''task''. Because of this similarity with CL, existing methods to solve this problem can be categorized with the CL literature lens, for example: [Lazic et al. 2021](http://proceedings.mlr.press/v139/lazic21a/lazic21a.pdf) used a rehearsal method (replay buffer/experience replay in deep RL terminology) to tackle the above policy update, while [Schulman et al., 15](https://arxiv.org/abs/1502.05477) uses a parameter regularization approach. While these cover two of the three main classes of CL methods [(De Lange et al., 2021)](https://arxiv.org/pdf/1909.08383), the novelty of this paper is in investigating a method pertaining to the third class (parameter isolation) to tackle this problem, as this class of methods has strong performance in CL benchmarks [(See Sec. 6 of De Lange et al., 2021)](https://arxiv.org/pdf/1909.08383), yet remains largely understudied in deep RL. We have updated the manuscript to clarify this, towards the end of Sec. 3 (blue text).
>
> Finally, as we are only looking at single task RL problems, any good benchmark for single task deep RL is a good benchmark for our work. In CL terminology, measuring plasticity amounts to measuring performance on the task, which in deep RL is usually done by periodically evaluating the policy return as we do in our experimental evaluation. In addition, we also measure stability by looking at the intra-seed performance variation with time. We have clarified this at the beginning of the Experiments section (blue text).
>
> We hope these explanations and changes in the text resolve the confusion on the nature of our work and we thank you for your feedback.

---

> ### Author Response · Authors · 2024-11-24
>
> **A2. Training and inference speed.** Following up from one of our previous responses, we have now updated the manuscript to include the training time and inference speed of StaQ, as a function of memory size $M$ and state space dimension, also comparing it to the baselines. This is mentioned in Table 1 (page 8) in the main text and detailed in full in App. C (page 32), in red text. As expected, the training and inference times of StaQ do not have a significant dependence on $M$ or state space dimension, as the SNN computes all of the past Q-functions in parallel.

---

> ### Comment · Reviewer_oaMK · 2024-11-24
>
> Thank you for your response.
>
> I now fully understand the misunderstanding I had earlier. In continual reinforcement learning, some approaches address catastrophic forgetting by employing multiple networks. Initially, I was unclear why the authors, despite referencing the main challenge of continual learning (CL) as their motivation, did not focus on addressing task transitions. After the explanation, now I convince that the authors have skillfully leveraged the concept of CL to develop a well-grounded theoretical framework under a clear motivation for single-task settings. They have formulated their approach in a manner that demonstrates significant improvements in experimental performance. Thus, I increase my evaluation score.
>
>
> I would kindly suggest considering a slight revision of the manuscript to address potential misunderstandings, similar to the one I initially had, to ensure clarity for future readers.

---

> ### Author Response · Authors · 2024-11-28
>
> Dear Reviewer,
>
> Thank you for your positive response. Following your suggestion we have incorporated, in the final revision of the paper, our last discussion into Appendix E. We have also more clearly stated that we are addressing the single-task case, see line 215 in blue of the main paper. In addition, we have added how we understand stability-plasticity dilemma in the single-task case, see lines 404-406 in blue of the latest version.

---

### Official Review · Reviewer_8HXA · 2024-10-31

**Soundness:** 4
**Presentation:** 3
**Contribution:** 3
**Rating:** 6
**Confidence:** 4

**Summary:**

This paper proposes StaQ, a deep reinforcement learning (RL) algorithm inspired by continual learning (CL) techniques. StaQ addresses the instability often observed in deep RL training by using a policy update mechanism based on Policy Mirror Descent (PMD) with a growing neural architecture. This architecture, called Stacked Neural Networks (SNNs), stores and averages a large number (M) of past Q-functions, effectively stabilizing learning by smoothing out the policy updates.

The main idea in the paper is to leverage the stabilizing effect of averaging Q-functions in PMD. PMD updates the policy based on a weighted average of past Q-functions. StaQ approximates this by storing the last M Q-functions in the SNN and using them for the policy update. Moreover, the authors claim that the SNN architecture allows for efficient computation of these averages on GPUs.

The algorithm also includes a theoretical analysis showing that for sufficiently large M, a modified version of finite-memory PMD converges to the optimal policy. Experiments on various benchmark tasks demonstrate that StaQ achieves competitive performance compared to standard baselines like PPO, DQN.

**Strengths:**

1) Integrating the SNN architecture with PMD is a novel approach that effectively addresses the instability issue in deep RL.
2) The authors provide a theoretical analysis of the convergence properties of the finite-memory PMD, offering insights into the algorithm's behavior.
3) The experimental results demonstrate the effectiveness of StaQ in achieving stable and competitive performance on MinAtar and Mujoco envs.
4) The policy update is optimization-free, simplifying the overall algorithm and potentially increasing stability. (in discrete cases)

**Weaknesses:**

1) While the policy is stabilized, the Q-function learning still suffers from catastrophic forgetting, requiring ad-hoc workarounds like epsilon-softmax exploration.
2) The reliance on entropy regularization for exploration may not be sufficient for challenging exploration tasks, as evidenced by the performance on MountainCar?
3) The current SNN implementation only supports MLPs, limiting its applicability to tasks requiring other architectures like CNNs or LSTMs.

**Questions:**

1) The theoretical analysis assumes exact Q-function computation. How does the approximation error introduced by using neural networks affect the convergence guarantees in practice?
2) Moreover, the paper mentions using the min or mean of two target Q-functions. What is the motivation behind this choice, and how does it impact performance in different environments? Maybe this is already discussed in the paper and I missed it?
3) How does the computational cost of StaQ scale with the memory size M and the size of the state-action space?
4) Could alternative CL techniques for mitigating catastrophic forgetting in the Q-function, such as rehearsal methods or regularization-based methods, be integrated into StaQ?
5) Just curious: How can the SNN architecture be extended to support other types of neural networks like CNNs and RNNs? Have the authors thought about this (since its mentioned as a potential future direction)

---

> ### Author Response · Authors · 2024-11-19
>
> Dear Reviewer 8HXA,
>
> We thank you for your comprehensive review and interesting questions. We answer them in detail below.
>
> **Q1. Practical effects of approximation error in Q-function.**
>
> **A1.** In previous theoretical studies of approximate PMD, it was shown that in the presence of approximation error in the Q-function, linear convergence of PMD is maintained up to reaching an error floor that depends on the Q approximation error. While these results do not take into consideration the deletion of past Q-functions introduced in StaQ, it is important to note that up to the first deletion, StaQ is an exact PMD algorithm as far as policy update goes. In our experiments, this means that for $M=300$, and with 5000 steps per iteration, for the first 1.5 million steps, there is no deletion and StaQ is using an exact PMD policy update. In Fig. 6, we can see that peak performance is reached at 1.5 million steps for some tasks but not others; however, as it can be seen in our experiments, deleting Q-functions has no noticeable effect on learning in all considered cases, and does not introduce any perceptible changes in the learning dynamics compared to the first 1.5 million ''exact steps''. Thus, we believe it is fair to say that StaQ behaves almost exactly like a true PMD policy update, and inherits its robustness to noise in the Q-function..
>
> ---
>
> **Q2. Motivation and impact of mean vs min averaging of Q targets.**
>
> **A2.** In the literature both approaches exist: in a discrete-action setting, use of the minimum of the target Q-functions was introduced in Maxmin Q-Learning ([Lan et al. 2020](https://arxiv.org/pdf/2002.06487)), and is also common in actor-critic algorithms such as TD3 ([Fujimoto et al.](https://arxiv.org/pdf/1802.09477v3)) and SAC ([Haarnoja et al.](https://arxiv.org/pdf/1801.01290)). On the other hand, taking the mean of the target Q-functions is the more traditional interpretation of an ensemble, and was used for example in the paper [Peer et al. 2021](https://arxiv.org/pdf/2103.00445) suggested by Reviewer oaMK. In our paper we treat this choice as a hyperparameter and chose the overall best for each class of environment (Classic Control, MuJoCo, MinAtar). As discussed in App. B4 (under the heading "Policy evaluation and exploration"), we find that the minumum generally results in more stable training, but struggles with weak reward signals due to its tendency to underestimate. Our observations here are in line with the literature on the impact of over/underestimation bias of the Q-function, see e.g. Sec. 3 of ([Lan et al. 2020](https://arxiv.org/pdf/2002.06487)).
>
>
> ---
>
> **Q3. Scaling of StaQ with M and state-action space.**
>
> **A3.** We will shortly add a table showing the runtime of StaQ for different choices of M and different settings of the state-action space. We will complete our answer then.
>
> ---
>
> **Q4. Can Continual Learning (CL) techniques be used in the Q-function of StaQ.**
>
> **A4.** Yes, we impose no restriction on policy evaluation, and virtually any approach that would improve the accuracy or stability of the Q-function can be used in StaQ, including a better management of the replay buffer, as discussed in the future work section. Regularization techniques have also been tried in the past, for instance in the OpenAI implementation of TRPO/PPO, it was proposed to use a trust region on the value function (See Eq. (29) of [Schulman et al. 16](https://arxiv.org/pdf/1506.02438)), and this could also be used in our setting. True to the motivation of our paper, we think however that parameter isolation methods remain poorly studied in RL, and are a potent alternative as they improve stability without compromising plasticity, and the current compute power can largely support their cost. Along these lines, a potentially interesting CL idea (that of splitting the learning between a fast and slow component) is discussed in the future work section.
>
> ---
>
> **Q5. SNNs for CNN and LSTM layers.**
>
> **A5.** Recurrent networks such as LSTMs or GRUs are based on matrix multiplication operations that are similar to MLPs. It should be rather trivial, at least conceptually, to support these operations in a stacked neural architecture by stacking the matrices of several recurrent networks and performing these matrix operations in batch, as with the provided code. However, implementations of recurrent architectures for processing a sequence of inputs are usually coded in a lower level language (in C for instance). Contrary to MLPs, we would need to alter automatic differentiation libraries at a lower level to properly support stacking of recurrent models. But this is an engineering problem, not a conceptual one. For CNNs, we believe the ```groups``` option in the ```Conv2d``` class of PyTorch can be used to implement stacked networks, although we have not tested it yet.

---

> ### Author Response · Authors · 2024-11-23
>
> Dear Reviewer, following up on **Q3.**, we have now updated the manuscript to include the training time and inference speed of StaQ, as a function of memory size $M$ and state space dimension, also comparing it to the baselines. Please see this mentioned in Table 1 (page 8) in the main text and detailed in App. C (page 32), in red text. As expected, the training and inference times of StaQ do not have a significant dependence on $M$ or state space dimension, as the SNN computes all of the past Q-functions in parallel.

---

> > ### Comment · Reviewer_8HXA · 2024-11-25
> >
> > Thank you for providing some detailed answers to my questions and adding in some extra details - such as for Q3. This gives me more clarity of the work and I believe this would be a good paper to the community. I have read the other reviews and their opinions and would like to maintain my score.

---

> ### Author Response · Authors · 2024-11-28
>
> Dear Reviewer, thank you again for your feedback. We are glad to hear that you believe this would make a good paper for the community.
>
> Following our discussion about **Q1**, on the impact of a Q-function approximation error on convergence, we have updated our manuscript. We have generalised the theoretical analysis to include the case where the policy update is performed using an approximation $\tilde{Q}\_\tau^k$ of the Q-function $Q\_\tau^k$, which is bounded by $||\tilde{Q}\_\tau^k - Q\_\tau||\_\infty \le \epsilon\_{eval}$. This new theoretical analysis complements our first answer to **Q1**, that similarly to previous theoretical studies of approximate PMD, the linear convergence of StaQ is maintained until it reaches an error floor that depends on the Q approximation error $\epsilon_{eval}$.
>
> In practice, and to complement our previous answer to Q1, we can see on the Classic Control tasks and Hopper, where perhaps $\epsilon_{eval}$ is very low, that the final variance in policy return between runs is extremely small and all seeds seem to converge to similarly performing policies. However, as the task dimensionality grows, the error floor due to $\epsilon_{eval}$ increases and the policies do not concentrate as much on similarly performing policies.
>
> Regarding the new proofs, since our contributions are focused on the policy update, we have left the statements in the main paper unchanged, analyzing only the error introduced by the deletion mechanism in the policy. But the proofs in App. A now treat the more general case of a presence of both an error in policy update and policy evaluation, as formalized by the new text in brown color at the start of the first theorem's proof. The two error types do not interact, and the policy evaluation error only adds an independent error floor as previously analyzed in the approximate PMD setting [(Zhan et al. 21)](https://arxiv.org/abs/2105.11066). The results are thus not surprising but for the sake of completeness, we think they can still be interesting. Note that the old proofs can be recovered by setting the policy evaluation error $\epsilon_{eval} = 0$ and by replacing the approximate Q-function $\tilde{Q}\_{\tau}$ by the true Q-function $Q\_{\tau}$.

---

### Official Review · Reviewer_U8vT · 2024-11-04

**Soundness:** 3
**Presentation:** 2
**Contribution:** 3
**Rating:** 6
**Confidence:** 3

**Summary:**

This paper proposes a growing neural architecture (named StaQ) for a regularized policy iteration algorithm in the deep reinforcement learning domain. This dynamic algorithm makes use of all past Q-functions saved in memory to improve the policy by reducing catastrophic forgetting. To remain memory viable a convergent algorithm is derived from a fixed budged of Q-functions, resulting in smaller non infinite memory usage.

**Strengths:**

The clarity of the paper is good and can be easily understood. The proposed method to improve stabilty and mitigating catastrophic forgetting is relevant and a logical area of research in a reinforcement learning context. The chosen environments for evaluation make sense and provide strong validation for the StaQ method. The experiments are detailed and with multiple relevant baselines. The algorithm should be reproduceable with the given information. Limitaions and problems are explicitly acknowledged, specifically the exploration method, that can be improved. The Suggestions for future research provide a guide for logical future improvements and relevant research topics.
Overall this research could be a valuable addition for the reinforcement learning community.

**Weaknesses:**

Although the paper uses well-known baselines (DQN, TRPO, PPO), it also should include baselines like SAC and EWC, which were mentioned in the introduction chapter 1. While a rigorous mathematical foundation that clearly demonstrates the theoretical underpinnings of the proposed method is provided, there are instances where the density of formulas make the text hard to read. Reducing or summarizing some of the mathematical details in favor of intuitive explanations might improve readability. This is especially true for chapter 3 Preliminaries.

**Questions:**

Given that entropy regularization doesn’t solve the exploration problem in RL, could additional exploration strategies have synergistic effects with StaQ’s policy averaging?

---

> ### Author Response · Authors · 2024-11-15
>
> Dear Reviewer U8vT,
>
> Thank you for your valuable feedback. We address each comment in detail, one by one below.
>
> ---
>
> **Q1. No SAC or EWC baselines**
>
> **A1.** We would like first of all to remind the reviewer that StaQ is only compatible with discrete action spaces and hence we only use discrete-action deep RL baselines. We consider in the paper four baselines: TRPO, PPO, DQN and Munchausen DQN (M-DQN).
>
> **SAC.** Soft Actor Critic (SAC) is not compatible with discrete action spaces, but M-DQN can be seen as an adaptation of SAC to discrete action spaces with an additional KL-divergence regularizer. Please see the discussion in the M-DQN paper (https://proceedings.neurips.cc/paper/2020/file/2c6a0bae0f071cbbf0bb3d5b11d90a82-Paper.pdf) on page 3, between Eq. (1) and (2). The M-DQN paper also discusses Soft-DQN in Eq. (1), which they call the
> "most straightforward discrete-actions version of Soft Actor-Critic (SAC)", and which can be obtained from M-DQN by simply setting the KL-divergence regularization weight to zero. However, we did not consider Soft-DQN as a baseline because the M-DQN paper shows that M-DQN generally outperforms Soft-DQN.
>
> We also note that by setting $M=1$ in StaQ, we remove the KL-divergence regularization and only keep the entropy bonus. This baseline can also be seen as an adaptation of SAC to discrete action spaces: indeed, if we set $M=1$ in Eq. (14) we recover the policy logits
> $\xi_{k+1} = \frac{\alpha}{1-\beta^M}\sum_{i=0}^{M-1}\beta^i Q_{\tau}^{k-i} = \frac{\alpha}{1-\beta}Q_{\tau}^{k} = \frac{Q_{\tau}^{k}}{\tau}$ (because $\alpha \tau = 1-\beta$), such that $\pi_{k+1} \propto \exp\left(Q_{\tau}^k / \tau\right)$. While for SAC, the actor network is obtained by minimizing this problem (Eq. 4 in https://arxiv.org/pdf/1812.05905) $\pi_{k+1} = \arg\min \text{KL}\left(\pi \Big| \frac{\exp\left(Q_{\tau}^k/\tau\right)}{Z_{\text{norm.}}}\right)$, but in the discrete action setting, we can sample directly from $\exp\left(Q_{\tau}^k/\tau\right)$ --- which is the minimizer of the above KL-divergence term --- and we do not need an explicit actor network. As such StaQ with $M=1$ could be seen as an adaptation of SAC to discrete action spaces. Please note that we have already included $M=1$ in the ablation study for some environments.
>
> In summary, because M-DQN outperforms Soft-DQN and because StaQ with $M=1$ resembles SAC (as it uses only soft Q-functions and no KL-divergence regularization), we do not believe that Soft-DQN would be useful as an additional baseline, however we would be open to feedback on this.
>
>
>
> **EWC.** Elastic Weight Consolidation (EWC) is a continual learning approach for supervised learning problems, and is not designed to tackle instabilities specific to deep RL. However, as discussed in the paper (line 73), TRPO uses a similar natural gradient approach to online EWC (https://arxiv.org/pdf/1801.10112) and can be seen as online EWC applied to RL. TRPO (and thus a form of EWC) is used as a baseline and behaves rather well in terms of stability: on the stability plots in the Appendix (Fig. 7 and 8), TRPO is usually the second most stable algorithm after StaQ.
>
>
> ---
>
> **Q2. Clarity of Section 3**
>
> **A2.** We would appreciate if you could point out to specific equations in Sec. 3 that you found hard to read. We would be happy to improve the writing.
>
> ---
>
> **Q3. Synergy between additional exploration strategies and StaQ**
>
> **A3.** Addressing the exploration/exploitation dilemma is the key challenge of tabular RL. In the deep RL setting, there is the additional challenge of instabilities during training caused by the use of neural networks. Our paper focuses only on the latter challenge. However, we believe that the benefits of novel exploration strategies for deep RL would be more apparent if the lack of exploration is the only cause of under-performance, otherwise good exploration might be lost in the "noise" of performance oscillation and collapse, if instabilities remain in deep RL. We believe that exploration in the deep setting will be easier to approach if the research community could get rid of the instabilities of deep RL -- and we believe that this paper brings novel insights towards this goal.

---

> > ### Comment · Reviewer_U8vT · 2024-11-25
> >
> > Dear Authors,
> > sorry for the late response to your rebuttal.
> > Thank you for your detailed answers to the questions I had. Your thorough rebuttal has alleviated all my concerns regarding your paper.
> > Regarding Q2, this might be a personal opinion and not necessarily cause for not accpeting. It is not a specific equation that i find hard to read but the density in the chapter. When half the chapter is equations it is harder to follow the narative of the chapter and points that are made might not come across as well.

---

> > > ### Author Response · Authors · 2024-11-28
> > >
> > > Dear Reviewer, thank you for acknowledging and appreciating our answers.
> > >
> > > A small note on the final revision of the paper: we have added a few clarifications in the Section Preliminaries, to the extent that space constraints allowed. We hope that these additions improve readability.

---

### Author Response · Authors · 2024-12-04
**Summary of change**

We thank the reviewers for their detailed reviews and even lengthier discussions. Here are the main changes present in the final revision, color coded to address the different concerns of the reviewers over the initial submission.

- **Blue text:** Addressed presentation issues in Sec. 3 as requested by Reviewer U8vT, and to clarify the connection between our paper and Continual RL as requested by Reviewer oaMK.
- **Brown text:** Reviewer 8HXA asked about the behavior of the algorithm in the presence of a policy evaluation error. Our initial theoretical analysis only considered errors in the policy update caused by the finite memory constraint, but we now have extended the proofs in the appendix to also cover errors due to policy evaluation.
- **Red text:** We extended ablation studies for more environments as requested by Reviewer b4CY, added learning and inference timings for different values of M and state-action spaces as requested by Reviewer 8HXA and oaMK, and added a discussion showing the relation between SAC, Soft-DQN, M-DQN and StaQ to support discussion initiated by both Reviewer U8vT and b4CY.
- **Green text:** We changed the Fig. 3a to an environment that is less confusing for the readers and more representative of the general behavior of StaQ, as requested by Reviewer b4CY.

---

> ### Author Response · Authors · 2024-12-04
> **Concerns of Reviewer b4CY**
>
> Reviewer b4CY also expressed some concerns regarding the choice of the environments for our experiments. The main arguments of the reviewer were that environments except MuJoCo ones are either low-dimensional, like the Classic Control environments, or are not a common choice of benchmarks for Deep RL such as Minatar, especially considering that Atari is a more popular choice. Another reason mentioned by the reviewer is that, on some of those environments, StaQ has lower performance and is outperformed by DQN or M-DQN (unlike on MuJoCo tasks). In their last message, Reviewer b4CY also cast some doubt on the choice of our baselines and on them being state-of-the-art.
>
>
> We disagree with Reviewer b4CY and our arguments to support our decision in the choice of baselines or benchmarks are provided in our last set of replies to Reviewer b4CY.

---

### Meta-Review · Area_Chair_nttA · 2024-12-19

**Metareview:**

This paper proposes an RL algorithm, StaQ, inspired by continual learning. It dynamically grows its neural architecture to address the non-stationary nature of data.

This paper has a solid theoretical contribution and has a novel perspective of improving stability using ideas from continual learning. It also achieves competitive performance in the MuJoCo tasks.

However, reviewers shared concerned about the insufficient evaluation of the performance on more commonly used benchmarks and the lack of comparison to more recent baselines. The method also struggles in tasks that require exploration. Additionally, some reviewers are concerned on the limitation in discrete action space and MLP architectures.

**Additional Comments On Reviewer Discussion:**

There have been extensive discussion between authors and reviewers about the concerns in their initial reviews. The authors's rebuttal addressed some concerns including readability, the lack of comparison to some baselines and evaluation in continual learning tasks. However, one reviewer's concern on the choice of empirical performance on classical control tasks and its suboptimal performance is not well addressed (details in the "final comment" from reviewer b4CY). In the following AC-reviewer discussion, all reviewers reached a consensus on this concern, and agree this submission does not meet the criterion for publication at ICLR.

---

### Decision · Program_Chairs · 2025-01-22

Reject